# An autoimmune pleiotropic SNP modulates *IRF5* alternative promoter usage through ZBTB3-mediated chromatin looping

Zhao Wang [1,2,3,13] ✉, Qian Liang [2,4,13], Xinyi Qian [2,13], Bolang Hu [3,13], Zhanye Zheng [2], Jianhua Wang [1], Yuelin Hu [3], Zhengkai Bao [3], Ke Zhao [2], Yao Zhou [1], Xiangling Feng [2], Xianfu Yi [1,5], Jin Li [6], Jiandang Shi [7], Zhe Liu [8], Jihui Hao [9], Kexin Chen [10], Ying Yu [2], Pak Chung Sham [11], Wange Lu [7], Xiaoyan Wang [12] ✉, Weihong Song [3] ✉ & Mulin Jun Li [1,2,10] ✉

Genetic sharing is extensively observed for autoimmune diseases, but the causal variants and their underlying molecular mechanisms remain largely unknown. Through systematic investigation of autoimmune disease pleiotropic loci, we found most of these shared genetic effects are transmitted from regulatory code. We used an evidence-based strategy to functionally prioritize causal pleiotropic variants and identify their target genes. A top-ranked pleiotropic variant, rs4728142, yielded many lines of evidence as being causal. Mechanistically, the rs4728142-containing region interacts with the *IRF5* alternative promoter in an allele-specific manner and orchestrates its upstream enhancer to regulate *IRF5* alternative promoter usage through chromatin looping. A putative structural regulator, ZBTB3, mediates the allele-specific loop to promote *IRF5*-short transcript expression at the rs4728142 risk allele, resulting in *IRF5* overactivation and M1 macrophage polarization. Together, our findings establish a causal mechanism between the regulatory variant and fine-scale molecular phenotype underlying the dysfunction of pleiotropic genes in human autoimmunity.

Autoimmune comorbidity has been extensively documented across diverse human populations, yet the causal factors remain elusive[1,2]. Genome-wide association studies (GWAS) have identified hundreds of autoimmune disease-associated loci, half of which are shared by multiple traits[3,4], suggesting that genetic predisposition underlies comorbidity susceptibility and that genetic sharing is pervasive among autoimmune diseases. Cross-phenotype meta-analysis and pleiotropy estimation models have identified many pleiotropic variants associated with at least two distinct autoimmune diseases across the genome[5–7]. However, the true causal variants and their molecular mechanisms have not been systematically examined[8].

As the majority of pleiotropic loci are located in the noncoding genomic region, identifying the causal regulatory variants and

interpreting how they regulate gene expression, and then explaining comorbidity susceptibility continues to be a challenge[9]. Over the past decade, large-scale expression quantitative trait locus (eQTL) studies have been performed to explore gene expression associated with regulatory variants across most human tissues[10]. Emerging data and our previous resources show that eQTLs can be mediated through fine-scale molecular processes; for example, eQTLs can exert their genetic effects on gene expression by altering the chromatin state[11], chromatin interaction[12], transcription factor (TF) binding[13], promoter and transcript usage[14,15], or other transcriptional events[16,17]. These studies provide promising insights into deciphering the causal paths from cell-type-specific noncoding regulatory variants to different scales of molecular phenotypes, and will

A full list of affiliations appears at the end of the paper. ✉e-mail: wangzhao19880923@wmu.edu.cn; wangxiaoyan@csu.edu.cn; weihong@wmu.edu.cn; mulinli@connect.hku.hk

eventually facilitate our understanding of the genetic basis of autoimmune diseases.

The causal variants in shared genetic loci associated with multiple autoimmune diseases usually affect the pleiotropic genes and common immune-related pathways, such as the interleukin-23 (IL-23) pathway in the pathogenesis of autoimmune comorbidity, which many genetic studies have highlighted in particular[18]. Another pleiotropic gene, *IRF5* (interferon regulatory factor 5), has been associated with multiple immune-related traits and diseases[19]. The TF IRF5 regulates the inflammatory and immune responses, guides human monocytes toward M1 macrophage polarization, and plays a crucial role in human autoimmunity[20–23]. Several single-nucleotide polymorphisms (SNPs) and small insertions/deletions in the *IRF5* gene or regulatory regions have been validated to independently cause different autoimmune diseases by altering gene expression, splicing, and RNA stability[24–28], but whether there are other fine-scale molecular mechanisms that shape the pleiotropic effect at the *IRF5* locus requires in-depth investigation.

Through systematic annotation, prediction, and prioritization of known autoimmune disease pleiotropic variants and their target genes, we identify rs4728142 as a top causal regulatory variant whose genetic effects are shared by many autoimmune diseases. We also clarify that rs4728142 modulates the chromatin binding of a putative structural regulator, ZBTB3, and then affects *IRF5* alternative promoter usage by supervising a short-range chromatin loop between its upstream enhancer and the *IRF5*-short transcript promoter, which causes *IRF5* overactivation and monocyte/macrophage dysfunction.

## Results

### Functional annotation and target gene analysis of autoimmune disease-associated pleiotropic variants

Previous investigations on the shared genetic effect of autoimmune diseases have identified numerous pleiotropic loci[29]. To identify the underlying causal pleiotropic variants and molecular pathways modulating autoimmune comorbidity, we systematically curated genome-wide studies (including cross-phenotype meta-analysis and pleiotropy analysis based on different statistical models) to identify the potential pleiotropic or shared genetic loci associated with at least two autoimmune diseases. We gathered 440 sentinel pleiotropic variants for 24 autoimmune diseases from 12 genome-wide studies and performed systematic bioinformatics analysis (Fig. 1a, b, Supplementary Fig. 1a, and Supplementary Data 1 and 2). Seronegative immune-driven phenotypes, such as Crohn's disease (CD), ulcerative colitis (UC), ankylosing spondylitis (AS), and psoriasis (PS), showed the most extensive genetic sharing across the human genome (Fig. 1b). To evaluate whether the curated sentinel pleiotropic variants could be reproduced using public autoimmune disease GWAS data, we comprehensively collected 54 GWAS summary statistics from public resources (Supplementary Data 3) and performed variant-level pleiotropy estimation using gwas-pw (model 3)[30]. Although many previous GWAS summary data of immune-related diseases are not publicly accessible, we confirmed that 142 of the 440 curated pleiotropic variants shared genetic influences (posterior probability ≥0.9 in the shared genetic effect model) on more than one trait regardless of pleiotropy type (horizontal or vertical) and effect direction (Supplementary Data 4). To inspect whether these pleiotropic variants are likely causal variants for specific autoimmune diseases, we performed GWAS fine-mapping analysis of 48 fine-mappable autoimmune disease GWAS datasets (Supplementary Data 5). We found that 56.59% of the curated pleiotropic variants are located in the 95% credible sets of plausible causal variants for at least one trait (Supplementary Data 6), implying the functional importance of these variants.

Next, we annotated the pleiotropic single-nucleotide variations (SNVs) functionally by assigning the most severe consequence and diverse regulatory evidence (Fig. 1c and Supplementary Data 7). The analysis identified 18 missense SNVs and three splice-altering SNVs in known immune-related disease genes, such as *IFIH1*[27], with three missense variants, and *CARD9*, with both missense and splice-altering variants. As most of the pleiotropic variants are located in the noncoding genomic region, we retrieved their noncoding function scores and predicted their tissue/cell-type-specific regulatory potential. Using the regulatory trait concordance (RTC) score[31], we investigated the colocalization (RTC score ≥0.9) between the pleiotropic variants and eQTLs based on five blood-derived whole-genome sequencing (WGS) eQTL datasets (whole blood, lymphoblastoid cells, T cells, monocytes, and neutrophils). We found that of the 440 pleiotropic variants, 292 strongly colocalized with at least one blood *cis*-eQTL, indicating that these variants may directly regulate the expression of immune-related genes (Supplementary Data 8). We tested whether the pleiotropic variants are enriched in certain chromatin states or TF bindings for specific blood cell types. The results from the ChromHMM core 15-state model showed significant enrichments in the enhancer and promoter regions, particularly in CD4⁺ T cells, CD8⁺ T cells, CD14⁺ monocytes, and CD56⁺ natural killer (NK) cells (Supplementary Fig. 1b). TF enrichment analysis showed that several hematopoietic TFs (e.g., MYB, STAT1, GATA3, and SPI1), transcription and chromatin remodeling factors (e.g., BRD4, MED1, and CDK9), and inflammatory TFs (e.g., IRF1) were significantly enriched at pleiotropic variant loci with evidence from multiple cell types (Supplementary Fig. 1c).

We applied three complementary strategies to identify the candidate target genes of these pleiotropic variants, including the neighboring genes detected by DEPICT (v1_rel194)[32], colocalized genes obtained by RTC, and the interacting genes received by promoter capture Hi-C (PCHi-C) (Supplementary Data 9). For pleiotropic variants that may directly alter protein function or splicing, we directly assigned them to the affected genes. As expected, these pleiotropic genes were markedly enriched in the T-cell differentiation, virus infection, and immunological disease pathways using either merged sets or single set (Fig. 1d and Supplementary Fig. 1d–f). Using pleiotropic variant–target gene relationships on each autoimmune disease, we constructed a bipartite network and visualized the genetic architecture among the investigated autoimmune diseases by compressing all pleiotropic variants from the same autoimmune disease into one node (Fig. 1e). The seronegative immune-driven diseases (including CD, UC, AS, PS, and primary sclerosing cholangitis [PSC]) and inflammatory systemic disorders (e.g., systemic lupus erythematosus [SLE] and systemic sclerosis [SSC]) were separately clustered and pushed away from other autoimmune diseases with different genetic sharing profiles in the collapsed network, which is highly consistent with previous findings[29]. We also found that rheumatoid arthritis (RA) was clustered with celiac disease (CeD), type 1 diabetes (T1D), and multiple sclerosis (MS) with respective distinct shared genes (Fig. 1e). The results showed that 33.29% (251/754) of the identified pleiotropic genes were shared by at least three diseases, in which *SH2B3* was the top pleiotropic gene shared by 12 diseases, implying widespread genetic sharing among autoimmune diseases. To delineate the downstream molecular mechanisms of these pleiotropic variants, we inferred the causal relationships of the autoimmune disease-associated genes using *cis/trans*-eQTLs summary statistics information and constructed a gene regulatory network (Fig. 1f). We found that 54 *cis*-genes, 1093 *trans*-genes, and 12 *cis/trans*-genes were connected in this pleiotropy network. According to the omnigenic model[33,34], for example, the *DAP* gene is causally linked with diverse *cis/trans* effects and might function as a core gene that directly affects the survival of immune cells; the inflammatory TF gene *IRF5* could act as a peripheral gene that transmits its effect to the downstream core genes (Fig. 1f).

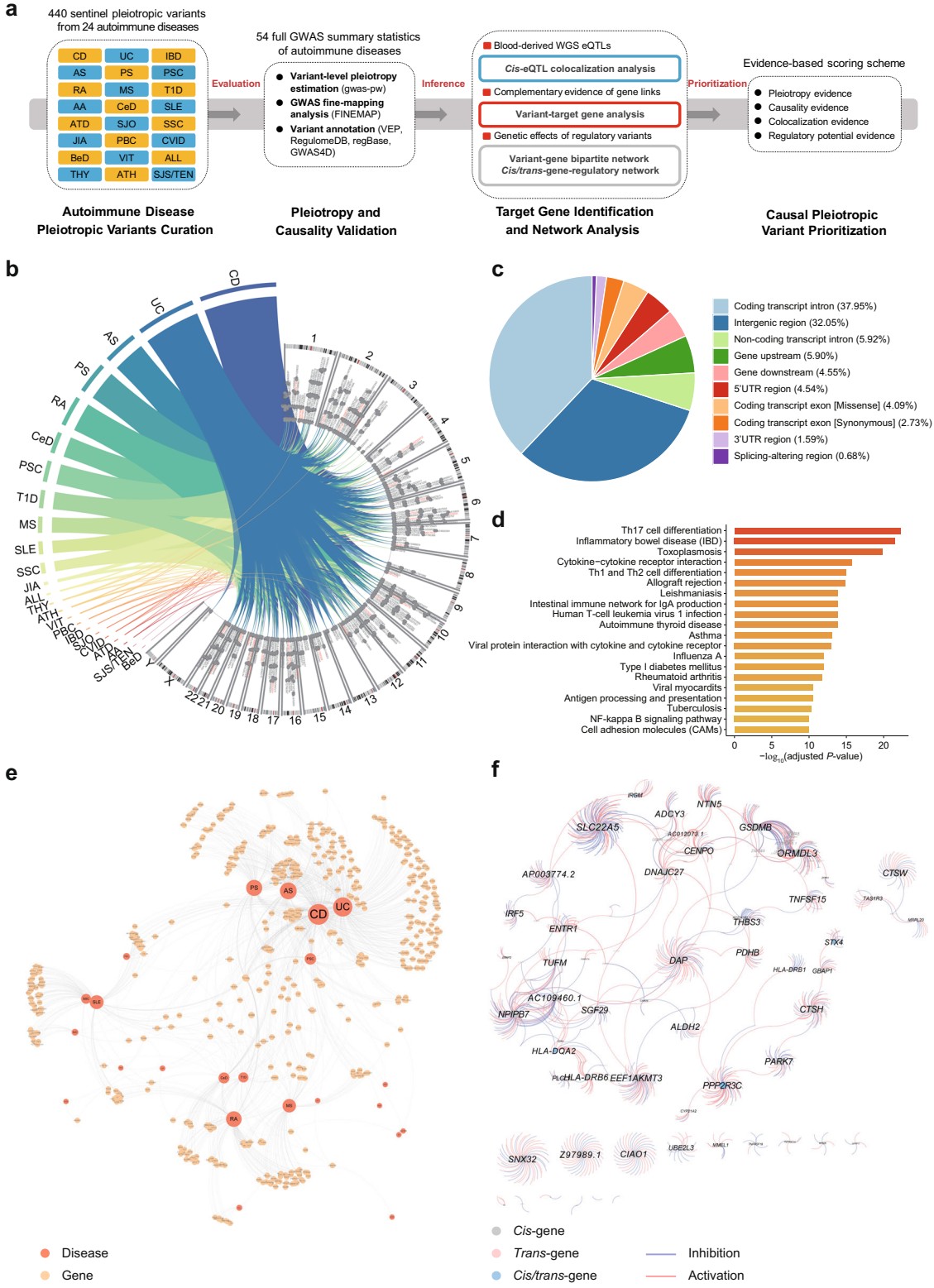

## Evidence-based prioritization identifies rs4728142 as a causal regulatory variant with hematopoietic lineage-specific effects

While the majority of reported pleiotropic variants have been shown to be functional by varying degrees (Supplementary Data 7), we implemented an evidence-based scoring scheme to rank them considering the reproducibility of pleiotropy, autoimmune disease causality, eQTL colocalization, and variant regulatory potential. We found 345 candidate causal regulatory SNVs with different levels of evidence and widespread genetic sharing among autoimmune diseases

(Supplementary Data 10). The top pleiotropic variant, rs4728142, was associated with six autoimmune diseases (AS, PS, RA, SLE, Sjogren's syndrome [SJO], and UC) at initial curation, and variant-level pairwise pleiotropy analysis estimated that 19 autoimmune diseases shared its genetic effects. GWAS fine-mapping, blood eQTL colocalization, and regulatory variant functional prediction also supported rs4728142 as a causal regulatory variant (Supplementary Data 10). rs4728142 (GRCh37/hg19 chr7: 128,573,967) is located 3−5 kb upstream of the *IRF5* gene transcription start sites (TSSs), and IRF5 functions as an

**Fig. 1 | Functional annotation and target gene analysis of autoimmune disease-associated pleiotropic variants. a** Computational workflow for systematically functional annotation and prioritization of autoimmune disease-associated pleiotropic variants. GWAS genome-wide association studies, WGS whole-genome sequencing, eQTLs expression Quantitative Trait Loci. **b** Systematically curated genome-wide studies for identification of the potential pleiotropic or shared genetic loci associated with at least two autoimmune diseases. 440 sentinel pleiotropic variants for 24 autoimmune diseases from 12 genome-wide pleiotropy studies were collected and analyzed. Numbers and letters mark chromosome order numbers. Abbreviations for autoimmune diseases: CD Crohn's disease, UC ulcerative colitis, AS ankylosing spondylitis, PS psoriasis, RA rheumatoid arthritis, CeD celiac disease, PSC primary sclerosing cholangitis, T1D type 1 diabetes, MS multiple sclerosis, SLE systemic lupus erythematosus, SSC systemic sclerosis, JIA juvenile idiopathic arthritis, ALL any allergies, THY thyroiditis, ATH asthma, VIT vitiligo, PBC

primary biliary cirrhosis, IBD inflammatory bowel disease, SJO Sjogren's syndrome, CVID common variable immunodeficiency, ATD autoimmune thyroid disease, AA alopecia areata, SJS/TEN Stevens–Johnson syndrome/toxic epidermal necrolysis, BeD Behcet's disease. **c** Functional annotation and variant consequence distribution of the curated pleiotropic variants. UTR untranslated region. **d** Kyoto Encyclopedia of Genes and Genomes (KEGG) pathway enrichment analysis for all identified target genes affected by the pleiotropic variants, the overrepresentation test is used to calculate adjusted *P* value. **e** Genetic relationships among the investigated autoimmune diseases by constructing a bipartite network between the pleiotropic variants and their target genes. **f** Gene regulatory network for causal relationships of the autoimmune disease-associated genes using the Genotype-Tissue Expression (GTEx) whole blood *cis/trans*-eQTLs summary statistical information.

inflammatory TF that plays an important role in the type I interferon (IFN) pathway and mediates the induction of proinflammatory cytokines such as tumor necrosis factor-alpha (TNF-α), IL-6, IL-12, and IL-23[35,36], and contributes to the pathogenesis of many autoimmune diseases[23]. Primary studies have shown that rs4728142 is a likely causal variant of different autoimmune diseases, such as UC and SLE[26,37], and it could affect TF binding in its DNA functional element in vitro[25,26]. However, the causal regulatory mechanism and how such genetic effect transmits to the development of common autoimmune disorders remain elusive.

In addition, the immune-related risk alleles have complex and lineage-specific effects, but most of the genetic effects of the autoimmune risk loci have been evaluated and tested using human B cells. Remarkably, prior studies have reported that the *IRF5* risk haplotypes in autoimmune diseases lack B cell-intrinsic effects and alternatively could exert prominent effects in human monocyte-derived cells[20,38,39]. To estimate the most likely causal cell type underlying the genetic effect of rs4728142, we used g-chromVAR, a high-resolution cell-type enrichment method[40], to distinguish disease-causal cell types by computing the open chromatin enrichments of autoimmune disease fine-mapped variants across 18 hematopoietic cell types. We found that autoimmune diseases (e.g., UC and SLE) associated with rs4728142 were enriched in granulocyte-macrophage progenitors, implying that the genetic effect of these autoimmune diseases may act in monocyte-macrophage lineage cells (Supplementary Fig. 2a). Moreover, the genomic region surrounding the pleiotropic variant rs4728142 shows low chromatin accessibility in monocytes, whereas the region's chromatin is not accessible in other hematopoietic cells (Supplementary Fig. 2b), and the potential target gene *IRF5* also showed distinct expression in monocytes according to the DICE dataset (2019 release)[41] and ImmuNexUT dataset[42] (Supplementary Fig. 2c, d). By querying on QTLbase[16], we noticed that rs4728142 obtains the largest effect size being eQTLs in monocytes than other tissue/cell types (Supplementary Fig. 2e). These evidence highlight the monocyte/macrophage-relevant role of rs4728142 in the development of autoimmune comorbidities.

### rs4728142 affects the activity of a promoter-like enhancer adjacent to the *IRF5* promoter

Given the prominent predicted regulatory potential and unknown biological function of rs4728142, we investigated the epigenomic landscape surrounding this pleiotropic variant. Examination of the consolidated DNase I hypersensitive site sequencing (DNase-seq) and histone modification chromatin immunoprecipitation followed by sequencing (ChIP-seq) data on the CD14⁺ monocytes from the Roadmap Epigenomics Project[43] showed that rs4728142 harbored both enhancer (H3K27ac and H3K4me1) and promoter (H3K27ac and H3K4me3) markers in a dyadic open chromatin region close to the *IRF5* promoter (Supplementary Fig. 4a). Next, we performed assay for transposase-accessible chromatin using sequencing

(ATAC-seq) and ChIP-seq/ChIP-qPCR assays of H3K27ac, H3K4me1, and H3K4me3 on SC cells, a monocyte-derived cell line established from human peripheral blood mononuclear cells with heterozygous alleles (G and A) at rs4728142, representing a good cell model for studying the genetic effect of rs4728142. Consistent with the public data, the rs4728142 surrounding region in the SC cells exhibited chromatin states associated with both promoter and enhancer activities (Fig. 2a, b). Interestingly, rs4728142 is located on the downstream boundary of the open chromatin peak among the cell types examined and colocalized with nucleosomes marked by H3K27ac, H3K4me1, and H3K4me3, suggesting this pleiotropic variant located in active chromatin could associate with nucleosome-bound proteins in modulating its regulatory activity and *IRF5* gene expression. To investigate whether the rs4728142-containing region shows a likely allele-specific effect on enhancer activity, we first applied allelic imbalance analysis using ChIP-seq of H3K4me1, H3K27ac, H3K4me3, and ATAC-seq, based on our SC cells with heterozygous genotype at rs4728142. Expectedly, we noticed that rs4728142 shows distinct allelic imbalance for both H3K4me1 and H3K27ac marks, in which risk allele A exhibits more read coverage than allele G (Supplementary Fig. 3a). Second, by querying our QTLbase database[16], we found that rs4728142 has been identified as histone modification QTLs (e.g., H3K27ac QTLs) in Monocytes[44] (Supplementary Fig. 3b), suggesting an allele-specific enhancer activity at rs4728142. Third, we also performed the allele-specific ChIP-qPCR for H3K4me1 and H3K27ac on the rs4728142-containing region in SC cells. As a result, allele A at the rs4728142-containing region has more H3K4me1 and H3K27ac enrichment than allele G (Supplementary Fig. 3c). Sequencing of the ChIP-PCR products amplified at the rs4728142 locus detected an allele-biased binding for rs4728142-A when immunoprecipitated by anti-H3K4me1 and anti-H3K27ac antibody (Supplementary Fig. 3d), indicating allele A of rs4728142 may have more enhancer activity than allele G in SC cells.

To evaluate the regulatory activity of rs4728142-associated DNA elements in vitro, we performed luciferase reporter assays by cloning short sequences (282 bp around rs4728142, GRCh37/hg19 chr7: 128,573,817–128,574,098, termed the rs4728142-containing region) containing the nonrisk allele G (rs4728142-G) or the risk allele A (rs4728142-A), into luciferase reporter vectors with or without promoter in 293T and SC cells, respectively (Fig. 2a and Supplementary Fig. 4b, c). We observed that the rs4728142-containing region reduced luciferase activity slightly as an enhancer or promoter, in which the fragments with rs4728142-A had lower luciferase expression levels compared with those with rs4728142-G (Fig. 2c, d). Meanwhile, when testing the regulatory activity of an adjacent upstream sequence (GRCh37/hg19 chr7: 128,573,081–128,573,816) spanning the H3K27ac, H3K4me1, and H3K4me3 peaks, we found that the fragment greatly increased luciferase activity as an enhancer or promoter (Fig. 2c, d), indicating the dual regulatory potentials

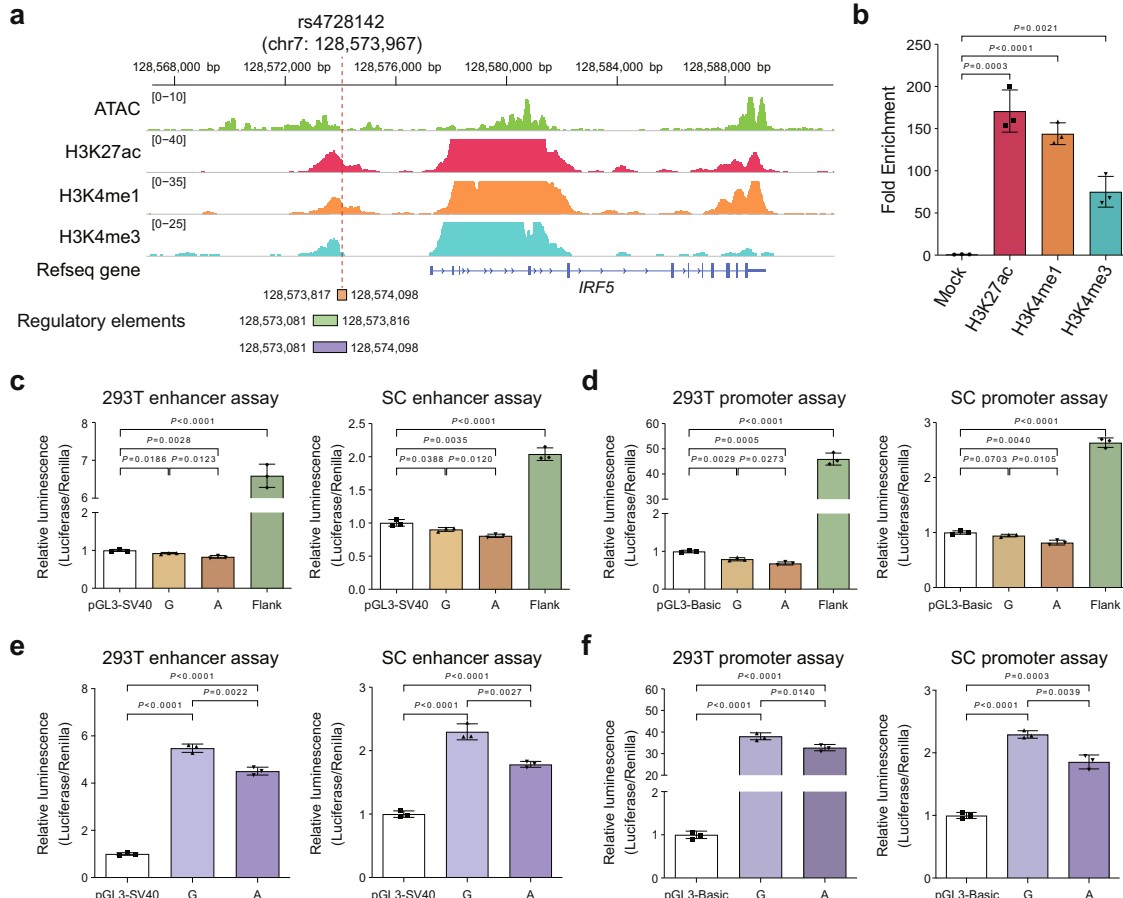

**Fig. 2 | rs4728142 affects the activity of a promoter-like enhancer adjacent to *IRF5* promoter. a** Assay of Transposase Accessible Chromatin sequencing (ATAC-seq), and H3K27ac, H3K4me1, and H3K4me3 Chromatin immunoprecipitation followed by sequencing (ChIP-seq) signals at the interferon regulatory factor 5 (*IRF5*) nearby region in SC cells, a human monocyte-derived cell line with heterozygote alleles (the nonrisk allele G and the risk allele A) at rs4728142. Red dotted line denotes the rs4728142 location. Sequence location information (GRCh37/hg19) for regulatory elements (including the rs4728142-containing sequence, the rs4728142-associated upstream enhancer sequence, and the integrated sequence) used in the luciferase assays is shown at the bottom of the graph. **b** ChIP enrichments of H3K27ac, H3K4me1, and H3K4me3 at the rs4728142-containing region in SC cells as determined by ChIP-qPCR. **c**, **d** Luciferase reporter assays utilizing vectors harboring the rs4728142-containing sequences with different alleles (G or A) or the upstream regulatory element (Termed Flank) for enhancer (**c**) or promoter (**d**) assay in 293T and SC cells. **e**, **f** Luciferase reporter assays utilizing vectors containing the integrated sequences harboring the upstream regulatory element and rs4728142-containing sequence with different alleles (G or A) for enhancer (**e**) or promoter (**f**) assay in 293T and SC cells. pGL3-Basic vector without a promoter was used for the promoter assay, and pGL3-Promoter vector with SV40 promoter was used for the enhancer assay. Luciferase signals were normalized to renilla signals. Data are represented as the means ± standard deviations (SD), *n* = 3 biologically independent samples, and unpaired two-tailed Student's *t* test is used to calculate *P* values in (**b**–**f**). Source data are provided as a Source Data file.

of this sequence. To test the effect of rs4728142 on its upstream regulatory element, we cloned the integrated sequences (GRCh37/hg19 chr7: 128,573,081–128,574,098) harboring the upstream element and rs4728142-containing region with different alleles (G or A) into luciferase reporter vectors (Fig. 2a). Interestingly, compared with rs4728142-G, rs4728142-A decreased both enhancer and promoter activities in 293T and SC cells (Fig. 2e, f and Supplementary Fig. 4b, c), implying that the regulatory activity of the upstream element depends on the rs4728142-containing sequence and that risk allele A weakens the overall regulatory activity of the integrated sequence via unknown mechanisms. Considering that enhancers function as key *cis*-regulatory elements that affect gene transcription independent of their orientation or distance[45], we further cloned the regulatory sequences downstream of the luciferase reporter gene (Supplementary Fig. 4d). As expected, similar results were observed as those upstream clones in both the 293T and SC cells (Supplementary Fig. 4e, f). Together, our data demonstrate that rs4728142-A reduces the regulatory potential of its upstream promoter-like enhancer (i.e., rs4728142-associated enhancer) in the luciferase reporter system.

## The rs4728142-containing region interacts with the *IRF5* downstream alternative promoter in an allele-specific manner

Our in silico target gene analysis linked rs4728142 to the nearest gene, *IRF5*, by the DEPICT (v1_rel194) and eQTL colocalization methods, and we next sought to elucidate the underlying mechanism. Chromatin loops regulate gene expression by positioning distal enhancers in proximity to the target genes. Therefore, one possible mechanism by which the upstream enhancer of rs4728142 affects *IRF5* expression is direct loop formation/disruption between rs4728142 and the *IRF5* promoter. To test this hypothesis, we used the circularized chromosome conformation capture (4C) assay[46] to examine the interacting loci of the rs4728142-containing region using the rs4728142-containing region as bait on SC cells. Remarkably, a short-range interaction with the highest interaction frequency was detected between the rs4728142-containing region and the downstream region of *IRF5* TSS1 (Fig. 3a). Based on previous findings, *IRF5* has multiple TSSs dedicated to distinct alternative promoters and *IRF5* isoforms (Supplementary Fig. 5a) that differ in their ability to transactivate the *IFN* genes[47] and modulate autoimmune disease risk[24,27,48]. We found clear evidence that the 4C fragment interacting with the

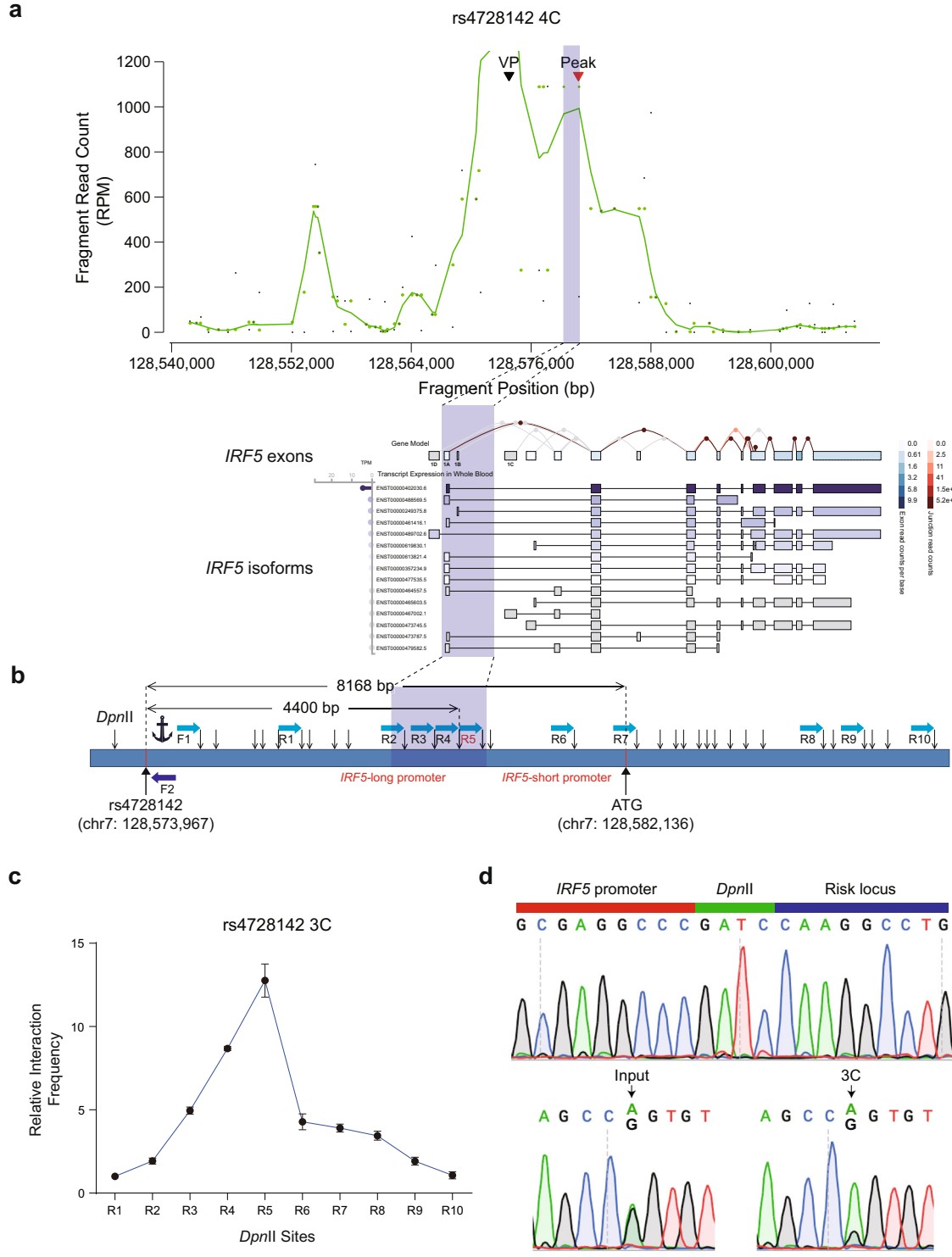

**Fig. 3 | The rs4728142-containing region interacts with the *IRF5* downstream alternative promoter in an allele-specific manner. a** Circularized Chromosome Conformation Capture (4C) assay for the rs4728142-containing region. RPM reads per million. VP (viewpoint) denotes the *Dpn*II cutting fragment harboring rs4728142. Peak denotes the fragment with the highest interaction frequency. General location information (in the shadow) of the fragments with high interaction frequencies corresponding to the *IRF5* downstream alternative promoter is shown in the bottom graph. *IRF5* transcript diagram was extracted from the GTEx portal based on the GENCODE annotation. **b** A schematic view of Chromosome Conformation Capture (3C) assay at the rs4728142-containing region. The *Dpn*II cutting fragment containing rs4728142 was measured as the bait indicated by an anchor,

and a series of fragments downstream of rs4728142 were measured as preys. Blue rectangle denotes the chromosome, black arrows denote the *Dpn*II cutting sites, light blue arrows denote the location and direction of primers, dark blue arrow "F2" denotes the reverse primer used in allele-specific detection, and "ATG" represents the common start codon in the second exon. **c** Enrichment quantification of 3C assay at the rs4728142-containing region normalized to control in SC cells. Data are represented as the means ± SD, and *n* = 3 biologically independent samples. Source data are provided as a Source Data file. **d** Sanger sequencing chromatograms of 3C ligation fragment (top) and rs4728142 locus from Input (bottom left) and 3C (bottom right) samples. Black arrows denote the rs4728142 location.

rs4728142-containing region was located between exon 1B and exon 1C of *IRF5*, which was close to a downstream alternative promoter of *IRF5*.

To confirm the striking short-range interaction, we performed the chromosome conformation capture (3C) assay. Consistent with the 4C result, the 3C data demonstrated a distinct physical interaction between the rs4728142-containing region and the *IRF5* downstream alternative promoter region (Fig. 3b, c and Supplementary Fig. 5b, c). Besides, compared with the surrounding 1-kb signals, there was a higher interaction frequency between the 1-kb bin with rs4728142 and the 1-kb bin close to the alternative promoter of *IRF5*, as shown using Hi-C data in GM12878 cells (Supplementary Fig. 5d). Given the heterozygous genotype of rs4728142 on the SC cell line, we designed specific 3C primers to amplify the ligation DNA sequences incorporating both the rs4728142 viewpoint and the fragment with the highest interaction frequency on *IRF5*. Notably, rs4728142-A had a preference in the 3C ligation products (Fig. 3d), suggesting that this interaction is allele-specific and highly reproducible.

### The rs4728142-containing region orchestrates chromatin looping to regulate *IRF5* alternative promoter usage

The chromatin interaction region with the highest looping intensity separated the *IRF5* alternative promoters into upstream (TSSs for exon 1D, 1A, and 1B) and downstream promoters (TSSs for exon 1C) (Supplementary Fig. 6a). The remarkable allele-specific loop effect of rs4728142 on the *IRF5* downstream alternative promoter spurred our investigation of whether the variant modulates the expression pattern of the long and short *IRF5* transcripts. To this end, we classified *IRF5* transcripts into longer (*IRF5*-long) and shorter (*IRF5*-short) transcripts according to the upstream and downstream alternative promoters, separated by the loop region. We hypothesized that the rs4728142-containing region orchestrates its upstream enhancer to locate specific target regions by loop formation, resulting in the preference change in *IRF5* promoter usage.

Based on our ChIP-seq/ATAC-seq profiles of the SC cells and the distribution of the restriction sites, a high-resolution 4C assay was performed using the only enhancer region (GRCh37/hg19 chr7: 128,573,036–128,573,717) adjacent to the downstream SNP as the viewpoint. Indeed, high interaction frequencies between the rs4728142-associated enhancer and the *IRF5*-long/short promoter regions were detected (Fig. 4a). Consistent with the 4C results, 3C assay also revealed high contact probabilities at either the *IRF5*-long or -short promoters (Fig. 4b, c and Supplementary Fig. 6b), implying that the rs4728142-associated enhancer regulates multiple *IRF5* alternative promoters at the heterozygous status of rs4728142. Compared with the interaction pattern between the rs4728142-containing region without the upstream enhancer sequence and *IRF5* alternative promoters, the interaction fragments obtaining relatively high looping intensity were dispersed and downshifted when using the rs4728142-associated enhancer as the viewpoint (Figs. 3a and 4a). We also confirmed via luciferase reporter assays that the loop-associated sequence (GRCh37/hg19 chr7: 128,579,302–128,580,416) in the *IRF5*-short promoter exhibited high promoter activity (Supplementary Fig. 6c–e), and acquired the similar phenotypes of enhancer activity with the aforementioned SV40 promoter in rs4728142-related luciferase assays (Supplementary Figs. 7 and 8). To validate the function of the rs4728142-associated enhancer in regulating *IRF5* transcription, we designed CRISPR activation (CRISPRa) and CRISPR interference (CRISPRi) assays to manipulate the regulatory activity of the rs4728142-associated enhancer in SC cells, and used the specific primers[27] to detect the expression changes of *IRF5*-short (exon 1C) and *IRF5*-long (exon 1D) transcripts. When the enhancer was fully activated by CRISPRa, both the *IRF5*-short (exon 1C) and *IRF5*-long (exon 1D) transcripts were markedly upregulated (Fig. 4d); on the contrary, the transcript expression levels were significantly downregulated when the enhancer was inhibited by CRISPRi (Fig. 4e).

To evaluate whether the rs4728142-containing region affects the function of its upstream enhancer and eventually modulates the expression levels of the *IRF5*-short and *IRF5*-long transcripts, we constructed a single cell-derived SC-Cas9 cell line and acquired single cell-derived double-knockout (dKO) and single-knockout (sKO) cells for the rs4728142-containing region using CRISPR technology. Specifically, the dKO SC cells were obtained from a double small guide RNA (sgRNA) deletion experiment by specifically targeting a ~120-bp surrounding sequence of the SNP, while the sKO SC cells were obtained from a single sgRNA knockout experiment by specifically targeting the SNP site (Fig. 4f and Supplementary Fig. 6f). As expected, both dKO and sKO consistently decreased the expression of the *IRF5*-short transcripts and increased the expression of the *IRF5*-long transcripts, but had no effect on a nearby gene, *TNPO3* (Fig. 4g and Supplementary Fig. 6g), suggesting that the rs4728142-containing region is involved in managing *IRF5* promoter usage. On the other hand, our gene-based eQTL and colocalization results showed that rs4728142-A was significantly associated with higher *IRF5* gene expression and obtained a shared genetic effect with autoimmune disease risk. Analysis of transcript-level eQTLs based on WGS and RNA-seq of 194 BLUEPRINT CD14[+] monocyte donors showed that rs4728142 was identified with opposite effects on the *IRF5*-short and *IRF5*-long transcripts, in which rs4728142-A was markedly associated with higher *IRF5*-short transcript expression and lower *IRF5*-long transcript expression (Fig. 4h). A recent transcript usage QTL study on macrophages exposed to inflammatory and metabolic stimulus also reported that rs4728142 was markedly associated with *IRF5* promoter usage[15] (Supplementary Data 11). Together with the 3C-derived assays, these results demonstrate that rs4728142-A contributes to the loop formation between its upstream enhancer and *IRF5*-short promoter, thereby regulating *IRF5* promoter usage and transcript expression.

### A putative structural regulator, ZBTB3, mediates the allele-specific chromatin looping to regulate *IRF5*-short transcript expression at the rs4728142 locus

Based on the results of the luciferase reporter assays, 3C-derived assays, and eQTL analysis, rs4728142 displays apparently incompatible effects on luciferase system, chromatin looping, and gene expression. That is, rs4728142-A represses both enhancer and promoter activities in linear DNA form, but plays a positive role in facilitating short-range looping and increasing the overall gene expression of *IRF5*. To decipher which mechanism underlies such a puzzle, we screened TF binding motifs over different alleles of rs4728142 and flanking sequences by CIS-BP (Build 2.00)[49] and FIMO (v5.4.1)[50]. Consistent with previous findings[26], we identified a promising TF, ZBTB3, which received a strong motif enrichment score for rs4728142-A instead of the nonrisk rs4728142-G (Fig. 5a and Supplementary Data 12). We then used electrophoretic mobility shift assay (EMSA) to investigate the differences in protein binding between rs4728142-G and rs4728142-A using SC cell nuclear extracts and anti-ZBTB3 antibody. The results revealed that ZBTB3 preferentially bound to rs4728142-A (Fig. 5b), indicating that ZBTB3 might be a key binding factor involved in the function execution of the rs4728142-associated enhancer. ZBTB3 is a poorly characterized protein that contains a DNA-binding zinc finger and a transcription-repressing BTB/POZ domain. One of the important paralogs of this gene, *ZBTB7B*, acts as a repressor of CD8 genes in T cells[51], implying that ZBTB3 may have a similar function in the gene regulation of immune cells.

Although the putatively repressive role of ZBTB3 supported the phenotype of rs4728142-A in the luciferase reporter assay, whether it mediates the allele-specific loop formation in the SNP locus warrants further investigation. Previous studies had determined that several zinc finger proteins, such as CTCF and YY1, promote enhancer–promoter interaction along with mediators, cohesin, and other transcriptional cofactors[52–55]. A chromatin immunoprecipitation plus mass

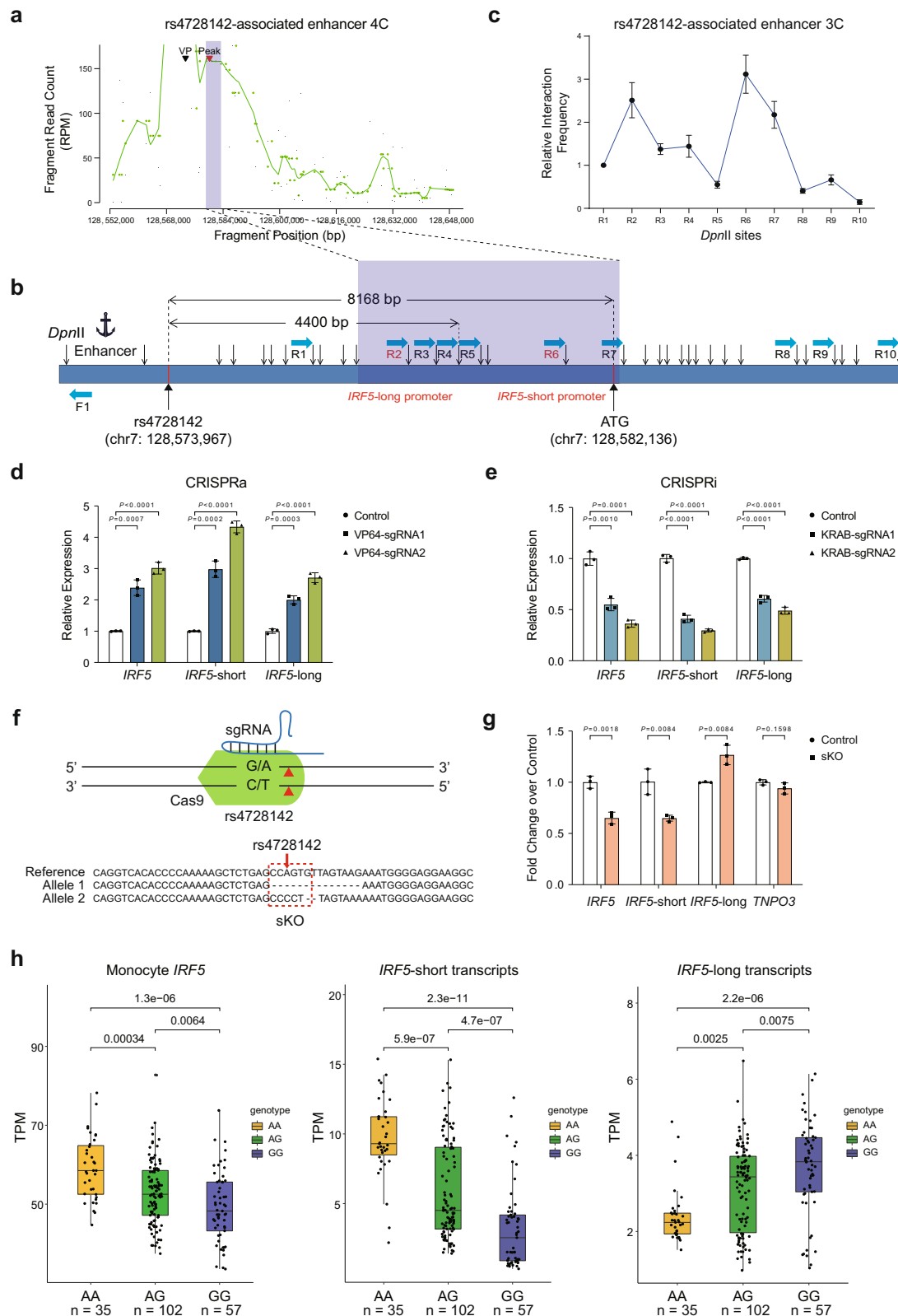

**a** rs4728142-associated enhancer 4C

**c** rs4728142-associated enhancer 3C

**b**

**d** CRISPRa

**e** CRISPRi

**f**

**g**

**h** Monocyte *IRF5* / *IRF5*-short transcripts / *IRF5*-long transcripts

spectrometry assay revealed 26 candidate enhancer–promoter structural factors, in which ZBTB11 had been documented[55,56]. Therefore, we speculated that ZBTB3 could function as a structural factor mediating the chromatin looping between the rs4728142-associated enhancer and the *IRF5*-short promoter. By TF motif scanning of the fragment sequence (GRCh37/hg19 chr7: 128,578,363–128,578,537) with the highest 4C signal at the *IRF5* downstream alternative promoter, we were fortunate to detect a putative ZBTB3 binding site (GRCh37/hg19 chr7: 128,578,496–128,578,503) in the downstream region of the fragment sequence (enrichment score = 0.498) (Fig. 5c), which was convergent to the rs4728142-associated ZBTB3 binding site. This binding potential was tested with EMSA by incubating SC cell nuclear extracts and anti-ZBTB3 antibody with the biotin probes of the candidate binding site. The candidate ZBTB3 binding sequence within the *IRF5* downstream

**Fig. 4 | The rs4728142-containing region orchestrates chromatin looping to regulate *IRF5* alternative promoter usage. a** 4C assay for the enhancer adjacent to the downstream rs4728142-containing region (rs4728142-associated enhancer). RPM reads per million. VP (viewpoint) denotes the *Dpn*II cutting fragment harboring the rs4728142-associated enhancer. **b** A schematic view of 3C assay at the rs4728142-associated enhancer. The *Dpn*II cutting fragment harboring the rs4728142-associated enhancer was measured as the bait indicated by an anchor, and the fragments downstream of the enhancer were measured as preys. Blue rectangle denotes the chromosome, black arrows denote the *Dpn*II cutting sites, light blue arrows denote the location and direction of primers, and "ATG" represents the common start codon in the second exon. General location information (in the shadow) of the fragments with relatively high interaction frequencies corresponding to *IRF5* promoters is highlighted. **c** Enrichment quantification of 3C assay at the rs4728142-associated enhancer normalized to control in SC cells. **d, e** *IRF5* transcript expression in the rs4728142-associated enhancer activation or inhibition SC cells as determined by RT-qPCR. sgRNAs targeting the rs4728142-associated enhancer were used in CRISPR activation (CRISPRa) (**d**) and CRISPR interference (CRISPRi) (**e**) assays. **f** Strategy of rs4728142 single-knockout (sKO, top) and Sanger sequencing result as analyzed by CRISP-ID Web Portal (bottom). Red triangle denotes the cutting site, and red dotted box denotes the ZBTB3 core motif sequence. **g** *IRF5* transcript expression in the rs4728142-sKO SC-Cas9 cells compared with the unedited SC-Cas9 cells (control) as determined by RT-qPCR. *TNPO3* is the gene adjacent to the upstream *IRF5* gene. **h** eQTL analysis for correlation between the rs4728142 genotype and *IRF5* transcript expression using 194 monocyte RNA-seq samples from the BLUEPRINT dataset. Box and whisker plot; boxes depict the upper and lower quartiles of the data, and whiskers depict the range of the data. Data are represented as the means ± SD, *n* = 3 biologically independent samples, and unpaired two-tailed Student's *t* test is used to calculate *P* values in (**c**, **d**, **e**, **g**). Source data are provided as a Source Data file.

alternative promoter bound ZBTB3 in vitro, and when we deleted two bases (ΔCT) in the ZBTB3 core motif, the binding activity was lost (Fig. 5d). To test whether ZBTB3 binds both rs4728142 and the *IRF5* downstream alternative promoter directly, the corresponding biotin probes were incubated with ZBTB3 recombinant protein instead of with nuclear extracts. As expected, both rs4728142-A and the candidate ZBTB3 binding sequence in the *IRF5* downstream alternative promoter showed direct binding to the ZBTB3 recombinant protein (Fig. 5e). To validate the structural role of ZBTB3 in loop formation, we performed ChIP-seq/ChIP-qPCR assays for CTCF, ZBTB3, and RAD21 on SC cells, and found that the genome-wide ZBTB3 binding sites generally colocalized with CTCF/RAD21 binding sites (Supplementary Fig. 9a, b). Interestingly, there were signal enrichments and peaks at both the rs4728142 locus and the *IRF5*-short promoter region for CTCF, ZBTB3, and cohesin (Fig. 5f and Supplementary Fig. 9c), suggesting that the ZBTB3-mediated loop at the rs4728142 locus may relate with CTCF or cohesin. In addition, we also found a RAD21 chromatin interaction analysis by paired-end tag sequencing (ChIA-PET) loop that connected the rs4728142-containing region and *IRF5*-short promoter region in the GM12878 lymphoblastoid cell line (Supplementary Data 13), which concurs with our experimental results. In line with the EMSA results, sequencing of the ChIP-qPCR products amplified at the rs4728142 locus detected increased allele-specific binding for rs4728142-A when immunoprecipitated by anti-CTCF, ZBTB3, or RAD21 antibody (Fig. 5g). Together, these observations support the model that ZBTB3 acts a structural regulator mediating the allele-specific loop formation at the rs4728142 locus in monocytes.

To investigate the functional relationship between the expression of ZBTB3 and the *IRF5*-short isoforms in monocytes, we performed a correlation analysis of 194 BLUEPRINT CD14⁺ monocyte samples using genotype and transcript expression. Globally, *ZBTB3* and *IRF5* gene expression was correlated positively. Interestingly, we also found a positive correlation between *CTCF* and *IRF5* expression (Fig. 5h), further suggesting that ZBTB3 and CTCF may cooperatively regulate *IRF5* expression. However, we did not observe a positive correlation between *ZBTB3* and *IRF5* on 462 Geuvadis lymphoblastoid B cells (Supplementary Fig. 9d). As the rs4728142 genotype changed from AA to AG to GG, the correlations between *ZBTB3* and the highest expressed *IRF5*-short transcript (ENST00000473745) gradually decreased in the monocytes; when this locus was homozygous AA, we observed a significant positive correlation (Supplementary Fig. 9e). Consistent with the genetic effect on the aggregated *IRF5*-short and *IRF5*-long transcripts (Fig. 4h), rs4728142 impacted the expression of the highest expressed *IRF5*-short transcript (ENST00000473745) and *IRF5*-long transcript (ENST00000249375) in reciprocally and coordinately (Supplementary Fig. 9f, g), further supporting the allele-specific effect of rs4728142 in regulating *IRF5* transcript expression. Moreover, ZBTB3 overexpression in SC cells increased both *IRF5* gene and *IRF5*-short (exon 1C) transcript expression levels, but decreased *IRF5*-long

(exon 1D) transcript expression (Fig. 5i). On the contrary, a nearly reciprocal result was detected after ZBTB3 knockout by CRISPR (Fig. 5j). The ZBTB3 binding site disruption experiment in the rs4728142-sKO SC cells also yielded similar phenotypes with the ZBTB3-overexpression and knockout cells (Fig. 4g).

To further validate the function of rs4728142 in the loop formation and loop switching, we performed rs4728142 and rs4728142-associated enhancer 3C assays on SC-sKO cells. As expected, compared with SC cells, the rs4728142-containing region in SC-sKO cells had a weaker interaction with R5 (between *IRF5*-long and -short promoters, Fig. 3b) (Supplementary Fig. 10a), indicating deletion of rs4728142-containing region can disrupt the loop formation between rs4728142-containing region and the *IRF5* downstream alternative promoter; and the rs4728142-associated enhancer had a stronger interaction with R2 (*IRF5*-long promoter core region, Fig. 4b) and a weaker interaction with R6 (*IRF5*-short promoter core region, Fig. 4b) (Supplementary Fig. 10b), indicating deletion of rs4728142-containing region can promote the loop switching from the promoter of *IRF5*-short isoforms to the promoter of *IRF5*-long isoforms. Next, to further test whether ZBTB3 is directing the chromatin looping, we performed 3C assays on the ZBTB3-knockout (KO) and ZBTB3-overexpression (OE) SC cells. As a result, ZBTB3-KO significantly weakened the interaction between rs4728142 and R5 (between *IRF5*-long and -short promoters, Fig. 3b) (Supplementary Fig. 10c), and switched the rs4728142-associated enhancer from R6 (*IRF5*-short promoter core region, Fig. 4b) to R2 (*IRF5*-long promoter core region, Fig. 4b) (Supplementary Fig. 10d). On the contrary, ZBTB3-OE significantly strengthened the interaction between rs4728142 and R5 (between *IRF5*-long and -short promoters, Fig. 3b) (Supplementary Fig. 10e), and switched the rs4728142-associated enhancer from R2 (*IRF5*-long promoter core region, Fig. 4b) to R6 (*IRF5*-short promoter core region, Fig. 4b) (Supplementary Fig. 10f). Taken together, these results demonstrate that rs4728142 modulates *IRF5* alternative promoter usage in monocytes through ZBTB3-mediated chromatin looping.

## rs4728142 modulates the autoimmune disease-related pathways and macrophage M1 polarization

To investigate the molecular processes and cellular phenotypes underlying the causal pleiotropic variant, we applied RNA-seq to the unedited SC-Cas9 (control) and rs4728142-sKO SC cells. Of the 700 differentially expressed genes in the sKO cells versus the control cells (adjusted *P* value <0.05, |log₂fold change| >1), 311 genes were upregulated and 389 were downregulated (Fig. 6a). Pathway enrichment analysis of the sKO cells showed that the differentially regulated genes were related to hematopoietic cell lineage, antigen processing and presentation, and multiple autoimmune disease signaling pathways in the Kyoto Encyclopedia of Genes and Genomes (KEGG) database (Fig. 6b). Remarkably, gene set enrichment analysis (GSEA) revealed significant enrichments for several

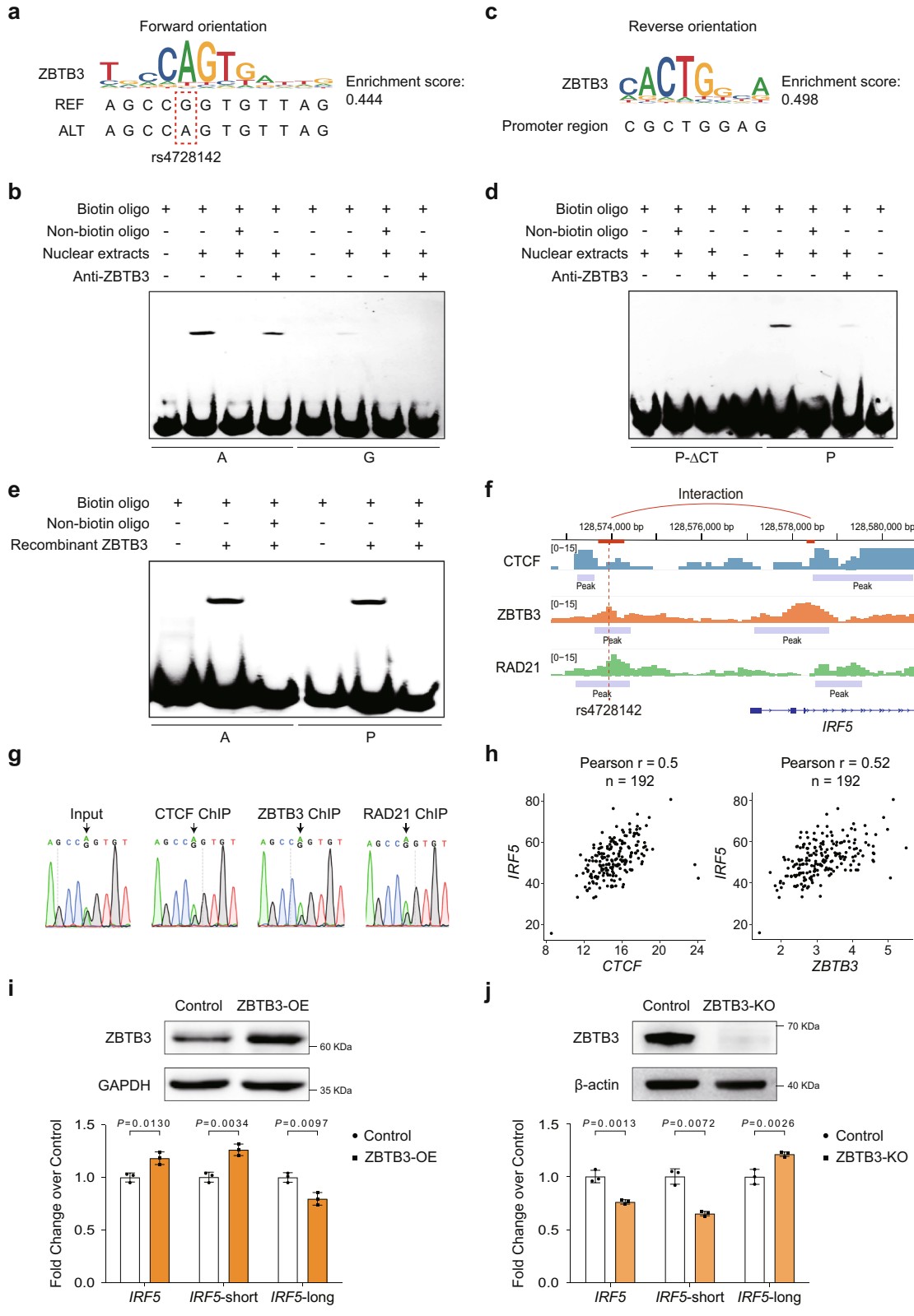

autoimmune diseases that were also linked to the genetic pleiotropy of rs4728142, such as inflammatory bowel disease (IBD), SLE, RA, and the intestinal immune network for IgA production (Fig. 6c). In addition, analysis of RNA-seq data from autoimmune disease patients[57,58] suggests that *IRF5* gene and *IRF5*-short transcripts show higher expression in RA or SLE patients than healthy individuals (Supplementary Fig. 9h).

Given that IRF5 regulates proinflammatory M1 macrophage polarization, which contributes to the elevation of the inflammatory and immune response[20,22,59], we examined whether rs4728142 modulates M1 macrophage polarization through its genetic effect on the *IRF5* gene. As SC cells are mostly undifferentiated monocytes, we used phorbol 12-myristate 13-acetate (PMA) and lipopolysaccharide (LPS) to induce M1 macrophages[60,61]. In the normal culture state, the SC cells

**Fig. 5 | ZBTB3 mediates the allele-specific chromatin looping to regulate *IRF5*-short transcript expression at the rs4728142 locus. a** ZBTB3 motif binding sequence at the rs4728142 locus with a forward orientation as analyzed by CIS-BP and FIMO. REF reference allele, ALT alternative allele. **b** EMSA assay for the sequences harboring rs4728142 with different alleles using SC nuclear extracts and anti-ZBTB3 antibody. **c** ZBTB3 motif binding sequence in the *IRF5* downstream alternative promoter region possessing the highest interaction frequency with the rs4728142-containing region, has the reverse (convergent) orientation compared with the ZBTB3 motif at the rs4728142 locus. **d** EMSA assay for the sequences possessing the highest interaction frequency with the rs4728142-containing region using SC nuclear extracts and anti-ZBTB3 antibody. P-ΔCT denotes the sequence deleting CT in the ZBTB3 core motif. **e** EMSA assay for the sequence harboring rs4728142-A and the sequence possessing the highest interaction frequency with the rs4728142-containing region using ZBTB3 recombinant protein. **f** CTCF, ZBTB3, and RAD21 ChIP-seq signals at the *IRF5* nearby region in SC cells. Highlighted boxes denote the rs4728142-containing 3C/4C viewpoint and its highest interaction frequency region. **g** Sanger sequencing chromatograms of CTCF, ZBTB3, and RAD21 ChIP-qPCR at the rs4728142 locus. **h** Correlation analysis of *CTCF* and *ZBTB3* with *IRF5* expression using RNA-seq data of 194 human CD14[+] monocyte samples from the BLUEPRINT dataset. **i** Western blotting for validating the ZBTB3 overexpression (ZBTB3-OE) and RT-qPCR for detecting the *IRF5* transcript expression in ZBTB3-OE SC cells compared with the SC cells transfected with empty vector (control). **j** Western blotting for validating the ZBTB3 knockout (ZBTB3-KO) and RT-qPCR for detecting the *IRF5* transcript expression in ZBTB3-KO SC-Cas9 cells compared with the unedited SC-Cas9 cells (control). Each experiment was repeated three times independently with similar results. Data are represented as the means ± SD, $n = 3$ biologically independent samples, and unpaired two-tailed Student's *t* test is used to calculate *P* values in (**i, j**). Source data are provided as a Source Data file.

were the same size, round or oval suspension cells without agglomeration (Fig. 6d, left). After PMA induction, the cells that were mainly round or oval grew close to the wall, and agglomerations appeared (Fig. 6d, middle). Under LPS stimulation, the SC macrophages adhered to the wall to grow, and there were numerous long fusiform or spindle-like changes in the cells, with obvious pseudopodia (Fig. 6d, right), indicating that the M1 macrophages had been successfully induced. Following the same procedures, we induced the dKO and sKO SC cells into the M1 macrophages, and detected the macrophage M1 polarization markers (*ATF3*, *CCR7*, *COX2*, *INDO*, and *SLC7A5*) by RT-qPCR[20]. After induction, the morphology of all cell lines was not significantly different from that of the control cells. As expected, the M1 polarization markers were all markedly decreased in the dKO (Fig. 6e) and sKO (Fig. 6f) induced cells compared with the unedited SC-Cas9 induced cells (control). Collectively, these data indicate that the rs4728142 locus modulates macrophage M1 polarization and therefore may affect autoimmune disease occurrence and development.

Next, we acquired the rs4728142 with homozygous GG (rs4728142-GG) and homozygous AA (rs4728142-AA) SC cells by Cas9-initiated homology-directed repair (HDR) (Fig. 7a and Supplementary Fig. 11). We used RT-qPCR to detect the expression level of *IRF5* gene and two major forms of *IRF5* transcripts. Expectedly, compared with the rs4728142-GG SC cells, the rs4728142-AA SC cells increased the expression of *IRF5* gene and *IRF5*-short transcripts, and decreased the expression of *IRF5*-long transcripts, but had no effect on the nearest gene, *TNPO3* (Fig. 7b). Next, we induced the rs4728142-GG and rs4728142-AA SC cells to M1 macrophages, and detected the macrophage M1 polarization markers (*ATF3*, *CCR7*, *COX2*, *INDO*, and *SLC7A5*) by RT-qPCR. Compared with the rs4728142-GG-induced SC cells, all M1 polarization markers in the rs4728142-AA-induced SC cells were significantly increased (Fig. 7c), indicating that rs4728142 could modulate the macrophage M1 polarization. In addition, we analyzed the expression of key M1 proinflammatory cytokines[62,63]. As a result, compared with the rs4728142-GG-induced SC cells, the expression of *IL-1β*, *IL-6*, *IL-8*, and *TNFα* in the rs4728142-AA-induced SC cells were significantly increased (Fig. 7d). Together, the above data suggest that the allele A of rs4728142 could promote proinflammatory cytokines expression through up-regulating the expression of *IRF5*-short transcripts and *IRF5* gene, and down-regulating the expression of the *IRF5*-long transcripts, which then contributes to the M1 macrophage polarization.

In summary, based on our results, we propose a model in which an autoimmune disease pleiotropic variant, i.e., rs4728142, directs a short-range chromatin loop between its upstream enhancer and the *IRF5*-short promoter by affecting the chromatin binding of a putative structural regulator, ZBTB3, which causes aberrant *IRF5* activation and autoimmune disease risk (Fig. 7e). In the nonrisk allele status of rs4728142-G, the upstream promoter-like enhancer could freely regulate different *IRF5* alternative promoters or other genes without insulation. However, rs4728142-A steadily establishes specific regulation of the rs4728142-associated enhancer on the *IRF5*-short promoter through ZBTB3-mediated chromatin looping. This elaborate regulation of *IRF5* promoter usage leads to increases in both *IRF5*-short transcripts and ensemble gene expression of *IRF5*, ultimately resulting in monocyte/macrophage dysfunction and high autoimmune disease risk.

## Discussion

Widespread pleiotropic effects on autoimmune disease have been observed across the human genome, yet the molecular mechanisms and biological effects underlying the causal pleiotropic variants are largely unknown, particularly for the variants in the noncoding genome. In the present study, we performed systematic annotation, GWAS fine-mapping, eQTL colocalization, and target gene analysis of 440 known autoimmune disease pleiotropic variants, and ranked rs4728142 as the top causal regulatory variant whose genetic effects are shared by multiple autoimmune diseases. Given the unknown function of rs4728142, we investigated its molecular mechanism and found that this variant displays unclassical regulatory potential on *IRF5* gene expression. Our experimental results show that the risk allele of rs4728142 switches *IRF5* alternative promoter usage by affecting ZBTB3, a likely structural regulator, mediating chromatin looping and ultimately leading to the aberrant transcriptional regulation of *IRF5*-short transcripts and monocyte/macrophage dysfunction.

Although some studies have reported genome-wide pleiotropic loci on autoimmune diseases by cross-phenotype meta-analysis or variant-level genetic sharing models, no work has integrated known pleiotropic variants and comprehensively analyzed their genetic causality and biological function. Focusing on noncoding regulatory variants, we systematically investigated the regulatory potential and target genes of known pleiotropic variants, and were able to prioritize the causal regulatory variants based on different types of evidence. Both the causal variant prioritization and the target gene identification strategies can be expanded to other GWAS results when all necessary data are available, and some of these functions had been integrated to our previously developed tool VarNote-REG[42]. Our findings provide a valuable resource for studying the function of noncoding pleiotropic variants causally associated with autoimmune diseases. However, there remain unsolved problems. First, due to the limited availability of public GWAS summary statistics, most of our curated pleiotropic signals lacked information on pleiotropy type (horizontal or vertical) and direction (consistent or heterogeneous). gwas-pw (model 3) variant-level pleiotropy estimation of 54 public autoimmune disease GWAS summary data showed that many pleiotropic signals showed opposite effects of the risk allele for different autoimmune diseases (Supplementary Data 4), implying unique disease-causal mechanisms at some shared loci. Second, our fine-mapping analysis confirmed that 56.59% of the curated pleiotropic variants had plausible causal effects for at least one trait, suggesting that extra effects are needed to ascertain the true causal variants among previously reported pleiotropic loci. Third,

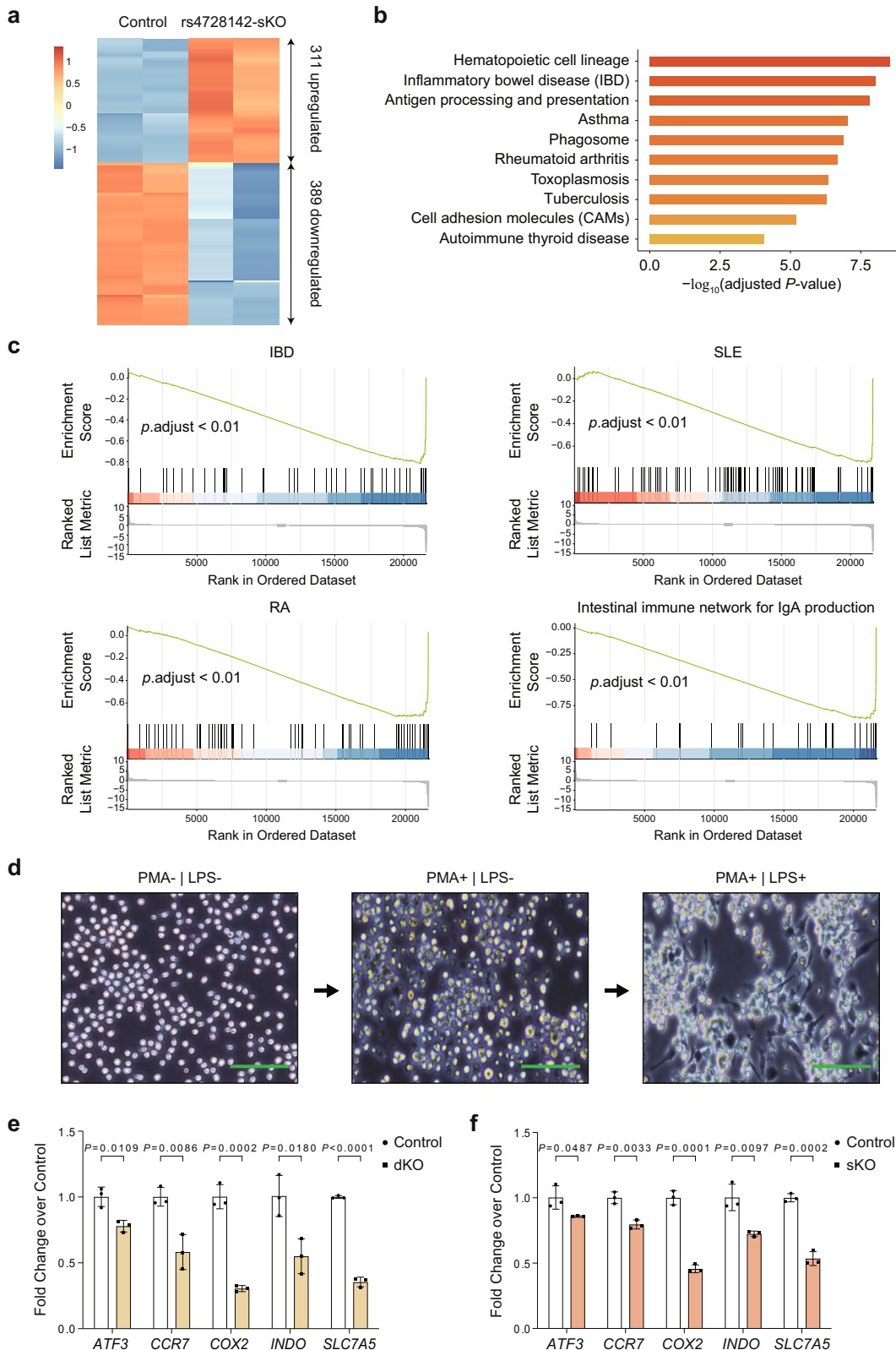

although we functionally annotated these pleiotropic variants at the coding and noncoding genomic regions, we might have omitted many variants obtaining non-eQTL effects and affecting uncharacterized molecular phenotypes[64].

Along with the advancements in genomic and epigenomic technologies, an increasing number of fine-scale molecular traits beyond gene expression have been extensively applied to investigate the genetic effect of disease alleles[16,65]. The genome-wide associations between genetic variants and promoter or transcript usage have been well documented[14,15], but deciphering of the biological mechanism underlying these associations is greatly lacking. Here, we provide a mechanistic explanation of how a causal variant orchestrates chromatin looping to regulate the gene alternative promoter selection and eventually modulate transcript expression. Compared with a recent study

**Fig. 6 | rs4728142 modulates the autoimmune disease-related signaling pathways and macrophage M1 polarization. a** Heatmap of the differentially expressed genes between the rs4728142 single-knockout (sKO) and unedited SC-Cas9 cells (control) as measured by RNA-seq (adjusted *P* value <0.05 and |Log₂Fold Change| >1). **b** Top ten enriched Kyoto Encyclopedia of Genes and Genomes (KEGG) pathways associated with the differentially expressed genes after the rs4728142 locus disruption. Pathways were ranked by −log₁₀ (adjusted *P* value). **c** Top significant gene set enrichment analysis (GSEA) with the differentially expressed genes after the rs4728142 locus disruption. Many sKO-affected genes in the autoimmune disease-related gene sets were enriched for the downregulated genes. IBD inflammatory bowel disease, SLE systemic lupus erythematosus, RA rheumatoid arthritis.

**d** Induction of monocyte-derived M1 macrophages. SC cells with high activities (left) were induced under 50 ng/μl phorbol 12-myristate 13-acetate (PMA). After inducing the cells for 2 days (middle), lipopolysaccharide (LPS, final concentration: 100 ng/μl) was added, and then the cells were cultured for 2 days (right). Scale bars, 100 μm. **e**, **f** Expression of M1 macrophage polarization markers (*ATF3, CCR7, COX2, INDO*, and *SLC7A5*) in the rs4728142 double-knockout (dKO) (**e**) or rs4728142-sKO (**f**) induced cells compared with the unedited SC-Cas9 induced cells (GA, control) as determined by RT-qPCR. Data are represented as the means ± SD, *n* = 3 biologically independent samples, and unpaired two-tailed Student's *t* test is used to calculate *P* values in (**e**, **f**). Source data are provided as a Source Data file.

demonstrating risk SNP-mediated promoter–enhancer switching[66], our work reveals that the disease-causal variant rs4728142 hijacks enhancer to interact with different alternative promoters of the *IRF5* gene. Although we found that the likely causal cell type for rs4728142 and related autoimmune diseases could be monocyte-macrophage lineage cells, whether the genetic effect has a great impact on other hematopoietic lineage-specific cells requires further investigation.

Intriguingly, the rs4728142 risk allele repressed the enhancer and promoter activities of the surrounding DNA sequence in the reporter system. Public and our ATAC-seq, DNase-seq, and histone modification ChIP-seq of the relevant cell types show that rs4728142 is located on the boundary of an open chromatin and could be associated with the nucleosome-bound proteins. Nevertheless, the rs4728142 risk allele establishes the physical interaction between the SNP region and the *IRF5* downstream alternative promoter region, promoting the expression of *IRF5*-short transcripts. To reconcile these at first glance paradoxical observations, we discovered that a zinc finger and BTB/POZ domain-containing protein, ZBTB3, is involved in loop formation despite its repressive function. The zinc finger domain determines the transcriptional specificity of ZBTB proteins by binding to the sequence-specific DNA elements. By contrast, the BTB/POZ domain directly interacts with transcriptional co-repressors, and mediates chromatin remodeling and gene silencing/activation[67]. ZBTB3 affects cancer cell growth[68] and embryonic stem cell self-renewal[69]. While little is known about the function of ZBTB3 in gene regulation, other ZBTB family proteins, such as ZBTB7B, ZBTB4, and ZBTB38, can bind DNA and repress transcription[51,70], implying that ZBTB3 may also act as a transcription repressor. Indeed, consistent with a recent study using motif scanning and EMSA[26], we found that ZBTB3 binds the rs4728142-containing region and reduces its regulatory activities in an allele-specific manner. Moreover, the rs4728142 risk allele shows a more remarkable repressive effect on its upstream enhancer but facilitates short-range loop formation and gene expression activation, which clarifies a potentially function of ZBTB3 in the regulation of chromatin looping. Here, our bioinformatics analysis and experiments provide primary evidence supporting the premise that ZBTB3 may function as a structural factor to mediate the chromatin loop between the rs4728142 locus and the *IRF5* downstream alternative promoter.

Several zinc finger proteins, including YY1[54,55] and ZNF143[71,72], are involved in cohesin-associated chromatin looping and facilitate enhancer–promoter interaction within a topological associated domain (TAD) through CTCF-dependent or -independent mechanisms. Like YY1 and ZNF143, ZBTB3 is also a zinc finger protein, whose genomic motif is usually at about 10 bp (or one DNA helical turn) upstream of the conventional CTCF binding site[73]. Interestingly, we found that ZBTB3 directly binds the rs4728142-containing region and its interaction locus in the *IRF5* downstream alternative promoter with a convergent orientation. Besides, this ZBTB3-mediated binding and loop formation is accompanied by weak CTCF and RAD21 occupancy. In contrast, a recent paper has revealed that, at sites where CTCF binding corresponds to gene activation, there is significant enrichment for ZBTB3, and ZBTB3 might be a CTCF cofactor that mediates gene activation by loop formation in regenerating liver[74]. Another study has

reported that Kaiso, also termed ZBTB33, interacts directly with CTCF, and the Kaiso–CTCF interaction negatively regulates CTCF insulator activity[75]. Although genome-wide and more rigorous evidence is still lacking, existing results and ours support the idea that ZBTB3 might be a CTCF-dependent structural regulator of enhancer–promoter interactions.

Our transcript-level eQTL analysis and CRISPR dKO/sKO/single-base editing assays demonstrated that the rs4728142-containing region not only modulates *IRF5* promoter usage, but also affects global *IRF5* expression. According to the latest FANTOM5 cap analysis of gene expression signatures[76] and GENCODE gene annotations (Supplementary Figs. 5a and 6a), there are at least four *IRF5* alternative promoters and 15 IRF5 protein isoforms. The rs4728142 risk allele increases whole *IRF5*-short transcript expression, but we did not uncover which isoforms are specifically affected. Given the complexity of *IRF5* transcription and isoform expression[47], whether rs4728142 redistributes the expression of IRF5 protein isoforms with different coding sequences or whether it only affects the efficiency of *IRF5* gene expression by modulating the transcription initiation rate or splicing pattern require more in-depth investigations. In addition, recent studies have shown that IRF5 regulates proinflammatory M1 macrophage polarization tending towards improving the inflammatory and immune response, contributing to autoimmune disease occurrence and development[20,22,59]. Consistently, we validated the finding that the genetic effect from the rs4728142 risk allele can transmit to the dysfunction of M1 macrophage polarization. However, understanding why rs4728142 and IRF5 display remarkable pleiotropic effects on different autoimmune diseases requires long-term exploration in the future.

## Methods
### Curation of pleiotropic loci for autoimmune diseases
By searching in PubMed and literature, we curated genome-wide studies for the identification of potential pleiotropic or shared genetic loci associated with at least two autoimmune diseases. Studies based on specific genetic locus or non-European cohorts (not including multi-ancestry study) were excluded during curation process. We collected sentinel pleiotropic variants that obtain a significant *P* value or genetic sharing signal in tested independently associated loci from each curated study, and recorded the associated disease categories regardless of horizontal/vertical pleiotropy or direction of effect size.

### Collection of autoimmune disease GWAS summary statistics
We collected publicly available GWAS summary statistics of autoimmune diseases from two major sources according to the investigated cohorts: UKBB and non-UKBB cohorts. Summary statistics of UKBB cohort were downloaded from Neale Lab UKBB (v3) (http://www.nealelab.is/uk-biobank), Gene ATLAS[77], and GWAS ATLAS (2019 release)[78]. Summary statistics of non-UKBB cohorts were retrieved from several public databases, including GWAS Catalog (v1.0.2)[79] and ImmunoBase Open Targets Genetics (19.05.05) (https://genetics.opentargets.org/immunobase). To remove duplicate GWAS summary statistics among these resources, we identified redundancy by data attributes (by requiring different incorporated samples, quality

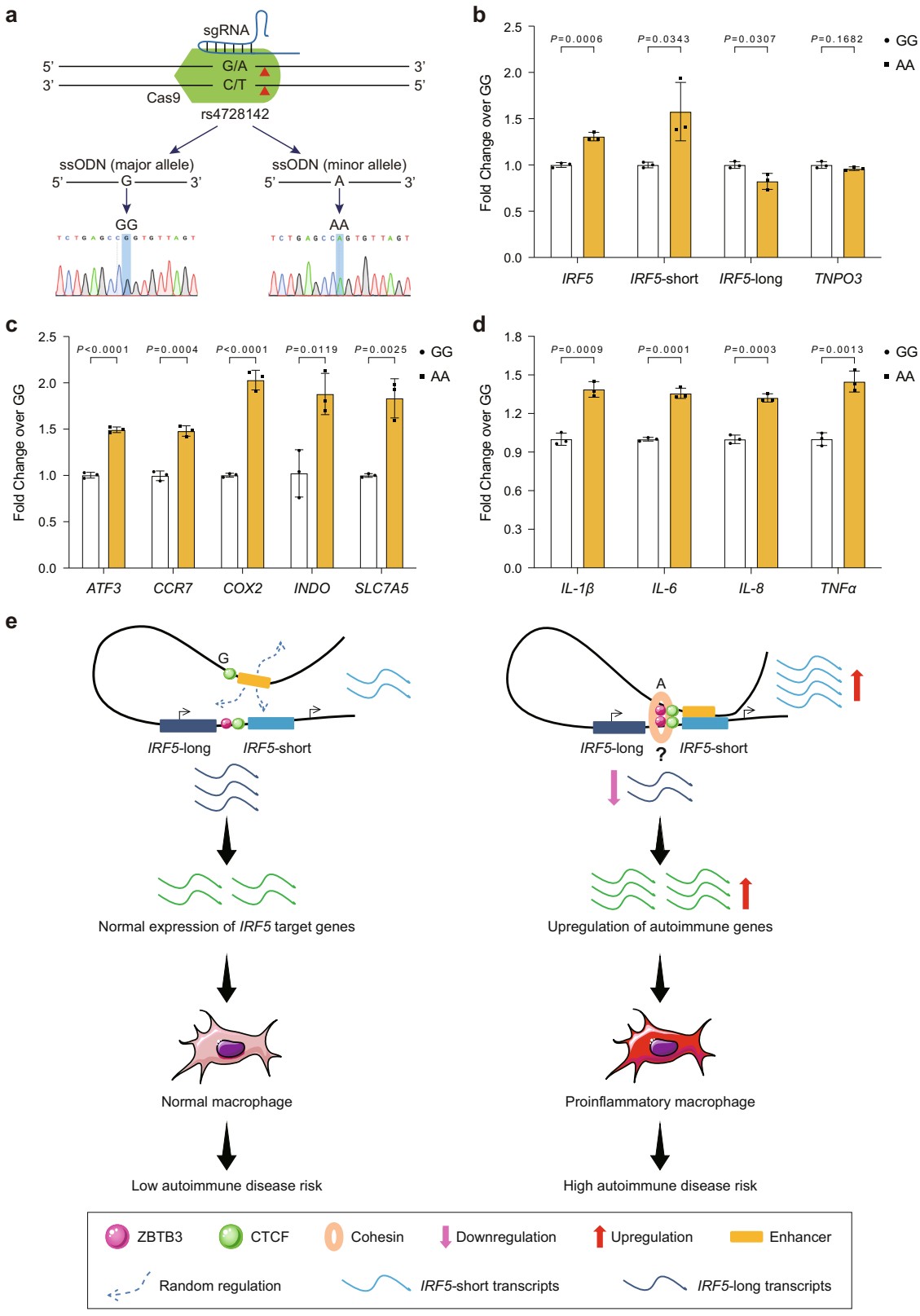

control processes, and association models) and only retained the one with the most information. We excluded GWAS data for non-European cohorts (not including multi-ancestry study).

**GWAS summary statistics standardization**
To ensure that curated GWAS summary statistics are informative for following pleiotropy and fine-mapping analysis, we performed a series of standardization steps on the raw downloaded data. First, we converted recorded coordinates to GRCh37/hg19 coordinates and completed variant information when either the coordinate or rsID was missing. The statistics were excluded when both the coordinates and rsID were missing. Second, we removed summary statistics without explicitly defined effect allele. When only the effect allele was available, the non-effect allele was inferred from 1000 Genomes Project (1KGP)

**Fig. 7 | A model depicting the function and molecular mechanism of rs4728142 in autoimmune diseases. a** Strategy of single-base editing by CRISPR knock-in technology. The sKO sgRNA and single-stranded donor oligonucleotides (ssODN) with major allele G or minor allele A were co-transfected into the single cell-derived SC-Cas9 cells, and single cells were sorted for identification by Sanger sequencing. The red triangle denotes the cutting site. **b** RT-qPCR for *IRF5* transcript expression in rs4728142-AA SC cells compared with rs4728142-GG SC cells. *TNPO3* is the gene adjacent to the upstream *IRF5* gene. **c** RT-qPCR for M1 polarization markers (*ATF3*, *CCR7*, *COX2*, *INDO*, and *SLC7A5*) in rs4728142-AA-induced SC cells compared with rs4728142-GG-induced SC cells. **d** RT-qPCR for key M1 proinflammatory cytokines (*IL-1β*, *IL-6*, *IL-8*, and *TNFα*) in rs4728142-AA-induced SC cells compared with rs4728142-GG-induced SC cells. **e** A model depicting the function and molecular

mechanism of rs4728142 in autoimmune diseases. In nonrisk allele status of rs4728142-G, the upstream promote-like enhancer could freely regulate different *IRF5* alternative promoters or other genes without insulation. However, rs4728142-A steadily establishes specific regulation of the rs4728142-associated enhancer on *IRF5*-short promoter through ZBTB3-mediated chromatin looping. This elaborate regulation of *IRF5* promoter usage leads to increases in both *IRF5*-short transcripts and ensemble gene expression of *IRF5*, ultimately resulting in monocytes/macrophage dysfunction and high autoimmune disease risk. Question mark denotes undetermined ZBTB3 functional manner in the loop formation. Data are represented as the means ± SD, $n = 3$ biologically independent samples, and unpaired two-tailed Student's *t* test is used to calculate *P* values in (**b–d**). Source data are provided as a Source Data file.

biallelic sites, and we excluded variants if the non-effect allele could not be clearly determined. Third, since GWAS fine-mapping methods require both *P* value and effect size (odds ratio (OR) or BETA), we discarded summary statistics that only contain *P* value information in the fine-mapping step. In some cases for which standard error (SE) of BETA or confidence intervals of OR was missing, but the effect size was available, we estimated SE using effect size, *P*, and sample size according to the quantile function.

## Pleiotropy evaluation based on GWAS summary statistics

We performed variant-level pleiotropy analysis based on collected GWAS summary statistics of autoimmune diseases. For each curated pleiotropic locus, we extracted summary data in relatively independent LD blocks of EUR population defined by LDetect (first release)[80]. We inspected the pleiotropic signatures in paired GWAS LD blocks and calculated the posterior probability (PP) of a variant influencing both traits (potential pleiotropy) using gwas-pw (model 3)[30]. Since the overlapping samples or comorbid samples were largely unreported accompanying released GWAS summary statistics, we empirically set the expected correlation (-cor in gwas-pw) between two traits as described in a recent simulation study[81]. For the UKBB cohort studies, we were certain that they contain overlapping samples and incorporate similar categories of traits, we varied the expected correlation of 0.09, 0.18, 0.27, 0.36, 0.45 according to the hierarchy of MeSH tree. For example, the tree numbers of UC and CD are C06.405.469.432.249 and C06.405.469.432.500, respectively, which all belong to C06.405.469.432 parent node of IBD. We hence classified the relationship between these two diseases into Level 4, so the expected correlation in summary statistics between Heart Failure and Atrial Fibrillation was set to 0.36. For non-UKBB cohort studies which we cannot ascertain overlapping samples, the expected correlation was uniformly set to 0.

## Fine-mapping analysis

We performed fine-mapping analysis using autoimmune disease GWAS summary statistics at relatively independent LD blocks that contain curated pleiotropic variants. We required that selected LD blocks (termed causal blocks) had at least one genome-wide significant variant (*P* value ≤5E−8). The GWAS summary information for each causal block was then reformatted into the format required by the used fine-mapping tool. We estimated the LD information of GWAS variants in each LD block using genotype data of EUR population from the 1KGP reference panel, and we only retained biallelic variants and discarded sites with positive minor allele frequency. We mapped test variants to corresponding variants in the reference panel according to identical coordinates and alleles, and harmonized complementary alleles for reverse strands. We calculated pairwise Pearson correlation coefficients between genotypes of each variant in the LD block and constructed block-wise LD matrix. We performed fine-mapping based on summary statistics and matched LD matrix for each causal block of each autoimmune disease using FINEMAP (v1.4)[82]. We assumed that there was only one causal variant in a causal block and used the recommended parameters of the tool. FINEMAP reports the PP of each

variant being causal in the specific model. We recorded potentially causal variants in 95% credible set according to FINEMAP PP.

## Variant annotation and regulatory variant prediction

The pleiotropic variants were annotated using the variant effect predictor (VEP) based on Ensembl GRCh37/hg19 release 98. The most severe consequence for each pleiotropic variant and consequence-associated gene was recorded to represent the variant effect and variant nearest gene. We retrieved regulatory variant prediction scores for pleiotropic variants using RegulomeDB (v1.1)[83] and regBase (v1.1.1)[84]. For tissue/cell-type-specific prediction, we calculated regulatory probability by cepip (2018 release)[85] and GWAS4D (2019 update)[86] on the Roadmap blood-derived primary tissues/cells.

## eQTL colocalization analysis

We identified shared functional effect between an autoimmune disease pleiotropic variant and a blood eQTL by regulatory trait concordance (RTC)[31]. RTC accounted for local LD structure and quantifies residual effect and significance of the eQTL after correcting the gene expression for the genetic effect of given GWAS sentinel variant. We used five whole-genome sequencing eQTL datasets on blood-derived tissue/cell lines (individual genotypes and gene expression quantifications) to perform RTC colocalization analysis, including GTEx V7 whole blood dataset (620 unrelated donors)[87], Geuvadis lymphoblastoid cell line dataset (462 unrelated individuals)[88], BLUEPRINT CD16+ neutrophils, CD14+ monocytes, and CD4+ T cells datasets from 194 samples[44]. Recombination hotspot intervals for defining tested variants were downloaded from ref. [89]. We used the RTC function in QTLtools (v1.1)[90] to investigate the eQTL colocalization effect for each curated pleiotropic variant. First three principal components of individual genotype and 150 PEER factors which determined sample sizes (15 factors for $N < 150$, 30 factors for $150 ≤ N < 250$, 45 factors for $250 ≤ N < 350$, and 60 factors for $N ≥ 350$) estimated from expression data were used as the covariates.

## Chromatin state and transcription factor-binding enrichment

Tissue/cell-type-specific chromatin states estimated by the ChromHMM core 15-state model[91] were downloaded from Roadmap Epigenomics Project (2015 release)[43]. Transcriptional regulator binding sites profiled by ChIP-seq of a large number of transcriptional factors and chromatin remodelers were collected and downloaded from ENCODE[92], Cistrome[93], and DeepBlue (2020 update)[94]. For these collected chromatin features and profiles, we removed redundancy and outliner as described in epiCOLOC (2020 release)[95] and selected profiles in blood-related tissues/cell types to perform enrichment analysis. We tested the associations between tissue/cell-type-specific chromatin state or transcription regulator and autoimmune disease pleiotropic variants using GIGGLE (first release)[96].

## Target gene analysis

We used four methods to link pleiotropic variants to their potential target genes: (1) DEPICT (v1_rel194) genes were identified by

overlapping genes in the high LD region of each pleiotropic variants[32]; (2) RTC genes were obtained using eGenes of colocalized eQTLs; (3) PCHi-C genes were retrieved by assigning pleiotropic variants located in significant interaction regions (interaction score >5) to bait promoter regions using promoter capture Hi-C results of 17 human primary hematopoietic cell types[97]; (4) large effect genes were recorded for pleiotropic variants directly altering protein function or splicing of specific genes. Gene functional enrichment analyses (GSEA) for these target genes on Gene Ontology (GO) and Kyoto Encyclopedia of Genes and Genomes (KEGG) pathway were performed using clusterProfiler (v3.8)[98].

### Pleiotropic variant-gene bipartite network construction

We connected a pleiotropic variant and a gene if this gene was identified as a target gene of observed pleiotropic variant. This operation was performed in all collected autoimmune disease pleiotropic variants and resulted in a variant-gene bipartite network. By compressing all pleiotropic variants from the same autoimmune disease into one node, we investigated and visualized the similarity of pleiotropic genes among all autoimmune diseases with force-directed layout algorithm[99]. This algorithm is an intuitive method to spatially organize network nodes and edges within a two-dimensional plane. Nodes in the network will repel each other as they were like charged bubbles, and each edge will act like a spring to pull a pair of connected nodes together. As a result, autoimmune diseases associated with similar disease genes will be clustered and pushed away from other autoimmune diseases associated with different target genes in the final network, which allows us to observe the genetic architecture among these autoimmune diseases in a globally and flexible manner.

### Cis/trans-gene-regulatory network construction for autoimmune disease pleiotropic variant

For each autoimmune disease pleiotropic variant that colocalizes with GTEx whole blood eQTL by RTC test, we identified all cis- and trans-genes using GTEx V7 whole blood eQTL summary statistics. Using genotypes and whole blood expression data from 620 unrelated individuals, we constructed triplets (pleiotropic variant genotypes/cis-gene expression profiles/trans-gene expression profiles) and filtered ones in which the cis- and trans-gene expression traits were less-correlated (P value >0.05 of Pearson method) to reduce multiple testing burden. We used causal inference test[100] to estimate the causal (variant → cis-gene → trans-gene) and reactive (variant → trans-gene → cis-gene) model P values for each triplet. The test P value is identified using the intersection-union test[100] and corrected by the Bonferroni method (P value <0.05/number of tests). The causal model is confirmed if corrected $P_{\text{causal\_model}}$ <0.05 and corrected $P_{\text{reactive\_model}}$ >0.05. In contrast, if corrected $P_{\text{causal\_model}}$ >0.05 and corrected $P_{\text{reactive\_model}}$ <0.05, it is confirmed to the reactive model. Otherwise, the independent model is retained. We constructed gene regulatory network for autoimmune disease pleiotropic variants using the cis- or trans-relationships identified from the causal inference test. Nodes in the network are cis- or trans-genes, and edges are directional connections.

### Causal regulatory variants prioritization

We aggregated the causal and functional evidence to prioritize regulatory potential of the collected pleiotropic variants, including (1) whether the pleiotropic variant could be reproduced using public autoimmune disease GWAS data; (2) whether the pleiotropic variant is likely causal variant in the 95% fine-mapping credible set for specific autoimmune diseases; (3) whether colocalization is observed (RTC score ≥0.9) between the pleiotropic variant and expression quantitative trait loci (eQTLs) based on five blood-derived whole-genome sequencing eQTL datasets; and (4) whether the pleiotropic variant is highly scored with regulatory potential (regBase (v1.1.1)[84] PHRED score

≥15 and cepip (2018 release)[85] combined score ≥0.6 and RegulomeDB (v1.1)[83] score ≥2c). Pleiotropic variants satisfy all above conditions were selected as candidate causal regulatory variants and were finally ranked based on the number of shared autoimmune diseases according to gwas-pw (model 3) estimation.

### Causal immune cell-type estimation

We used g-chromVAR (first release)[40] to infer likely causal immune cell types for autoimmune diseases based on GWAS summary statistics, fine-mapping results and chromatin accessibility profiles for 18 hematopoietic progenitors, precursors, and differentiated cells. We only selected the GWAS summary data with the largest identified risk loci for each autoimmune disease. The method weights chromatin accessibility features according to fine-mapped variant posterior probabilities and computes for each cell-type enrichment versus an empirical background matched for GC content and feature intensity. ATAC-seq profiles of hematopoietic and leukemic cell types, across 12 normal hematopoietic cell types were downloaded from GEO GSE74912 and uniformly processed by ENCODE ATAC-seq processing pipeline[101]. Gene expression comparison across 15 immune cells were conducted using the DICE database (2019 release)[41].

### Public genomic, epigenomic, and three-dimensional (3D) genome data annotation

The transcript and promoter annotations of IRF5 were retrieved from GENCODE V28[102] and FANTOM CAT[76]. The DNase I hypersensitive site and histone modification (H3K27ac, H3K4me1, and H3K4me3) public data on GM12878 (E116) tissue and Monocyte (E124) tissue used in this study were downloaded from Roadmap Epigenomics Project (2015 release)[43] website. The MNase-seq on human GM12878 was downloaded from ENCODE ENCSR000CXP. High-resolution Hi-C data on human GM12878[103] and contact map visualization were obtained from the 3D Genome Browser (2019 release)[104]. RAD21 ChIA-PET data on human GM12878 was downloaded from ENCODE ENCSR981FNA and processed by CID[105] and ChIA-PET2[106].

### Transcript-level gene expression quantitative trait loci mapping

In order to be consistent with the GTEx RNA-seq quantification method, the mRNA sequencing data from the Geuvadis and BLUEPRINT Epigenome Project was re-analyzed. The Transcripts Per Kilobase Million (TPM) produced from RSEM (v1.3.2)[107] was used to quantify gene expression level. Linear regression model[108] was used to analyze the association between rs4728142 and IRF5 expression level. The expression level of each IRF5 transcript was quantified using RSEM (v1.3.2). Then, the transcripts were classified into two groups according to the location of the transcription start point. The transcripts which the transcription start point located at upstream of the contact locus in IRF5 promoter with the highest interaction frequency were referred to as IRF5-long transcripts, otherwise IRF5-short transcripts. The TPM value was calculated to estimate the expression level of each group transcript. Linear regression model[108] was used to analyze the association between rs4728142 alleles and the transcript expression level.

### Cell culture

Human peripheral blood monocyte-derived SC cell line (American Type Culture Collection [ATCC], CRL-9855) was cultured in Iscove's Modified Eagle's Medium (IMDM; ThermoFisher, 31980030) containing 10% Fetal Bovine Serum (FBS; ThermoFisher, 16140071). 293T cell line (ATCC, CRL-3216) and 293FT cell line (ThermoFisher, R70007) were cultured in Dulbecco's Modified Eagle's Medium (DMEM; ThermoFisher, 11965092) containing 10% FBS, and penicillin and streptomycin (ThermoFisher, 15070063). All cells were maintained in an incubator with 5% $CO_2$ at 37 °C. All cell lines were authenticated by STR and tested no mycoplasma contamination utilizing the MycoAlert Mycoplasma Detection Kit (Lonza, LT07-118).

## Cell line locus genotyping

Genomic DNA was extracted from SC cells using a commercial DNA isolation kit (QIAGEN, 51304) according to the kit's instructions. Genotyping PCR was performed with the Phanta Max Super-Fidelity DNA Polymerase (Vazyme, P505). Primers used for genotyping are as follows: forward primer, 5′-GCAGGCATCAGAGAGGAC-3′; and reverse primer, 5′-CTCTGGTTCTCGTACTTGAAG-3′. Cycling parameters utilized are as follows: 95 °C for 3 min, 32 cycles of 95 °C for 15 s, 60 °C for 15 s, 72 °C for 20 s, followed by 1 cycle of 72 °C for 5 min. After PCR reaction, the products were subjected to treatment with the ExoSAP-IT PCR Product Cleanup Reagent (ThermoFisher, 78201.1.ML) according to the manufacturer's instructions. The purified PCR products were subsequently sent to the Beijing Genomics Institute (BGI) for Sanger sequencing. The sequencing results were analyzed and visualized by the SnapGene Viewer (v4.3.10) software.

## ATAC-seq

ATAC-seq assay was performed[109]. SC cells were pretreated with DNase I (Worthington, LS002007) to digest DNA from dead cells and to remove free-floating DNA. The cells were counted with Crystal Violet (Sigma-Aldrich, C0775), and ~50,000 cells (for each sample) were resuspended in 1 ml cold ATAC-Resuspension Buffer (RSB; 3 mM MgCl$_2$ [Sigma-Aldrich, M2670], 10 mM Tris-HCl pH 7.5 [ThermoFisher, 15567027], and 10 mM NaCl [Sigma-Aldrich, S7653] in sterile H$_2$O). After centrifugation, the supernatant was discarded, and the cell pellets were resuspended in 50 µl cold RSB containing 0.1% Tween-20 (Sigma-Aldrich, P1379), 0.1% NP-40 (ThermoFisher, FNN0021), and 0.01% Digitonin (Sigma-Aldrich, D141). After lysis, 1 ml RSB containing 0.1% Tween-20 was added, and the tube was inverted several times to mix. After centrifugation, the supernatant was removed, and the nuclei were resuspended in 50 µl Transposition Mix (Vazyme, TD501). Then, the transposition reaction was incubated at 37 °C for 25 min with shaking at 1200×g. The reaction products were purified with the QIAquick PCR Purification Kit (QIAGEN, 28106). The ATAC-seq library was prepared using the TruePrep DNA Library Prep Kit V2 for Illumina (Vazyme, TD501) and TruePrep Index Kit V2 for Illumina (Vazyme, TD202) according to the manufacturer's instructions. The ATAC-seq analysis sequences with adapter and low quality were trimmed for all 150-bp pair-end ATAC-seq reads. The clean reads were aligned to the human reference genome (GRCh37/hg19) using Bowtie2 (v2.2.1)[110] in the default parameters. All the samples were separately processed, and duplicate reads were removed. The ATAC-seq peak regions of each sample were called using MACS2 (v2.1.1)[111] in the default parameters. The results were analyzed and visualized in IGV (v2.8.6)[112].

## Chromatin immunoprecipitation (ChIP) and sequencing

ChIP assays were performed[113]. For each sample, ~1 × 10$^7$ SC cells were cross-linked with 1% formaldehyde (Sigma-Aldrich, F8775) in 10% FBS/phosphate-buffered saline (PBS) for 10 min at room temperature and quenched with 0.125 M glycine (Sigma-Aldrich, 50046). The cells were then collected and lysed twice with ice-cold lysis buffer (20 mM Tris-HCl pH 7.5 [ThermoFisher, 15567027], 2 mM EDTA [ThermoFisher, 15575020], 1% Triton X-100 [Sigma-Aldrich, T8787], 0.1% SDS [ThermoFisher, 15525017], 0.1% Sodium Deoxycholate [Sigma-Aldrich, D6750], and 1× Protease Inhibitors [ThermoFisher, 78425]) for 10 min with slow rotations at 4 °C. The nuclear lysates were sonicated to obtain DNA fragments of 200–500 bp using a Biorupter system (Diagenode, B01060001) by high energy, with a working time of 30 s and a resting time of 30 s, 30 cycles, until the solutions turned clear. After removing insoluble debris, the lysates were separately incubated with specific antibodies against Rabbit IgG (1:100, Cell Signaling Technology, 7074), CTCF (1:100, Abcam, ab128873), ZBTB3 (1:100, Novus Biologicals, NBP1-82079), RAD21 (1:100, Abcam, ab992), H3K4me1 (1:100, Cell Signaling Technology, 5326), H3K4me3 (1:100, Cell Signaling Technology, 9751), or H3K27ac (1:100, Abcam, ab4729),

and then purified by protein A-agarose beads (Millipore, 16–157). The beads were washed four times by TE buffer (10 mM Tris-HCl pH 8.0 [ThermoFisher, 15568025] and 1 mM EDTA), TSE I buffer (10 mM Tris-HCl pH 8.0, 150 mM NaCl, 2 mM EDTA, 1% Triton X-100, and 0.1% SDS), TSE II buffer (20 mM Tris-HCl pH 8.0, 500 mM NaCl, 2 mM EDTA, 1% Triton X-100, and 0.1% SDS), and Buffer III (10 mM Tris-HCl pH 8.0, 1 mM EDTA, 1% NP-40, 0.25 M LiCl [Sigma-Aldrich, L4408], and 1% sodium deoxycholate) before elution (20 mM Tris-HCl pH 7.5, 5 mM EDTA, 50 mM NaCl, and 1% SDS), reverse cross-linking, and subsequent DNA extraction. ChIP-seq libraries were generated using the VAHTS Universal DNA Library Prep Kit for Illumina V3 (Vazyme, ND607) according to the kit's instructions and then sequenced on a HiSeq X Ten platform (Illumina). For data analysis, the sequences with adapter and low quality were trimmed for all 150-bp pair-end reads. The clean reads were mapped against the human reference genome (GRCh37/hg19) using BWA (v0.7.17)[114]. All the samples were separately processed, and the unique mapped reads were kept. The peaks were called by MACS2 (v2.1.1)[111] using IgG as a control. The results were analyzed and visualized in IGV (v2.8.6)[112]. For ChIP-qPCR, relative fold enrichment was calculated using the comparative Ct method normalizing to the input sample. Data in bar graphs represent the means ± standard deviations (SD). Primers used are listed in Supplementary Data 14. The ChIP-seq colocalization was analyzed using deepTools (v3.4.3)[115].

## Luciferase reporter assay

The genomic sequences (GRCh37/hg19 chr7: 128,573,817–128,574,098) at rs4728142-containing region with different alleles (G or A), the element sequence (GRCh37/hg19 chr7: 128,573,081–128,573,816) adjacent to the downstream rs4728142-containing region spanning the H3K4me1 and H3K4me3 ChIP-seq peaks, and the integrated sequences (GRCh37/hg19 chr7: 128,573,081–128,574,098) harboring the upstream element and rs4728142-containing region with different alleles (G or A), were amplified from the genomic DNA of SC cells with Premix Taq enzyme (TaKaRa, RR901A), and then cloned into upstream of the SV40 promoter or downstream of the luciferase gene in the pGL3-Promoter vector (Promega, E1761), or drove a firefly luciferase gene in the pGL3-Basic vector (Promega, E1751). In addition, the putative *IRF5*-short promoter sequence (GRCh37/hg19 chr7: 128,579,302–128,580,415) which interacts with the rs4728142-associated enhancer was amplified and cloned into the pGL3-Basic vector. We used the pGL3-*IRF5*-short vector to construct all of the luciferase plasmids similar to the pGL3-Promoter vector, and the related rs4728142 knockout (KO) plasmids in the enhancer assays. After Sanger sequencing, concentrations of the empty vector and recombinant plasmids were exactly determined by the Qubit dsDNA HS Assay Kit (ThermoFisher, Q32851). Each plasmid with the same copy numbers (the amount is equal to 1 µg of the empty vector) together with 40 ng of pRL-TK Renilla Luciferase vector (Promega, E2241) was co-transfected into 293T cells in 24-well plates by the Lipofectamine 2000 transfection reagent (ThermoFisher, 11668019) or into SC cells in 24-well plates by an electroporation system (BIO-RAD, 165-2660, 165-2661, and 165-2667). After 12 h, the transfected cells were collected and lysed before detecting luciferase activities using the Dual-Luciferase Reporter Assay System (Promega, E1960) on a Promega GloMax 20/20 (Promega, E5311). Relative luminescent signals were determined by normalizing firefly luciferase signals to renilla luciferase signals. Primers used are listed in Supplementary Data 14.

## Circularized chromosome conformation capture (4C) assay

4C assays were conducted[46,116]. SC cells were counted, and ~1 × 10$^7$ cells were used for each 4C experiment. The cells were cross-linked with 1.5% formaldehyde, quenched with 0.125 M glycine, lysed with cold lysis buffer (50 mM Tris-HCl pH 7.5, 150 mM NaCl, 5 mM EDTA, 0.5% NP-40, 1% Triton X-100, and 1× Protease Inhibitors), digested with *Dpn*II (NEB, R0543M) for 4 h at 37 °C while shaking at 1000×g for more

than three times until the digestion efficiency reached over 90%, and then ligated with T4 DNA ligase (NEB, M0202M) in a 7-ml reaction system after enzyme inactivation. After reversing the cross-links by adding Protein K (ThermoFisher, AM2548) for overnight at 65 °C and RNase A for 45 min at 37 °C, the ligation products were purified by phenol/chloroform/isoamyl alcohol (25:24:1) (ThermoFisher, 15593-049). The purified DNA samples were digested with *Nla*III (NEB, R0125L) for the rs4728142-containing region (GRCh37/hg19 chr7: 128,573,714–128,574,338) or *Cvi*QI (NEB, R0639L) for the enhancer (GRCh37/hg19 chr7: 128,573,036–128,573,717) adjacent to the downstream rs4728142-containing region, and then ligated with T4 DNA ligase for overnight in a 14-ml reaction system. The ligation samples were purified by phenol/chloroform/isoamyl alcohol (25:24:1) and further purified utilizing the QIAquick PCR Purification Kit. The purified products were amplified by VAHTS HiFi Amplification Mix (Vazyme, N616) with 8 × 200 ng DNA. 4C-seq libraries were generated using the VAHTS Universal DNA Library Prep Kit for Illumina V3 according to the kit's instructions and then sequenced on a HiSeq X Ten platform. For data analysis, the quality control step was processed for the 4C-seq library using FastQC. The sequences with adapter and low quality were discarded for further analysis. A Bioconductor package Basic4Cseq (v1.34)[117] was used to filter, analysis, and visualize the 4C-seq data. Primers used are listed in Supplementary Data 14.

## Chromosome conformation capture (3C) assay

In solution 3C assays were performed[118,119]. We chose the rs4728142-containing region (GRCh37/hg19 chr7: 128,573,714–128,574,338) and the enhancer (GRCh37/hg19 chr7: 128,573,036–128,573,717) adjacent to the downstream rs4728142-containing region as the viewpoints which are the same as the 4C assays. First, for each sample, a total of $1 \times 10^7$ SC cells were cross-linked with 1% formaldehyde for 8 min. Second, the cross-linked cells were digested with several times to increase the digestion efficiency[120]. Third, to acquire the correct PCR product, we selected specific primers and optimized extending time of PCR performed with the Premix Taq enzyme. Optimal PCR conditions are as follows: 1 cycle at 95 °C, 5 min; 28 recycles at 95 °C (30 s), 55–60 °C (1 min), and 72 °C (10–20 s); and 1 cycle 72 °C, 5 min. As a reference control, the sequence covering all fragments was amplified by the Phanta Max Super-Fidelity DNA Polymerase and subjected to the equal treatment as the experimental group. The PCR products were electrophoresed on 1% agarose gels and then analyzed by ImageJ2 (ver. 10.2) software (https://imagej.net/ImageJ). The 3C interaction frequency of each chromatin fragment was normalized to its reference control. To investigate whether the interaction between the rs4728142-containing region and the *IRF5* downstream alternative promoter region has an allele-specific manner, a couple of primers were designed by Primer Premier 6.25 software (http://www.premierbiosoft.com/), and the ligation sequence incorporating both the 3C bait (GRCh37/hg19 chr7: 128,573,714–128,574,338) and its highest interaction frequency fragment (GRCh37/hg19 chr7: 128,578,363–128,578,537) was amplified with the Phanta Max Super-Fidelity DNA Polymerase. The correct PCR product was identified by electrophoresing on a 1% agarose gel and sequenced by Sanger sequencing. The sequencing results were analyzed and visualized by SnapGene Viewer (v4.3.10) software (https://www.snapgene.com/). Primers used are listed in Supplementary Data 14.

## CRISPR activation (CRISPRa) and CRISPR interference (CRISPRi) assays

Single cell-derived stable SC-dCas9-VP64 and SC-dCas9-KRAB cell lines were generated using the Lenti-dCas9-VP64-blast (Addgene, 61425) and Lenti-dCas9-KRAB-blast (Addgene, 89567) vectors. Lentiviral particles were generated in 293FT cells using the pMD2.G (Addgene, 12259) and psPAX2 (Addgene, 12260) packaging plasmids. SC cells were separately transduced the lentiviruses at a multiplicity of

infection (MOI) of 0.1 for 24 h, and then selected with 100 µg/ml Blasticidin (ThermoFisher, 461120) for 4 days. The single cell-derived stable SC-dCas9-VP64 and SC-dCas9-KRAB cells were acquired by gradient dilution and validated by western blotting. sgRNAs targeting the rs4728142-associated enhancer were designed by CHOPCHOP (v3) Web Portal[121] and ordered from BGI. Subsequently, the synthetic sgRNA sequences were cloned into the pGL3-U6-sgRNA-EGFP plasmid (Addgene, 107721) by using Stbl3 Competent Cells (ThermoFisher, C737303). The empty vector and recombinant plasmids with equal quantities were separately transfected into the SC-dCas9-VP64 and SC-dCas9-KRAB cell lines using a nucleofector device (4D-Nucleofector, Lonza) and the 4D-NucleofectorTM LV Kit XL (Lonza, V4LP-3520). After 48 h, total RNAs were extracted for RT-qPCR as mentioned below. Primers used are listed in Supplementary Data 14.

## RNA isolation, RT-qPCR, and sequencing

For RNA isolation, total RNA was extracted by TRIzol (ThermoFisher, 15596018) according to the manufacturer's instructions. For Quantitative real-time RT-PCR (RT-qPCR)[122], first-strand cDNA synthesis was carried out using the RT Master Mix for qPCR (gDNA digester plus) (MCE, HY-K0511). Each RT-qPCR was performed with ~200 ng or 1 µg of DNase-treated RNA and 10 nM primer pair solutions utilizing the SYBR Green RT-qPCR Mix Kit (ThermoFisher, 4368706). Gene transcription was normalized to human β-actin, and fold changes were calculated as described in the figure legends. Primers used are listed in Supplementary Data 14. For RNA-seq experiments, the freshly extracted total RNAs with high qualities were sent to the Novogene (Beijing, China) for constructing libraries and sequencing. RNA-seq reads were filtered by removing reads containing adapter and low quality. The reference genome (GRCh37/hg19) was index using STAR (2.7.7a)[123], and 150-bp paired-end reads were mapped to the human reference genome using STAR (2.7.7a) with the default parameters. GENCODE v28 annotation file was used to guide the STAR (2.7.7a) alignment step, and RSEM (v1.3.2)[107] was used to quantify the reads count of each gene in the genome assembly GRCh37/hg19. Gene differential expression analysis was performed with these reads count using the Bioconductor package DESeq2 (v1.28)[124] in R. Fold Change also calculated in DESeq2 (v1.28). Genes with an adjusted $P$ value <0.05 and |Log$_2$Fold Change| >1 were selected as differential expression genes. The KEGG pathway and GSEA analyses were performed with the Bioconductor package clusterProfiler (v3.8)[98].

## Genome editing in SC cells

For the rs4728142 dKO and sKO assays, sgRNAs for dKO and sKO were designed by CRISPOR (v4.99) Web Portal[125], and then separately cloned into the pGL3-U6-sgRNA-EGFP plasmid. Single cell-derived stable SC-Cas9 cell line was generated using the LentiCas9-Blast (Addgene, 52962) vector as mentioned above. The recombinant plasmids (couple plasmids for dKO) were then separately transfected into the single cell-derived SC-Cas9 cells using a nucleofector device (4D-Nucleofector, Lonza) and the 4D-Nucleofector LV Kit XL. For single-base editing, the sKO sgRNA and single-strand donor oligonucleotides (ssODN) with G or A allele were co-transfected into the SC-Cas9 cell line. After transfection, 1 µg/ml Puromycin (Sigma-Aldrich, P7255) was added at 24 h. After 48 h, the single cells with GFP fluorescence were sorted in 96-well plates by Flow cytometry (BD, FACS Aris II). After expanding culture of the sorted cells from 96-well plates to 24-well plates, genomes of the sorted cells were separately extracted using a commercial DNA isolation kit (QIAGEN, 51304). The sequences harboring the edited loci were amplified with the Premix Taq enzyme, and the products were subsequently sent to BGI for Sanger sequencing. The sequencing results were analyzed and visualized on CRISP-ID Web Portal[126] or SnapGene Viewer (v4.3.10) software. Primers and ssODN used are listed in Supplementary Data 14.

## Motif analysis

Potential TFs that bind the sequence at rs4728142 locus and its inter-active *IRF5* downstream alternative promoter region were analyzed using the rs4728142 viewpoint sequence (GRCh37/hg19 chr7: 128,573,714–128,574,338) and its capture sequence (GRCh37/hg19 chr7: 128,578,363–128,578,537) in the *IRF5* downstream alternative promoter with the highest interaction frequency using CIS-BP (Build 2.00)[49] and FIMO (v5.4.1)[50]. ZBTB3 motifs, PB0195.1 and PB0091.1, were downloaded from the JASPAR database (2020 release)[127]. We chose 21-bp sequences surrounding at rs4728142 with different alternative alleles (5′-GCTCTGAGCCG(A)GTGTTAGTAA-3′, left base "G" denotes the nonrisk allele, and "A" in brackets denotes the risk allele) and a 34-bp sequence at the *IRF5* alternative promoter region with the highest interaction frequency (5′-CCCGGGAGCCCCGCTGGAGGCTGGCTTG-GACCAC-3′) to analyze the detailed motif information. The UniPROBE database (2019-03-03)[128] was used to visualize the motif sequence.

## Electrophoretic mobility shift assay (EMSA)

Complementary oligonucleotides (GRCh37/hg19 chr7: 128,573,950–128,573,984) encompassing rs4728142 with different alleles (G or A) or the putative ZBTB3 binding locus (GRCh37/hg19 chr7: 128,578,485–128,578,514) in the *IRF5* alternative promoter region were designed and synthesized. One strand of the complementary oligo-nucleotides was labeled with a biotin at the 5′-terminal end, and the same quantities of the biotin-labeled oligonucleotide or biotin-unlabeled oligonucleotide together with its complementary biotin-unlabeled oligonucleotide was annealed to generate biotin-labeled probes or competitor probes by heating and gradient cooling. Nuclear extracts from SC cells were prepared with the NE-PER Nuclear and Cytoplasmic Extraction Reagents (ThermoFisher, 788333) according to the manufacturer's instructions. In all, 20 nM of the biotin-labeled probes were incubated with 2 μg of the freshly extracted nuclear extracts or 10 ng of ZBTB3 recombinant protein (Novus Biologicals, H00079842-P01) at room temperature of 20 min in the presence of 100 mM Tris (pH 7.5), 55 mM KCl (Sigma-Aldrich, P9333), 1 mM DTT (ThermoFisher, P2325), 500 ng Poly[d(I-C)] (Roche, 10108812001), 5% glycerol (Sigma-Aldrich, G2025), 0.05% NP-40, and 2.5 mM MgCl$_2$ in a 20 μl reaction system. Competitive binding assays were performed under the same condition, with the addition of the biotin-unlabeled probes (500 times concentrations than the biotin-labeled probe) prior to the addition of the biotin-labeled probe. For antibody pull-down assay, 20 ng of anti-ZBTB3 antibody (Novus Biologicals, NBP1-82079) was added to nuclear extracts before adding the biotin-labeled probes. The protein-DNA complexes were electrophoresed on 6% native polyacrylamide gels for 1.5 h at 100 V in 0.5× TBE buffer (Promega, V4251), and then transferred to an Immobilon-NY+ Nylon Membrane (Millipore, INYC00010) for 30 min at 380 mA. The membrane was cross-linked by UV irradiation, blocked with 5% nonfat milk, incubated with streptavidin labeled with horseradish peroxidase (HRP; Cell Sig-naling Technology, 3999), and then visualized by a Protein Imaging System (BIO-RAD, ChemiDoc XRS+). The oligonucleotide sequences used are listed in Supplementary Data 14.

## Overexpression and knockout of ZBTB3

For ZBTB3 overexpression, ZBTB3 coding sequence was amplified from the complementary DNA (cDNA) of SC cells, and then cloned into the pCMV-Tag2B vector (Stratagene, 211172) with a N-terminal Flag tag. Empty vector and pCMV-Tag2B-ZBTB3 plasmid with equal quantities were separately transfected into SC cells using a nucleofector device (4D-Nucleofector) and the 4D-NucleofectorTM LV Kit XL. After 48 h, western blotting was performed to validate the ZBTB3 overexpression as mentioned below. Then, total RNAs were extracted for RT-qPCR as mentioned above. For ZBTB3 knockout, sgRNA targeting ZBTB3 cod-ing region in the second exon was designed on the CRISPOR (v4.99) Web Portal[125], and then cloned into the pGL3-U6-sgRNA-EGFP plasmid

(Addgene, 107721) using restriction enzyme *Bsa*I (New England Bio-labs, R0535S). The recombinant plasmid was transfected into the sin-gle cell-derived SC-Cas9 cell line using a nucleofector device (4D-Nucleofector, Lonza) and the 4D-Nucleofector LV Kit XL as mentioned above. The single GFP-positive cells were sorted into 96-well micro-plates by Flow cytometry (BD, FACS Aris II). After expanding culture of the sorted cells, the ZBTB3 knockout was validated by western blot-ting. Then, total RNAs of the positive cells were extracted for RT-qPCR and the negative cells were used as control. Primers used are listed in Supplementary Data 14.

## Western blotting

For each sample, $1 \times 10^6$ cells were lysed in 100 μl RIPA Lysis and Extraction Buffer (ThermoFisher, 89901) by adding 1× Protease Inhi-bitor Cocktail (ThermoFisher, 78425) and phenylmethanesulfonyl fluoride (PMSF) (Sigma-Aldrich, 78830). The lysates were incubated on ice for 30 min and mixed adequately every 10 min. Then, the lysates were centrifuged at 14,000×*g* for 15 min to remove debris, and con-centrations of protein lysates were quantified by a Protein Quantitative Kit (ThermoFisher, 23235). The protein lysates were diluted to the same concentration, and equal quantities of different samples were mixed with Laemmli Buffer (BIO-RAD, 1610747). Proteins were resolved by 10% SDS-PAGE, transferred to a polyvinylidene difluoride (PVDF) membrane (ThermoFisher, 88585), and then the membrane was blocked with 5% nonfat milk overnight at 4 °C. Next, the PVDF membrane was incubated with rabbit anti-human GAPDH antibody (1:5000, Abclonal, AC001), rabbit anti-human ZBTB3 anti-body (1:1000, Abcam, ab106536) or Rabbit anti-human β-actin antibody (1:50,000, Abclonal, AC026), and incubated with goat anti-rabbit antibody labeled with HRP (1:5000, ThermoFisher, G-21234) after five washes with TBST buffer (ThermoFisher, TA-999-TT). When the above steps were finished, the membrane was washed five times with TBST buffer. Lastly, the membrane was developed color by ECL Substrate (ThermoFisher, 34096) and detected using a Protein Imaging System (BIO-RAD, ChemiDoc XRS+).

## Induction of monocyte-derived M1 macrophages

Phorbol 12-myristate 13-acetate (PMA) (Sigma-Aldrich, P8139) and Lipopolysaccharide (LPS) (Sigma-Aldrich, L2630) were used to induce monocyte-derived M1 macrophages[60]. In all, $2 \times 10^6$ SC-derived cells with high activities were inoculated into the chambers of six-well plates, and PMA (final concentration: 50 ng/μl) was added at 24 h. After inducing the cells for 2 days, the medium was discarded, fresh medium (IMDM + 10% FBS) with LPS (final concentration: 100 ng/μl) was added, and then the cells were cultured for 2 days. Changes of cell structure and morphology were observed and photographed under a micro-scope (OLYMPUS, CKX53).

## Statistical analysis

RStudio (version 1.0.136) with R (version 3.3.3) was used to analyze all statistical analyses in silico. GraphPad Prism version 8.1.1 software (La Jolla, CA) was used to analyze all experimental data. Data were shown as the means ± standard deviations (SD), "*n*" represents the number of technical replicates of the representative biological replicates unless otherwise mentioned. An unpaired two-tailed Student's *t* test was used to define statistical significance for the experimental data analyses. *P* values of less than 0.05 were considered statistically significant.

## Reporting summary

Further information on research design is available in the Nature Portfolio Reporting Summary linked to this article.

## Data availability

The data that support this study are available from the corresponding authors upon reasonable request. All sequencing data generated in

this study have been deposited in the Gene Expression Omnibus (GEO) database under accession "GSE168045". The data generated in this study are provided in the Supplementary Information/Source Data file and based on human reference (GRCH37/hg19). The ATAC-seq profiles of hematopoietic and leukemic cell types, across 12 normal hematopoietic cell types data used in this study are available in the GEO database under accession "GSE74912". Source data are provided with this paper.

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

## Acknowledgements

The work was supported by the following grants: the National Natural Science Foundation of China (32000640 to Z.W., 31871327 to M.J.L., and 32070675 to M.J.L.), and the Tianjin Committee of Science and Technology (19JCJQJC63600 to M.J.L.).

## Author contributions

Z.W., X.Y.W., W.H.S., and M.J.L. designed the studies and wrote the manuscript. Z.W., Q.L., X.Y.Q., B.L.H., K.Z., Y.L.H., and Z.K.B. performed the experiments and analyzed the data. Z.Y.Z., Z.W., J.H.W., Y.Z., X.L.F., X.F.Y., and M.J.L. conducted the bioinformatics and data analysis. J.D.S., J.L., Z.L., J.H.H., K.X.C., Y.Y., P.C.S., and W.G.L. contributed scientific expertise and reviewed the manuscript. All the authors read and approved the manuscript.

## Competing interests

The authors declare no competing interests.

## Additional information

[1]Department of Bioinformatics, The Province and Ministry Co-sponsored Collaborative Innovation Center for Medical Epigenetics, Key Laboratory of Immune Microenvironment and Disease (Ministry of Education), National Clinical Research Center for Cancer, Tianjin Medical University Cancer Institute and Hospital, Tianjin Medical University, Tianjin, China. [2]Department of Pharmacology, Tianjin Key Laboratory of Inflammation Biology, School of Basic Medical Sciences, Tianjin Medical University, Tianjin, China. [3]Oujiang Laboratory (Zhejiang Lab for Regenerative Medicine, Vision and Brain Health), Institute of Aging, Key Laboratory of Alzheimer's Disease of Zhejiang Province, The Second Affiliated Hospital, Wenzhou Medical University, Wenzhou, China. [4]Scientific Research Center, Wenzhou Medical University, Wenzhou, China. [5]School of Biomedical Engineering, Tianjin Medical University, Tianjin, China. [6]Department of Cell Biology, Tianjin Key Laboratory of Medical Epigenetics, School of Basic Medical Sciences, Tianjin Medical University, Tianjin, China. [7]State Key Laboratory of Medicinal Chemical Biology and College of Life Sciences, Nankai University, Tianjin, China. [8]Department of Immunology, Tianjin Key Laboratory of Medical Epigenetics, School of Basic Medical Sciences, Tianjin Medical University, Tianjin, China. [9]Department of Pancreatic Cancer, Tianjin Medical University Cancer Institute and Hospital, National Clinical Research Center for Cancer, Key Laboratory of Cancer Prevention and Therapy, Tianjin, China. [10]Department of Epidemiology and Biostatistics, Tianjin Key Laboratory of Molecular Cancer Epidemiology, Tianjin Medical University Cancer Institute and Hospital, Tianjin Medical University, Tianjin, China. [11]Centre for PanorOmic Sciences-Genomics and Bioinformatics Cores, The University of Hong Kong, Hong Kong, China. [12]Department of Gastroenterology, The Third Xiangya Hospital, Hunan Key Laboratory of Non-resolving Inflammation and Cancer, Central South University Changsha, Hunan, China. [13]These authors contributed equally: Zhao Wang, Qian Liang, Xinyi Qian, Bolang Hu. ✉e-mail: wangzhao19880923@wmu.edu.cn; wangxiaoyan@csu.edu.cn; weihong@wmu.edu.cn; mulinli@connect.hku.hk

