## [Peer Review File · Nature Communications]

An autoimmune pleiotropic SNP modulates IRF5 alternative promoter usage through ZBTB3-mediated chromatin loopingREVIEWER COMMENTS

Reviewer #1 (Remarks to the Author):

In this manuscript, Wang and colleagues integrated GWAS and epigenomics analysis indicating rs4728142 as a causal regulatory variant in autoimmune disease and may act in monocyte-macrophage lineage cells. They used CRISPR editing tool demonstrating the regulation of rs4728142-containing region on IRF5 expression and found rs4718142-containing region differentially regulating IRF5 long and short isoform involving disease pathogenesis. Mechanistically, they proposed rs4728142 alleles differentially bind with ZBTB3 and regulate loop formation between the rs4728142-containing region and the promoter region of IRF5 short isoform and long isoform. Their assumption is interesting, but they need more solid data to support the major conclusions.

1. The author proposed IRF5 short isoform is involved in the development of disease. In many autoimmune diseases, IRF5 expression is abnormally up-regulated, and what about the expression change of IRF5 short isoform and long isoform in autoimmune diseases such as RA, SLE, and so on? What's the mechanism that can explain different isoforms of IRF5 have different effects on the disease process?

2. Fig.4 shows rs4728142-containing region could form gene loop with IRF5 promoter region, and deletion of rs4728142 site decreased or increased IRF5 short isoform expression or IRF5 long isoform expression respectively. Whether deletion of this region can disrupt the loop formation? Or deletion of this region can enhance the loop switching from the promoter of IRF5 short isoform to the promoter of IRF5 long isoform?

3. The distance between rs4728142 and IRF5 promoter is very short, especially for the promoter of IRF5 long isoform. Using 3C-qPCR to demonstrate the loop that exists between these regions, it is necessary to design a control group to demonstrate the loop actually exists, so I suggest the author choose an IRF5 low expression cell line as a control to confirm the loop formation to preclude the noise signal.

4. The author performed RNA sequencing to analyze the differential genes between WT and KO cells. But it needs more analysis such as how many genes may be the direct targets of IRF5? Whether other genes are directly regulated by the rs4728142-containing region?

5. The author points out that ZBTB3 could mediate the allele-specific chromatin looping to regulate IRF5-short transcript expression at the rs4728142 locus, Could the author detect the chromatin looping after ZBTB3 knockdown? In addition, in Fig 5i and 5j, overexpression or knockdown ZBTB3 has an opposite effect on IRF5-short and IRF5-long isoform expression, this suggested that ZBTB3 mainly enhance the loop formation between rs4728142 enhancer and the promoter of IRF5 short isoform, and ZBTB3 knockdown disrupted this loop formation and enhance the loop between rs4728142 site and promoter region of IRF5 long isoform, Could the author provide experimental evidence demonstrating this loop switch?

6. In the dKO and sKO SC cells, IRF5 expression is downregulated and the expression of M1 polarization markers was also decreased, what about the expression of proinflammatory cytokine expression? It is important to investigate if over-expression of IRF5 short and long isoform within IRF5 KO cells results in a different phenotype?

7. The emergence of new CRISPR editing tools such as Prime editing provides an effective method to fulfill the base editing with high efficiency, could the author try this method to edit the rs4728142 variant, which could provide direct evidence for the allele-specific regulation of different isoforms of IRF5. I thought it would provide much stronger evidence to support the authors' conclusions

8. Could the author editing rs4728142 region in primary monocyte/macrophage, B cell , and T cell to detect the expression of different isoforms of IRF5? It is important to investigate whether this enhancer region could affect IRF5 expression in a cell type-dependent manner?

Reviewer #2 (Remarks to the Author):

Review 326835 Nature Communications:

An autoimmune pleiotropic 1 SNP modulates IRF5 alternative promoter usage through ZBTB3-mediated chromatin looping

In this study, Wang et al investigate the effects of the pleiotropic autoimmune disease variant rs4728142 on the transcriptional regulation of the key inflammation regulator gene IRF5. The authors demonstrate that the rs4728142 neighbouring genomic region displays enhancer/promoter activity in reporter gene assays; including the rs4728142 region itself in these assays strongly reduces enhancer/promoter activity. Using chromatin conformation capture assays (3C and 4C), the authors demonstrate that rs4728142 interacts with a region located between the IRF5 long promoter and the IRF5 short promoter. Interestingly, the A variant contributes more strongly to this short-range chromatin interaction than G variant in rs4728142. The authors use CRISPR-mediated activation (dCas9-VP64) and repression (dCas9-KRAB) to show that the enhancer regulates IRF5 expression. Importantly, engineered deletions in the rs4728142 region result in downregulation of the short IRF5 transcripts but upregulation of the long IRF5 transcript. The authors go on to show that a candidate transcriptional repressor protein ZBTB3 binds in vitro (EMSA) to rs4728142, with a preference for the A allele. ZBTB3 overexpression increased short IRF5 transcripts but decreased long IRF5 transcripts; CRISPR-mediated ZBTB3 ko had the opposite effect. Notably, in SC cells with deletions encompassing the rs4728142 region, macrophage induction and polarization is impaired.

I found this an interesting study that imaginatively uses a broad range of techniques to dissect the function of a pleiotropic disease variant. However, as detailed below I am not convinced that the data presented fully support the proposed mechanisms for rs4728142. Therefore, I am reluctant to recommend publications without major revisions.

Major points:

1. How similar is the chromatin landscape in monocytes and SC cells in the IRF5 locus? Comparing Figure 2a and Extended Data Figure 3a, I am not convinced the chromatin landscape surrounding rs4728142 is that similar – especially given that the H3K27ac, H3K4me1 and H3K4me3 peaks over the IRF5 gene have been ‘cut off’ in Figure 2a.

2. The description of the results in some cases lacks the necessary precision. As a result, I think the conclusions are not always supported by the data. Examples include:

Lines 195-200: “Moreover, the genomic region surrounding the pleiotropic variant rs4728142 had relatively higher chromatin accessibility in monocytes compared with other hematopoietic cells (Extended Data Fig. 2b), and the potential target gene IRF5 also showed distinct expression in monocytes according to the DICE dataset41 (Extended Data Fig. 2c), which highlights the lineage-specific role of rs4728142 in the development of autoimmune comorbidities.”

I can’t see how a minor ATAC-seq peak can be seen as evidence for “the lineage-specific role of rs4728142 in the development of autoimmune comorbidities.”

Lines 217-221: “Interestingly, rs4728142 is located on the downstream boundary of the open chromatin peak among the cell types examined, and colocalized with nucleosomes marked by H3K27ac, H3K4me1, and H3K4me3, suggesting a remarkable and unique function of this pleiotropic variant in modulating its regulatory activity and IRF5 gene expression.”

This kind of chromatin signature can be found in thousands of genomic regions. That doesn't mean this region does not have important regulatory function of course – but why it is remarkable and unique based on this chromatin signature is not clear to me at all.

3. I have several technical questions on the 3C and 4C:

3.1 The 4C lysis buffer seems to have an unusual composition? Most protocols I am familiar with use low NaCl (10 mM) – why do the authors here use essentially physiological NaCl concentrations (but higher-than-usual Tris concentration)? How does this buffer actually lyse the cells?

3.2 Modifications of HR 3C: I am skeptical as to whether modifications such as changes to the fixation time and formaldehyde concentration will really increase resolution. Can the authors back their claim up with data? At the very least, I would like to see evidence of increased restriction digest efficiency. Having said that, even if weaker crosslinking may be beneficial for the restriction digest, is this proposed advantage (if indeed it really is an advantage) perhaps offset by a suboptimal performance of other parts of the protocol? For example, I would worry about artefacts due to inefficient crosslinking. How does the % of trans interactions in HR 3C compare to traditional 3C? A very high percentage of trans interactions is usually a sign of insufficient crosslinking which increases the chances of artefact ligation products.

3.3 Are the authors using in solution ligation or in situ ligation (also referred to as nucleus ligation) in their 3C and 4C assays? This is not specified in the methods section but the high volume the ligation reaction is carried out in makes me suspect the authors have used in solution ligation, which has been shown to be inferior to in situ/in nucleus ligation (for example see Nagano et al Genome Biology 2015).

4. Figure 5f: I don't find the enrichment for ZBTB3/CTCF/RAD21 terribly convincing, especially given the y axis scale and the plateaus in the ZBTB3 and CTCF tracks over the IRF5 gene. How many ZBTB3 peaks are there genome-wide? Is the ZBTB3 enrichment at rs4728142 determined by visual inspection? Would a peak-calling algorithm confirm this?

5. Is the CRISPR knockout of ZBTB3 a true ko (Figure 5j)? There is still quite a bit of protein visible in the western blot.

6. I am not convinced the model proposed in Figure 7 is sufficiently supported by the data. First, I could not find evidence in the manuscript to support that the cohesin complex binds preferentially the A allele of rs4728142? Second, as the authors outline in the discussion, the reporter/enhancer gene results don't support this model. Third (see also above), the ZBTB3 enrichment at rs4728142 is modest at best. Fourth, I don't see direct evidence that ZBTB3 is involved in enhancer-promoter looping. This is testable – for example, is the E-P loop present in dKO or sKO SC cells? Do enhancer and promoter in the rs4728142 region interact in the absence of ZBTB3 (CRISPR ko of ZBTB3 gene)?

Minor points:

Line 134: should be 'interacting' instead of 'interacted' genes

Typo in Figure 2a: should be regulatory instead of reglatory

I think Extended Data Fig. 2b shows IRF5 expression data and Extended Data Fig. 2c shows chromatin accessibility (ATAC-seq).

Lines 195-200: "Moreover, the genomic region surrounding the pleiotropic variant rs4728142 had relatively higher chromatin accessibility in monocytes compared with other hematopoietic cells (Extended Data Fig. 2b), and the potential target gene IRF5 also showed distinct expression in monocytes according to the DICE dataset⁴¹ (Extended Data Fig. 2c), which highlights the lineage-

specific role of rs4728142 in the development of autoimmune comorbidities.”

It may be true that rs4728142 has relatively higher chromatin accessibility in monocytes, but in my view it would be fairer to say chromatin accessibility is low in monocytes and even lower in other immune cells.

Extended Data Fig. 2d: how relevant is the GM12878 data here, given that the rest of the manuscripts describes the role of rs4728142 in monocytes or SC cells (monocyte-derived cell line)?

Lines 215-217: “Consistent with the public data, the rs4728142 surrounding region in the SC cells exhibited both promoter and enhancer activities (all, $P < 0.01$) (Fig. 2a, b).”

Again – not an accurate description of what the data is showing. I think “...exhibited chromatin marks associated with both promoter and enhancer activities” is better.

Figure 2c-f: this strong dampening effect of rs4728142 on the enhancer/promoter activity of its flanking region is reminiscent of a transcriptional silencer – have the authors considered this possibility? It may be interesting especially in the context of ZBTB3 as a proposed transcriptional repressor. Related: the enrichment of active histone mods (H3K27ac, H3K4me1 and H3K4me3) is modest at best for the rs4728142 region itself – is it conceivable that this region rather limits the spreading of these active chromatin marks (see Figure 2a)? The authors refer to this region as located at the boundary of an enhancer several times in the manuscript, but I couldn’t figure out how this observation is reflected in the proposed mechanism for rs4728142.

In general, can the authors explain why they use both 3C and 4C? They are technically similar so 4C should not be used to validate results obtained with 3C, or vice versa.

Lines 269-270: “Compared with previous 3C versions^{47,48}, HR-3C utilizes a 4-bp cutting enzyme instead of the traditional 6-bp cutting enzyme...”

There are a lot (a lot!) of 3C/4C/Hi-C studies out there using 4 bp cutting enzymes such as MboI or DpnII. This is not new!

Figure 3c: does this result suggest no interactions with any other fragment than R5? This would be quite unlikely given the nature of 3C and 4C type assays. Why is this result so different from Figure 4c (I know the bait regions differ slightly, but can this be the only explanation?)

Figure 5 i and j minor point: for clarity please add ZBTB3 overexpression and ZBTB3 knockdown/knockout, respectively, to the legends next to the bar graphs (orange rectangle)

Model Figure 7 minor point: given that the IRF5 short transcripts are light blue, for clarity I would colour the IRF5 short promoter light blue too. To me, the IRF5 long promoter and IRF5 short promoter look the same colour, whereas the respective transcripts are coloured differently.

As a general note: I have the impression that the second half of the paper, including the discussion, is better stylistically and linguistically than the first.

Reviewer #3 (Remarks to the Author):

In this manuscript, the authors tried to delineate the causal pathways of how genetic non-coding variants leading to autoimmune diseases by studying the shared genetic events on the autoimmune comorbidity. A group of causal pleiotropic variants and their target genes were generated through systematic analysis. Within which, rs4728142 and its target IRF5 were top ranked. Further investigation on the underlying molecular mechanism showed that the risk allele of rs4728142-A tends to switch IRF5 alternative promoter usage mediated by ZBTB3 which is a chromatin looping structure regulator. The aberrant epigenetic and transcriptional regulation of IRF5 transcripts ultimately results in macrophage M1 polarization leading to autoimmune diseases.

The manuscript was well-written with sufficient background introduction and adequate discussion. This work provided a valuable resource for studying the function of noncoding pleiotropic variants causally associated with autoimmune diseases. Furthermore, the mechanism study on the structure of the enhancer and alternative promoter interaction determining the outcome of differentially expressed transcript isoforms of essential gene is very thorough with nice supporting data. Although ZBTB3 has already been shown to interact with rs4728142 risk allele in the prior published study, this work presented a relatively intact model and detailed mechanism explanation on the epigenetic and transcriptional regulation of IRF5 gene. Several weaknesses are identified that need to be addressed to improve the quality of the current manuscript.

1. Focusing on pleiotropic SNPs in autoimmune diseases could emphasize the biological significance and efficiently narrow down the candidate lists. However, in this study the characterized candidate and target gene regulatory mechanism were not clearly related to autoimmune comorbidity. Any supportive evidences indicating either the expression level or different isoforms of IRF5 contributes to comorbidity will improve the quality of this work.
2. As to the valuable data resources of pleiotropic SNPs in autoimmune diseases, I only found 3 candidates with similar score as rs4728142, according to the scoring system. Therefore, I'm curious about how many pleiotropic variant candidates have high confidence that they have causal regulatory roles on their targets? Which analysis have more weight contribute to the prediction? And whether this analysis could be expanded to non-pleiotropic SNP scoring?
3. What is ZBTB3 expression level and chromatin binding sites across immune cell types? ATAC-seq showed different open chromatin structure among monocyte, B cell and T cell. IRF5 showed high expression and specificity in monocytes and macrophage but not other lymphocytes might be owing to the pre-existing epigenetic traits and the presence of regulatory molecules such as ZBTB3. Furthermore, did ZBTB3 knock-down in Fig. 5j disrupt chromatin loop formation between the rs4728142 enhancer region and the short IRF5 promoter?
4. The authors compared gene expression profile of the control SC cells (rs4728142 A/G allele) to sKO or dKO cells, and concluded that the rs4728142 (A) allele's main effect is promoting M1 macrophage differentiation which drives autoimmune diseases. The evidence provided was not sufficient. To have a fair comparison, the authors need to generate SC cells with A/A and G/G alleles to compare to A/G control cells. If low efficiency of HDR is the problem, any other cell line available should be tested. The authors also need to test if the long IRF5 and the short IRF5 can influence M1 macrophage polarization in opposite ways. Will ZBTB3 over-expression and knock-down (shown in Fig. 5i and 5j) also affect SC cell's M1 polarization?

Reviewer #4 (Remarks to the Author):

The manuscript by Li and colleagues conducted systems survey on the human autoimmune disease pleiotropic loci, aiming to prioritize causal pleiotropic variants and genes. The study produced a rich set of functional annotation data and certainly inform in depth functional follow-up studies in the field. The authors then focused the top-ranked pleiotropic variant rs4728142 and proved its causal roles in regulating IRF5 isoform expression through a promoter usage mechanism. Thus, they identified previously unappreciated mechanisms within this well-known autoimmune disease-associated locus and may explain in a new angle how the risk haplotype defines precise expression of IRF5 isoforms. Overall, the work is well designed, the topic is important, and the manuscript is well written with a clear scientific logic behind it. I suggest the following points for a major revision to improve the work.

1. There should be in the figure 1 an overall workflow for systems functional annotation of autoimmune disease-associated pleiotropic loci, thus making the computational and statistical analysis pipeline sufficiently clear for the readers to understand.

2. Figure 2c-f and Extended Data Fig. 3e/f: How many times were each reporter assay repeated? There are huge difference in regulatory activity between adjacent upstream sequence of rs4728142 and the sequence plus rs4728142-containing region measured by promoter assays in 293T but not in SC cells as shown in the panels d and f. Does this mean rs4728142-containing sequence dependent, experimental reproducibility issue and/or cell-type-specific manner? Please clarify the discrepancies and perform additional experiments to support the finding?

3. In addition to the above in vitro luciferase reporter assays, can the authors count sequencing reads over rs4728142 in the ChIP-seq assays (Figure 2a)? Given that SC cells are heterozygous for rs4728142, we could expect to see allelic imbalance if the rs4728142-containing region shows a truly allele-specific enhancer activity. In parallel, allele-specific ChIP-qPCR and ChIP-Sanger sequencing could be further applied to corroborate in vitro reporter assays. In a similar vein, the analysis can be applied to the chromatin data of Extended Data Fig. 2c/d and 3a if the given cell lines are heterozygous for rs4728142. These analyses may inform us if this allele-specific enhancer activity is prevalent in vivo, and further corroborate its allele-specific interaction with the IRF5 downstream alternative promoter in the following session.

4. In Figure 3 and the relevant supporting data, the authors proved an allele-specific interaction between the rs4728142-containing region and the IRF5 downstream alternative promoter. The referee would suggest using RACE experiments to clone the full length of the short and long IRF5 transcripts despite that the authors discussed this shortcoming in the discussion. This is useful to specify the consequence of IRF5 promoter usage via the SNP region contributing to the expression of specific IRF5 isoforms. Moreover, the authors may perform similar enhancer reporter assays with the rs4728142 region targeting IRF5 alternative promoters instead of SV40 in the pGL3 construct.

5. In Figure 4d/f, why are IRF5-short or -long but not IRF5 expression so sensitive to CRISPRa, whereas all of IRF5 transcript expression are sensitive to CRISPRi for downregulation? Did the authors independently replicate the experiments? It appears that both IRF5 and its short transcript (Figure 4f/g) were downregulated upon targeted deletion of the SNP site (sKO), suggesting the site is likely to be important for the chromatin interaction. Similar enhancer reporter assays as suggested above may be applied with the SNP site deleted genomic sequences. Moreover, the authors could utilize the sKO cells (Figure 4f/g) to pinpoint whether the rs4728142 and surrounding nucleotides are essential for chromatin looping via similar 3C experiments as described above. Without defining the full length of the IRF5-short or -long, how did the authors design the transcript-specific qPCR primers? In Figure 4h, can the authors specify the IDs of IRF5-short and -long transcripts?

6. It appears that ZBTB3 is a transcriptional regulator to transform the causal effect of the rs4728142 site. The sKO cells with breakdown of ZBTB3 binding site at rs4728142 for 3C analysis may prove if ZBTB3 directly involve in chromatin interaction. Moreover, the authors could design ChIP-3C assay to further prove if ZBTB3 is directing the chromatin looping. For ZBTB3 ChIP results, Figure S6c - the authors need to explain clearly how the ChIP-qPCR results were normalized as the enrichment of CTCF and ZBTB3 is extremely high, whereas ChIP-seq signals showing in Figure 5d appears not very strong, in particular at the rs4728142 region. Additional independent replication of the assays is encouraging. Similar ChIP-qPCR assays for CTCF and ZBTB3 (Figure S6c) could be applied to the sKO cells, confirming whether the rs4728142 site matching ZBTB3 motif is essential for its chromatin association. Figure 5j: it appears that the KO is not very efficient. The authors could perform shRNA and/or siRNA against ZBTB3 knockdown assays to observe if the results are consistent with the KO?

7. An important finding of the work is to define a novel mechanism underlying the rs4728142-risk allele A containing enhancer in upregulating IRF5-short while inhibiting IRF5-long transcript. Thus, the authors could compare their functional difference, e.g. RNA-seq plus ChIP-seq of IRF5 isoforms. Current results of M1 polarization observed in rs4728142 sKO cells are virtually reflecting known biological roles of IRF5. The authors could back to check whether the relevant previous studies were indeed for the IRF5-short transcript as defined in this work.

8. Minor comments: Line 114: altering SNVs in known immune-related disease genes, such as IFIH1 – encourage to provide proper references; Lines 196-199: Extended Data Fig. 2b/c need to be changed in order.

We thank you and reviewers for evaluating our manuscript and giving us valuable comments. We have carefully addressed all the reviewers' comments below and in the manuscript accordingly. Accompanying this letter, please find the revised version of our manuscript.

Reviewer #1 (Remarks to the Author):

In this manuscript, Wang and colleagues integrated GWAS and epigenomics analysis indicating rs4728142 as a causal regulatory variant in autoimmune disease and may act in monocyte-macrophage lineage cells. They used CRISPR editing tool demonstrating the regulation of rs4728142-containing region on IRF5 expression and found rs4718142-containing region differentially regulating IRF5 long and short isoform involving disease pathogenesis. Mechanistically, they proposed rs4728142 alleles differentially bind with ZBTB3 and regulate loop formation between the rs4728142-containing region and the promoter region of IRF5 short isoform and long isoform. Their assumption is interesting, but they need more solid data to support the major conclusions.

Response: Thanks for the reviewer's excellent summary.

1. The author proposed IRF5 short isoform is involved in the development of disease. In many autoimmune diseases, IRF5 expression is abnormally up-regulated, and what about the expression change of IRF5 short isoform and long isoform in autoimmune diseases such as RA, SLE, and so on? What's the mechanism that can explain different isoforms of IRF5 have different effects on the disease process?

Response: Thanks for the constructive comments. To answer this question, we leveraged blood-derived (peripheral blood mononuclear cells) RNA-seq cohorts of paired RA/SLE patients and healthy individuals from two publications (Shchetynsky K et al. *Arthritis Res Ther.* 2017¹ and Mistry P et al. *Proc Natl Acad Sci U S A.* 2019²). Based on these independent datasets, we noticed that both *IRF5* gene (RA, $P = 0.0014$; SLA, $P = 0.0057$) and *IRF5*-short transcripts (RA, $P = 0.0044$) show higher expression in RA or SLE patients than healthy individuals, while *IRF5*-long transcripts (RA, $P = 0.082$; SLA, $P = 0.057$) only exhibit weak differences (**Fig. 1**). The evidence partially supports our major findings and indicates a likely pathogenic role of *IRF5* gene and *IRF5*-short transcripts in RA/SLE development. We have added this result into Extended Data Fig. 9h, the related description (L480–483), and figure legend (L1829–1832) in the revised manuscript.

To investigate the functional difference between the *IRF5*-short transcripts and *IRF5*-long transcripts (**Question 6**), we designed the sgRNAs targeting *IRF5*-short promoter or *IRF5*-long promoter, and used CRISPRa technology to specifically activate the *IRF5*-short transcripts or *IRF5*-long transcripts. Expectedly, sgRNA targeting *IRF5*-short promoter significantly promotes the expression of *IRF5* gene ($P < 0.0001$) and *IRF5*-short transcripts ($P < 0.0001$) without affecting the expression of *IRF5*-long transcripts, while sgRNA targeting *IRF5*-long promoter significantly promotes the expression of *IRF5* gene ($P <$

0.001) and *IRF5*-long transcripts ($P < 0.0001$) without altering the expression of *IRF5*-short transcripts (**Fig. 2a**). Next, we induced the *IRF5*-short promoter-activated or *IRF5*-long promoter-activated SC cells to M1 macrophages, and detected the macrophage M1 polarization markers (*ATF3*, *COX2*, *INDO*, *SLC7A5*, and *CCR7*) by RT-qPCR. As the result, all M1 polarization markers in both the induced-SC cells were significantly increased (all, $P < 0.01$) (**Fig. 2b**), inferring that both *IRF5*-short and *IRF5*-long transcripts could influence M1 macrophage polarization in the same way. However, it is quite difficult for us to distinguish the accurate transcript(s) which influence(s) the magnitude or condition of M1 macrophage polarization here.

Different immune cells may exert main effects on different immune diseases, and the transcripts of *IRF5* may be heterogeneously expressed in different immune cells. Among over 15 known *IRF5* transcripts, many can translate the same IRF5 full-length protein, but some transcripts may translate different forms of truncated proteins, even some cannot be translated and function as non-coding RNAs. These different forms of IRF5 protein could play different biological functions, and the immune effects on the disease process may be a result of one transcript or the interaction of multiple transcripts. Thus, the putative mechanism underlying the disease-causal effect of different *IRF5* isoforms on different autoimmune diseases is a complex and biologically significant issue which requires long-term exploration in the future.

Fig. 1 | Investigation of *IRF5* gene, *IRF5*-short, and *IRF5*-long transcripts expressions in autoimmune disease patients. a, Comparison of *IRF5* gene expression in blood RNA-seq samples between healthy individuals and RA/SLE patients (n = 23 healthy, n = 12 RA, n = 11 SLE, Mann Whitney U Test). **b**, Comparison of *IRF5*-short transcripts expression in blood RNA-seq samples between healthy individuals and RA/SLE patients (n = 23 healthy, n = 12 RA, n = 11 SLE, Mann Whitney U Test). **c**, Comparison of *IRF5*-long transcripts expression in blood RNA-seq samples between healthy individuals and RA/SLE patients (n = 23 healthy, n = 12 RA, n = 11 SLE, Mann Whitney U Test).

Fig. 2 | Both *IRF5*-short and *IRF5*-long transcripts promote M1 macrophage polarization. a, RT-qPCR for *IRF5* gene, *IRF5*-short, and *IRF5*-long transcripts in *IRF5*-short promoter-activated or *IRF5*-long promoter-activated SC cells. **b**, RT-qPCR for M1 polarization markers (*ATF3*, *COX2*, *INDO*, *SLC7A5*, and *CCR7*) in *IRF5*-short transcripts-activated or *IRF5*-long transcripts-activated induced-SC cells (n = 3). Data are represented as the means \pm standard deviations (SD), and unpaired two-tailed Student's *t*-test is used to calculate *P*-values in **a** and **b**: *, *P* < 0.05; **, *P* < 0.01; ***, *P* < 0.001; ****, *P* < 0.0001.

2. Fig.4 shows rs4728142-containing region could form gene loop with *IRF5* promoter region, and deletion of rs4728142 site decreased or increased *IRF5* short isoform expression or *IRF5* long isoform expression respectively. Whether deletion of this region can disrupt the loop formation? Or deletion of this region can enhance the loop switching from the promoter of *IRF5* short isoform to the promoter of *IRF5* long isoform?

Response: Thanks for the rigorous suggestions. According to the instructions, we performed the 3C assays on SC-sKO cells. As the result, compared with WT SC cells, the rs4728142-containing region in SC-sKO cells has a weaker interaction with R5 (between *IRF5*-long and *IRF5*-short promoters, see Fig. 3b in the revised manuscript) (*P* < 0.0001) (**Fig. 1**), indicating deletion of rs4728142-containing region can disrupt the loop formation between rs4728142-containing region and the *IRF5* downstream alternative promoter; and the rs4728142-associated enhancer has a stronger interaction with R2 (*IRF5*-long promoter core region, see Fig. 4b in the revised manuscript) (*P* < 0.001) and a weaker interaction with R5 (*IRF5*-short promoter core region, see Fig. 4b in the revised manuscript) (*P* < 0.001) (**Fig. 2**), indicating deletion of rs4728142-containing region can promote the loop switching from the promoter of *IRF5*-short isoforms to the promoter of *IRF5*-long isoforms. To further validate our results, we also performed 3C assays on the ZBTB3 knockout (ZBTB3-KO) and ZBTB3 overexpression (ZBTB3-OE) SC cells. As expected, we acquired the logically consistent results with the SC-sKO cells (**Question 5: Fig. 2–5**), which further confirms that rs4728142-containing region and ZBTB3 play a key function in the loop formation and *IRF5* promoter selection. We have added these results into Extended Data Fig. 10a and b, the related description (L443–464), and figure legend (L1836–1852) in the revised manuscript.

Fig. 1 | The 3C results between rs4728142-containing region and the *IRF5* downstream alternative promoter in SC and SC-sKO cells. Enrichment quantification of 3C assay at the rs4728142-containing region normalized to control in SC and SC-sKO cells ($n = 3$). **a**, Electrophoresis results of 3C assay for the rs4728142-containing region in SC and SC-sKO cells. **b**, Relative interaction frequencies between the rs4728142-containing region and representative *DpnII* cutting fragments indicated in Fig. 3b in the revised manuscript. Data are represented as the means \pm standard deviations (SD), and unpaired two-tailed Student's *t*-test is used to calculate *P*-values in **b**: *, $P < 0.05$; **, $P < 0.01$; ***, $P < 0.001$; ****, $P < 0.0001$.

Fig. 2 | The 3C results between rs4728142-associated enhancer and the *IRF5* promoter in SC and SC-sKO cells. Enrichment quantification of 3C assay at the rs4728142-associated enhancer normalized to control in SC and SC-sKO cells (n = 3). **a**, Electrophoresis results of 3C assay for the rs4728142-containing region in SC and SC-sKO cells. **b**, Relative interaction frequencies between the rs4728142-associated enhancer and representative *DpnII* cutting fragments indicated in Fig. 4b in the revised manuscript. Data are represented as the means \pm standard deviations (SD), and unpaired two-tailed Student's *t*-test is used to calculate *P*-values in **b**: *, *P* < 0.05; **, *P* < 0.01; ***, *P* < 0.001; ****, *P* < 0.0001.

3. The distance between rs4728142 and IRF5 promoter is very short, especially for the promoter of IRF5 long isoform. Using 3C-qPCR to demonstrate the loop that exists between these regions, It is necessary to design a control group to demonstrate the loop actually exists, so I suggest the author choose an IRF5 low expression cell line as a control to confirm the loop formation to preclude the noise signal.

Response: Thanks for your rigorous reminding and suggestions. After careful proofreading, we found that the 3C result of rs4728142-containing region in the first manuscript is indeed abnormal, this result may be caused by improper technical operation and lacks of control. To eliminate the influence of noise signal, we used 293T cells (*IRF5* hardly expresses) as a control, and performed the 3C assays on 293T and SC cells. As the result, compared with 293T cells, rs4728142-containing region in SC cells has a stronger interaction with R5 (between *IRF5*-long and *IRF5*-short promoters, see Fig. 3b in the revised manuscript) (*P* < 0.0001) (**Fig. 1**), and the rs4728142-associated enhancer has a stronger interaction with R2 (*IRF5*-long promoter core region, see Fig. 4b in the revised manuscript) (*P* < 0.001) and R5 (*IRF5*-short promoter core region, see Fig. 4b in the revised manuscript) (*P* < 0.0001) (**Fig. 2**). **We have updated this result (Fig. 1b) into Fig. 3c and Extended Data Fig. 5c in the revised manuscript.** In addition, we also performed 3C assays on other cell lines (sKO, ZBTB3-KO, and ZBTB3-OE SC cells) and made the SC cells as a control, we obtained the reliable experimental results consistent with our mechanism model (as answered in **Question 2: Fig. 1 and Fig. 2**, and **Question 5: Fig. 2–5**).

Fig. 1 | The 3C results between rs4728142-containing region and the *IRF5* downstream alternative promoter in SC and 293T cells. Enrichment quantification of 3C assay at the rs4728142-containing region normalized to control in SC and 293T cells ($n = 3$). **a**, Electrophoresis results of 3C assay for the rs4728142-containing region in SC and 293T cells. **b**, Relative interaction frequencies between the rs4728142-containing region and representative *DpnII* cutting fragments indicated in Fig. 3b in the revised manuscript. Data are represented as the means \pm standard deviations (SD), and unpaired two-tailed Student's *t*-test is used to calculate *P*-values in **b**: *, $P < 0.05$; **, $P < 0.01$; ***, $P < 0.001$; ****, $P < 0.0001$.

Fig. 2 | The 3C results between rs4728142-associated enhancer and the *IRF5* promoter in SC and 293T cells. Enrichment quantification of 3C assay at the rs4728142-associated enhancer normalized to control in SC and 293T cells (n = 3). **a**, Electrophoresis results of 3C assay for the rs4728142-containing region in SC and 293T cells. **b**, Relative interaction frequencies between the rs4728142-associated enhancer and representative *DpnII* cutting fragments indicated in Fig. 4b in the revised manuscript. Data are represented as the means \pm standard deviations (SD), and unpaired two-tailed Student's *t*-test is used to calculate *P*-values in **b**: *, *P* < 0.05; **, *P* < 0.01; ***, *P* < 0.001; ****, *P* < 0.0001.

4. The author performed RNA sequencing to analyze the differential genes between WT and KO cells. But it needs more analysis such as how many genes may be the direct targets of *IRF5*? Whether other genes are directly regulated by the rs4728142-containing region?

Response: Thanks for the comments. To analyze candidate target genes regulated by *IRF5* in our research context, we leveraged a high quality *IRF5* ChIP-seq peaks in human peripheral blood mononuclear cells from SLE patients³ and defined genes proximal of *IRF5*-binding sites as candidate *IRF5* target genes. By intersecting the candidate *IRF5* target genes with the differentially expressed genes in our sKO cells versus WT cells, we found 80 *IRF5* downstream genes affected via rs4728142-associated enhancer. Gene ontology (GO) enrichment analysis revealed that these genes are significantly enriched at several critical immune functions aligned to our expectation, including positive regulation of interleukin-2 (IL-2) production, myeloid cell homeostasis, and neutrophil homeostasis (Fig. 1).

In addition, to inspect other directly regulated genes (instead of *IRF5*) of rs4728142-containing region, we applied a non-parametric peak calling algorithm peakC⁴ based on our 4C-seq anchored at the rs4728142-associated enhancer. We found this enhancer could interact with *PRRT4*, *IMPDH1*, and *NRF1* genes within 2Mb genomic region. Among these genes, *PRRT4* is differentially expressed in sKO cells versus WT cells (\log_2 Fold = -1.48, adjusted *P* = 0.0006), and this gene is rarely studied and is documented to be associated with Myalgic Encephalomyelitis/Chronic Fatigue Syndrome⁵.

Fig. 1 | GO enrichment analysis of *IRF5* downstream target genes affected via rs4728142-associated enhancer.

5. The author points out that ZBTB3 could mediate the allele-specific chromatin looping to regulate *IRF5*-short transcript expression at the rs4728142 locus, Could the author detect the chromatin looping after ZBTB3 knockdown? In addition, in Fig 5i and 5j, overexpression or knockdown ZBTB3 has an opposite effect on *IRF5*-short and *IRF5*-long isoform expression, this suggested that ZBTB3 mainly enhance the loop formation between rs4728142 enhancer and the promoter of *IRF5* short isoform, and ZBTB3 knockdown disrupted this loop formation and enhance the loop between rs4728142 site and promoter region of *IRF5* long isoform, Could the author provide experimental evidence demonstrating this loop switch?

Response: Thanks for your constructive comments. Our previous data shows that the effect of ZBTB3 knockout (KO) is not completely. To obtain a complete ZBTB3-knock out (KO) cell line, we subcloned the ZBTB3-KO cell line we previously acquired, and then performed western blotting for ZBTB3 expression and RT-qPCR for the *IRF5* gene, *IRF5*-short transcripts, and *IRF5*-long transcripts. Western blotting results showed that ZBTB3 was almost completely knocked out (**Fig. 1, top**). RT-qPCR results showed that *IRF5*-short transcripts were significantly down-regulated ($P < 0.01$), *IRF5*-long transcripts were significantly up-regulated ($P < 0.01$), and *IRF5* gene were significantly down-regulated after ZBTB3 knockout ($P < 0.01$) (**Fig. 1, bottom**). **We have updated our new result into Fig. 5i in the revised manuscript.** Next, we performed 3C assays on the rs4728142-sKO, ZBTB3-KO, and ZBTB3-OE SC cells. As answered above (**Question 2: Fig. 1 and Fig. 2**), rs4728142-sKO significantly weakened the interaction between rs4728142-containing region and R5 (between *IRF5*-long and *IRF5*-short promoters, see Fig. 3b in the revised manuscript) ($P < 0.0001$), and switched the rs4728142-associated enhancer from R6 (*IRF5*-short promoter core region, see Fig. 4b in the revised manuscript) ($P < 0.001$) to R2 (*IRF5*-long promoter core region, see Fig. 4b in the revised manuscript) ($P < 0.001$). **We have added these results into Extended Data Fig. 10a and b, the related description (L443–464) and figure legend (L1836–1852) in the revised manuscript.**

In contrast, ZBTB3-KO significantly weakened the interaction between rs4728142 and R5 (between *IRF5*-long and *IRF5*-short promoters, see Fig. 3b in the revised manuscript) ($P < 0.0001$) (**Fig. 2**), and switched the rs4728142-associated enhancer from R6 (*IRF5*-short promoter core region, see Fig. 4b in the revised manuscript) ($P < 0.01$) to R2 (*IRF5*-long promoter core region, see Fig. 4b in the revised manuscript) ($P < 0.01$) (**Fig. 3**). On the contrary, ZBTB3-OE significantly strengthened the interaction between rs4728142 and R5 (between *IRF5*-long and *IRF5*-short promoters, see Fig. 3b in the revised manuscript) ($P < 0.01$) (**Fig. 4**), and switched the rs4728142-associated enhancer from R2 (*IRF5*-long promoter core region, see Fig. 4b in the revised manuscript) ($P < 0.05$) to R6 (*IRF5*-short promoter core region, see Fig. 4b in the revised manuscript) ($P < 0.01$) (**Fig. 5**). **We have added these results into Extended Data Fig. 10c–f, the related description (L443–464), and figure legend (L1836–1852) in the revised manuscript.** The above data fully confirmed that ZBTB3 mediates the loop formation and *IRF5* promoter selection.

Fig. 1 | ZBTB3 mediates expression switching of the *IRF5*-short and *IRF5*-long transcripts in SC cells. Western blotting for validating the ZBTB3 knockout (KO) (top), and RT-qPCR for detecting the *IRF5* transcript expression in ZBTB3-KO SC cells compared with the unedited SC cells (control) (n = 3) (bottom). Data are represented as the means \pm standard deviations (SD), and unpaired two-tailed Student's *t*-test is used to calculate *P*-values: *, *P* < 0.05; **, *P* < 0.01; ***, *P* < 0.001; ****, *P* < 0.0001.

Fig. 2 | The 3C results between rs4728142-containing region and the *IRF5* downstream alternative promoter in SC and ZBTB3-KO SC cells. Enrichment quantification of 3C assay at the rs4728142-containing region normalized to control in SC and ZBTB3-KO SC cells (n = 3). **a**, Electrophoresis results of 3C assay for the rs4728142-containing region in SC and ZBTB3-KO SC cells. **b**, Relative interaction frequencies between the rs4728142-containing region and representative *DpnII* cutting fragments indicated in Fig. 3b in the revised manuscript. Data are represented as the means \pm standard deviations (SD), and unpaired two-tailed Student's *t*-test is used to calculate *P*-values in **b**: *, *P* < 0.05; **, *P* < 0.01; ***, *P* < 0.001; ****, *P* < 0.0001.

Fig. 3 | The 3C results between rs4728142-associated enhancer and the *IRF5* promoter in SC and ZBTB3-KO SC cells. Enrichment quantification of 3C assay at the rs4728142-associated enhancer normalized to control in SC and ZBTB3-KO SC cells (n = 3). **a**, Electrophoresis results of 3C assay for the rs4728142-associated enhancer in SC and ZBTB3-KO SC cells. **b**, Relative interaction frequencies between the rs4728142-associated enhancer and representative *DpnII* cutting fragments indicated in Fig. 3b in the revised manuscript. Data are represented as the means \pm standard deviations (SD), and unpaired two-tailed Student's *t*-test is used to calculate *P*-values in **b**: *, *P* < 0.05; **, *P* < 0.01; ***, *P* < 0.001; ****, *P* < 0.0001.

Fig. 4 | The 3C results between rs4728142-containing region and the *IRF5* downstream alternative promoter in SC and ZBTB3-OE SC cells. Enrichment quantification of 3C assay at the rs4728142-containing region normalized to control in SC and ZBTB3-OE SC cells (n = 3). **a**, Electrophoresis results of 3C assay for the rs4728142-containing region in SC and ZBTB3-OE SC cells. **b**, Relative interaction frequencies between the rs4728142-containing region and representative *DpnII* cutting fragments indicated in Fig. 3b in the revised manuscript. Data are represented as the means \pm standard deviations (SD), and unpaired two-tailed Student's *t*-test is used to calculate *P*-values in **b**: *, *P* < 0.05; **, *P* < 0.01; ***, *P* < 0.001; ****, *P* < 0.0001.

Fig. 5 | The 3C results between rs4728142-associated enhancer and the *IRF5* promoter in SC and ZBTB3-OE SC cells. Enrichment quantification of 3C assay at the rs4728142-associated enhancer normalized to control in SC and ZBTB3-OE SC cells (n = 3). **a**, Electrophoresis results of 3C assay for the rs4728142-associated enhancer in SC and ZBTB3-OE SC cells. **b**, Relative interaction frequencies between the rs4728142-associated enhancer and representative *DpnII* cutting fragments indicated in Fig. 3b in the revised manuscript. Data are represented as the means \pm standard deviations (SD), and unpaired two-tailed Student's *t*-test is used to calculate *P*-values in **b**: *, *P* < 0.05; **, *P* < 0.01; ***, *P* < 0.001; ****, *P* < 0.0001.

6. In the dKO and sKO SC cells, *IRF5* expression is downregulated and the expression of M1 polarization markers was also decreased, what about the expression of proinflammatory cytokine expression? It is important to investigate if over-expression of *IRF5* short and long isoform within *IRF5* KO cells results in a different phenotype?

Response: Thanks for the good questions. To explore if rs4728142 knockout can affect the expression of proinflammatory cytokines, we used RT-qPCR to detect the key M1 proinflammatory cytokines^{6,7} in dKO and sKO SC cells. As the result, compared with control (SC-Cas9) cells, the proinflammatory cytokines (*IL-1 β* , *IL-6*, *IL-8*, and *TNF- α*) in

dKO and sKO SC cells were all significantly down-regulated (all, $P < 0.05$) (**Fig. 1**), inferring that rs4728142 could affect the expression of proinflammatory cytokines through regulating *IRF5* expression, and then modulate the M1 macrophage polarization.

As mentioned above (**Question 1**), it is quite hard to identify which transcript(s) result(s) in M1 macrophage polarization, so we cannot specifically over-express the exact *IRF5*-short or *IRF5*-long isoform(s) within *IRF5*-KO cells. To investigate the functional difference between the *IRF5*-short and *IRF5*-long transcripts, we designed the sgRNAs targeting *IRF5*-short or *IRF5*-long promoter, and used CRISPRa technology to specifically activate the *IRF5*-short or *IRF5*-long transcripts. Expectedly, sgRNA targeting *IRF5*-short promoter significantly promotes the expression of *IRF5* gene ($P < 0.0001$) and *IRF5*-short transcripts ($P < 0.0001$) without affecting the expression of *IRF5*-long transcripts, while sgRNA targeting *IRF5*-long promoter significantly promotes the expression of *IRF5* gene ($P < 0.001$) and *IRF5*-long transcripts ($P < 0.0001$) without altering the expression of *IRF5*-short transcripts (**Fig. 2a**). Next, we induced the *IRF5*-short or *IRF5*-long promoter-activated SC cells to M1 macrophages, and detected the macrophage M1 polarization markers (*ATF3*, *COX2*, *INDO*, *SLC7A5*, and *CCR7*) by RT-qPCR. As the result, all M1 polarization markers in both the induced-SC cells were significantly increased (all, $P < 0.01$) (**Fig. 2b**), inferring that both *IRF5*-short and *IRF5*-long transcripts could influence M1 macrophage polarization in the same way. However, as explained on **Question 1**, it is quite difficult for us to distinguish the accurate transcript(s) which influence(s) the magnitude or condition of M1 macrophage polarization here. The putative mechanism underlying the disease-causal effect of different *IRF5* isoforms on different autoimmune diseases is a complex and biologically significant issue which requires long-term exploration in the future.

Fig. 1 | rs4728142 modulates the key M1 proinflammatory cytokine expression. Expression of the key M1 proinflammatory cytokines (*IL-1β*, *IL-6*, *IL-8*, and *TNFα*) in the rs4728142-dKO (n = 3) (a) or rs4728142-sKO (n = 3) (b) induced-cells compared with the unedited SC-Cas9 induced-cells (GA, control) as determined by RT-qPCR. Data are represented as the means \pm standard deviations (SD), and unpaired two-tailed Student's *t*-test is used to calculate *P*-values in a and b: *, $P < 0.05$; **, $P < 0.01$; ***, $P < 0.001$; ****, $P < 0.0001$.

Fig. 2 | *IRF5*-short and *IRF5*-long transcripts promote M1 macrophage polarization. **a**, RT-qPCR for *IRF5* gene, *IRF5*-short, and *IRF5*-long transcripts in *IRF5*-short promoter-activated or *IRF5*-long promoter-activated SC cells. **b**, RT-qPCR for M1 polarization markers (*ATF3*, *COX2*, *INDO*, *SLC7A5*, and *CCR7*) in *IRF5*-short transcripts-activated or *IRF5*-long transcripts-activated induced-SC cells (n = 3). Data are represented as the means \pm standard deviations (SD), and unpaired two-tailed Student's *t*-test is used to calculate *P*-values in **a** and **b**: *, *P* < 0.05; **, *P* < 0.01; ***, *P* < 0.001; ****, *P* < 0.0001.

7. The emergence of new CRISPR editing tools such as Prime editing provides an effective method to fulfill the base editing with high efficiency, could the author try this method to edit the rs4728142 variant, which could provide direct evidence for the allele-specific regulation of different isoforms of *IRF5*. I thought it would provide much stronger evidence to support the authors' conclusions.

Response: Thanks for the constructive comments. According to the suggestion, we acquired the rs4728142 with homozygous GG (rs4728142-GG) and homozygous AA (rs4728142-AA) SC cells by Cas9-initiated homology-directed repair (HDR) with extensive efforts (**Fig. 1a**). We used RT-qPCR to detect the different isoforms of *IRF5*. Expectedly, compared with the rs4728142-GG SC cells, the rs4728142-AA SC cells increased the expression of *IRF5* gene and *IRF5*-short transcripts, and decreased the expression of *IRF5*-long transcripts, but had no effect on the nearest gene, *TNPO3* (**Fig. 1b**). Next, we induced the rs4728142-GG and rs4728142-AA SC cells to M1 macrophages, and detected the macrophage M1 polarization markers (*ATF3*, *COX2*, *INDO*, *SLC7A5*, and *CCR7*) by RT-qPCR. Compared with the rs4728142-GG induced-SC cells, all M1 polarization markers in the rs4728142-AA induced-SC cells were significantly increased (**Fig. 1c**), indicating that rs4728142 could modulate the macrophage M1 polarization. In addition, we detected the expression of key M1 proinflammatory cytokines^{6,8}. As the result, compared with the rs4728142-GG induced-SC cells, the expression of *IL-1 β* , *IL-6*, *IL-8*, and *TNF α* in the rs4728142-AA induced-SC cells were significantly increased (all, *P* < 0.05) (**Fig. 1d**). Together, the above data suggest that the allele A of rs4728142 could promote proinflammatory cytokine expression through up-regulating the expression of *IRF5*-short transcripts and *IRF5* gene, and down-regulating the expression of the *IRF5*-long transcripts, then contributes to the M1 macrophage polarization. **We have added these results into Fig. 7a–d, the related description (L503–521), and the attachment information (Supplementary Table 14) in the revised manuscript.**

Fig. 1 | Base editing at rs4728142 in SC cells and the related phenotype detection. a, the strategy of sing-base editing by CRISPR knock-in technology. The sKO sgRNA and single-stranded donor oligonucleotides (ssODN) with major allele-G or minor allele-A were co-transfected into the single cell-derived SC-Cas9 cells, and single cells were sorted for identification by Sanger sequencing. The green multilateral shape denotes Cas9 protein, and the red triangle denotes the cutting site. **b,** RT-qPCR for different *IRF5* transcripts in rs4728142-AA SC cells compared with rs4728142-GG SC cells (n = 3). **c,** RT-qPCR for M1 polarization markers (*ATF3*, *COX2*, *INDO*, *SLC7A5*, and *CCR7*) in rs4728142-AA SC cells compared with rs4728142-GG SC cells (n = 3). **d,** RT-qPCR for key proinflammatory cytokines (*IL-1 β* , *IL-6*, *IL-8*, and *TNF α*) in rs4728142-AA induced-SC cells compared with rs4728142-GG induced-SC cells (n = 3). Data are represented as the means \pm standard deviations (SD), and unpaired two-tailed Student's *t*-test is used to calculate *P*-values in **b–d**: *, *P* < 0.05; **, *P* < 0.01; ***, *P* < 0.001; ****, *P* < 0.0001.

8. Could the author editing rs4728142 region in primary monocyte/macrophage, B cell, and T cell to detect the expression of different isoforms of *IRF5*? It is important to investigate whether this enhancer region could affect *IRF5* expression in a cell type-dependent manner?

Response: Thanks for the comments. According to your suggestions, we originally planned to purchase Primary Peripheral Blood CD14⁺ Monocytes (ATCC, PCS-800-010), Primary B cells (ATCC, PCS-800-018), Primary CD4⁺ Helper T cells (ATCC, PCS-800-016), and Primary CD8⁺ Cytotoxic T Cells (ATCC, PCS-800-017), but due to the prevalence of the epidemic, during a long time, we were unfortunately told that we could not order them. Then, we isolated the primary monocyte, B, and T cells of human blood with commercial kits, and tried to edit them. However, after a large number of attempts including increasing serum concentration and slowing down nucleofection voltage, we have never obtained successfully edited primary cells due to the primary single cells are hard to grow up. We have seriously considered the suggestions and believe that it can

better illustrate the significance of our research using diverse primary immune cells in the future.

Alternatively, by inspecting the cell type-specific *IRF5* expression patterns in blood tissue from healthy subjects (Schmiedel BJ et al. *Cell*. 2018⁹) and autoimmune disease patients (Ota M et al. *Cell*. 2021¹⁰). We found that the gene expression level of *IRF5* is highest among Monocytes compared with other immune cells (**Fig. 1**), suggesting relatively essential functions of *IRF5* in Monocytes. Besides, after querying on QTLbase¹¹, we noticed that rs4728142 obtains the largest effect size being eQTLs in Monocytes than other tissue/cell types (**Fig. 2**). **We have added Fig. 1b and Fig. 2 into Extended Data Fig. 2d and e, and the related description (L200–203) in the revised manuscript.**

Fig. 1 | Comparison of *IRF5* gene expression levels among different immune cell types in two cohorts. a, *IRF5* expression in the DICE project, including 13 immune cell types isolated from 106 leukapheresis samples provided by 91 healthy subjects. b, *IRF5* expression in the ImmuNexUT project, including 28 distinct immune cell subsets from 337 patients diagnosed with 10 categories of immune-mediated diseases and 79 healthy volunteers.

All eQTL data (+/- 10M region)

Total: 256 entries, showing 1 to 8

Trait	Tissue	Effective Allele	P-Value	Effect Size	SE	FDR	Dataset
IRF5	Blood-Monocyte	-	4.760e-31	-12.9058	-	4.910e-28	24604202_EUR_2014
IRF5	Blood-Monocyte	-	3.400e-28	-11.9807	-	2.850e-25	24604202_EUR_2014
IRF5	Blood-Monocyte	-	8.260e-24	-11.0534	-	7.470e-21	24604202_EUR_2014
IRF5	Blood-Monocyte	-	2.910e-21	-10.5366	-	3.830e-18	24604202_EUR_2014
IRF5	Blood-T cell CD4+	A	2.910e-9	-5.9365	0.9692	-	28248954_EUR_2017
IRF5	Blood-T cell CD8+	A	4.190e-5	-4.0968	0.9843	0.3220	28248954_EUR_2017
IRF5	Blood	-	1.060e-21	-0.7002	0.0732	-	25951796_EUR_2015
IRF5	Blood	A	1.700e-133	-0.6845	0.0278	-	28065468_MIX_2017
IRF5	Skin	-	1.810e-8	-0.6639	0.1179	-	25951796_EUR_2015
IRF5	Blood	-	6.950e-16	-0.5912	0.0682	1.860e-12	26917434_EUR_2016
IRF5	Brain-Cerebellum	A	5.210e-13	-0.5448	0.0699	1.050e-8	22685416_EUR_2012
IRF5	Lymphocyte	G	2.590e-27	-0.5287	0.0488	-	25951796_EUR_2015

Fig. 2 | Effect size comparison of eQTL (rs4728142-*IRF5*) among different tissue/cell types in QTLbase.

Reviewer #2 (Remarks to the Author):

Review 326835 Nature Communications:

An autoimmune pleiotropic 1 SNP modulates IRF5 alternative promoter usage through ZBTB3-mediated chromatin looping

In this study, Wang et al investigate the effects of the pleiotropic autoimmune disease variant rs4728142 on the transcriptional regulation of the key inflammation regulator gene IRF5. The authors demonstrate that the rs4728142 neighbouring genomic region displays enhancer/promoter activity in reporter gene assays; including the rs4728142 region itself in these assays strongly reduces enhancer/promoter activity. Using chromatin conformation capture assays (3C and 4C), the authors demonstrate that rs4728142 interacts with a region located between the IRF5 long promoter and the IRF5 short promoter. Interestingly, the A variant contributes more strongly to this short-range chromatin interaction than G variant in rs4728142. The authors use CRISPR-mediated activation (dCas9-VP64) and repression (dCas9-KRAB) to show that the enhancer regulates IRF5 expression. Importantly, engineered deletions in the rs4728142 region result in downregulation of the short IRF5 transcripts but upregulation of the long IRF5 transcript. The authors go on to show that a candidate transcriptional repressor protein ZBTB3 binds in vitro (EMSA) to rs4728142, with a preference for the A allele. ZBTB3 overexpression increased short IRF5 transcripts but decreased long IRF5 transcripts; CRISPR-mediated ZBTB3 ko had the opposite effect. Notably, in SC cells with deletions encompassing the rs4728142 region, macrophage induction and polarization is impaired.

I found this an interesting study that imaginatively uses a broad range of techniques to dissect the function of a pleiotropic disease variant. However, as detailed below I am not convinced that the data presented fully support the proposed mechanisms for rs4728142. Therefore, I am reluctant to recommend publications without major revisions.

Response: Thanks for the reviewer's commendation together with the revision opportunity.

Major points:

1. How similar is the chromatin landscape in monocytes and SC cells in the IRF5 locus? Comparing Figure 2a and Extended Data Figure 3a, I am not convinced the chromatin landscape surrounding rs4728142 is that similar – especially given that the H3K27ac, H3K4me1 and H3K4me3 peaks over the IRF5 gene have been 'cut off' in Figure 2a.

Response: Thanks for the raised concern and sorry for the confused illustration in our main text. Yes, the magnitude of histone modifications and open chromatin signals differ between ENCODE Monocytes-CD14⁺ RO01746 cells and monocyte-derived SC cells (ATCC, CRL-9855), especially for H3K4me1. This could be attribute to the difference and cell line-based specificity of chromatin and transcription states at *IRF5* enhancer and promoter. To emphasize the histone modifications and open chromatin signals around rs4728142-containing region in our investigated SC cells, we have to present Fig. 2a in the current form, wherein the H3K27ac, H3K4me1, and H3K4me3 peaks over the *IRF5* gene

have been 'cut off'. To mitigate the confusion, we polished the sentence in our revised manuscript.

Besides, the key reason we previously used monocyte-derived SC cells (ATCC, CRL-9855) is that the genotype of the SC cells is heterozygous at rs4728142 and relatively easy for genome editing. During the revision, we isolated the primary monocyte of human blood with commercial kits and tried to edit them using CRISPR. Unfortunately, after many attempts, we have never obtained successfully edited primary cells due to the primary single cells are difficult to grow. Albeit some differences between SC cell lines and primary monocytes, the analysis results based on our extensive bioinformatics analysis on primary cells can be verified in SC cells, so we expect that primary monocytes may have a more significant phenotype and mechanism consistent with that of SC cells.

2. The description of the results in some cases lacks the necessary precision. As a result, I think the conclusions are not always supported by the data. Examples include:

Lines 195-200: "Moreover, the genomic region surrounding the pleiotropic variant rs4728142 had relatively higher chromatin accessibility in monocytes compared with other hematopoietic cells (Extended Data Fig. 2b), and the potential target gene *IRF5* also showed distinct expression in monocytes according to the DICE dataset⁴¹ (Extended Data Fig. 2c), which highlights the lineage-specific role of rs4728142 in the development of autoimmune comorbidities."

I can't see how a minor ATAC-seq peak can be seen as evidence for "the lineage-specific role of rs4728142 in the development of autoimmune comorbidities."

Response: Sorry for the overstated words. **We have polished the sentence (L195-197) in our revised manuscript.** Besides, we also inspected cell type-specific *IRF5* expression patterns in blood tissue from the ImmuNexUT project (Ota M et al. *Cell*. 2021¹⁰), which includes 28 distinct immune cell subsets from 337 patients diagnosed with 10 categories of immune-mediated diseases and 79 healthy volunteers. Together with the previous data, we found that the gene expression level of *IRF5* is highest among Monocytes compared with other immune cells, suggesting relatively essential functions of *IRF5* in Monocytes (**Fig. 1**). Besides, after querying on QTLbase¹¹, we noticed that rs4728142 obtains the largest effect size being eQTLs in Monocytes than other tissue/cell types (**Fig. 2**). **We have added Fig. 1b and Fig. 2 into Extended Data Fig. 2d and e, and the related description (L200–203) in the revised manuscript.**

Fig. 1 | Comparison of *IRF5* gene expression levels among different immune cell types in two cohorts. a, *IRF5* expression in the DICE project, including 13 immune cell types isolated from 106 leukapheresis samples provided by 91 healthy subjects. **b**, *IRF5* expression in the ImmuNexUT project, including 28 distinct immune cell subsets from 337 patients diagnosed with 10 categories of immune-mediated diseases and 79 healthy volunteers.

All eQTL data (+/- 10M region) Type keywords to filter Show All Data (256)

Total: 256 entries, showing 1 to 8 << < 1 2 >>

Trait#	Tissue#	Effective Allele	P-Value#	Effect Size*	SE#	FDR#	Dataset#
IRF5 ①	Blood-Monocyte	-	4.760e-31	-12.9058 ①	-	4.910e-28	24604202_EUR_2014
IRF5 ①	Blood-Monocyte	-	3.400e-28	-11.9807 ①	-	2.850e-25	24604202_EUR_2014
IRF5 ①	Blood-Monocyte	-	8.260e-24	-11.0534 ①	-	7.470e-21	24604202_EUR_2014
IRF5 ①	Blood-Monocyte	-	2.900e-21	-10.5366 ①	-	3.830e-18	24604202_EUR_2014
IRF5 ①	Blood-T cell CD4+	A	2.910e-9	-5.9365 ①	0.9692 ①	-	28248954_EUR_2017
IRF5 ①	Blood-T cell CD8+	A	4.190e-5	-4.0968 ①	0.9843 ①	0.3220	28248954_EUR_2017
IRF5 ①	Blood	-	1.060e-21	-0.7002 ①	0.0732 ①	-	25951796_EUR_2015
IRF5 ①	Blood	A	1.700e-133	-0.6845 ①	0.0278 ①	-	28065468_MIX_2017
IRF5 ①	Skin	-	1.810e-8	-0.6639 ①	0.1179 ①	-	25951796_EUR_2015
IRF5 ①	Blood	-	6.950e-16	-0.5912 ①	0.0682 ①	1.860e-12	26917434_EUR_2016
IRF5 ①	Brain-Cerebellum	A	5.210e-13	-0.5448 ①	0.0699 ①	1.050e-8	22685416_EUR_2012
IRF5 ①	Lymphocyte	G	2.590e-27	-0.5287 ①	0.0488 ①	-	25951796_EUR_2015

Fig. 2 | Effect size comparison of eQTL (rs4728142-*IRF5*) among different tissue/cell types in QTLbase.

Lines 217-221: “Interestingly, rs4728142 is located on the downstream boundary of the open chromatin peak among the cell types examined, and colocalized with nucleosomes marked by H3K27ac, H3K4me1, and H3K4me3, suggesting a remarkable and unique function of this pleiotropic variant in modulating its regulatory activity and *IRF5* gene expression.”

This kind of chromatin signature can be found in thousands of genomic regions. That doesn’t mean this region does not have important regulatory function of course – but why it is remarkable and unique based on this chromatin signature is not clear to me at all.

Response: Thanks for the valuable suggestions. We originally wanted to emphasize that the rs4728142-containing region could be one associated with nucleosome-bound protein, but this statement is not rigorous. We have changed it to “Interestingly, rs4728142 is located on the downstream boundary of the open chromatin peak among the cell types

examined, and colocalized with nucleosomes marked by H3K27ac, H3K4me1, and H3K4me3, suggesting that this pleiotropic variant located in active chromatin could associate with nucleosome-bound protein in modulating its regulatory activity and *IRF5* gene expression.” in the revised manuscript.

3. I have several technical questions on the 3C and 4C:

3.1 The 4C lysis buffer seems to have an unusual composition? Most protocols I am familiar with use low NaCl (10 mM) – why do the authors here use essentially physiological NaCl concentrations (but higher-than-usual Tris concentration)? How does this buffer actually lyse the cells?

Response: As the reviewer mentioned, some 3C-based protocols use low NaCl (10 mM)^{12,13}. In this study, we performed 3C and 4C assays mainly according to the 4C Technology: protocol and data analysis¹⁴. In fact, many other articles¹⁵⁻¹⁸ published by Professor Wouter de Laat who first applied 3C and 4C technologies on eukaryotic cells, have also reported the usage of this lysis buffer (50 mM Tris-HCl pH 7.5, 150 mM NaCl, 5mM EDTA, 0.5% NP-40, 1% TX-100, and protease inhibitors). We believe that the 4C lysis buffer may be the result of a series of explorations by the technology developers. In practice, to acquire the completely lysed SC cells, we used the lysis buffer to lyse SC cells on ice for more than 15 min (usually 30 min), and then determined the lysis of cells after methylgreen-pyronin staining: mix 3 µl of cells with 3 µl of methylgreen-pyronin staining on a microscope slide and cover with a coverslip, view staining under a microscope, and cytoplasm stains pink and the nuclei/DNA stains blue/green. We carried out the next step until the complete lysis of SC cells was confirmed. We can't find out the exact cell lysis principle of this buffer. In our opinion, physiological NaCl concentrations may be conducive to the lytic reaction under physiological state which may reduce the influence of lysis proceeding on three-dimensional (3D) structure of the genome to a certain extent, and mild lysate (0.5% NP40 and 1% Triton X-100) may be the main component of the lysis buffer. We appreciated the reviewer for the opportunity to clarify used protocols of our 3C and 4C assays.

3.2 Modifications of HR 3C: I am skeptical as to whether modifications such as changes to the fixation time and formaldehyde concentration will really increase resolution. Can the authors back their claim up with data? At the very least, I would like to see evidence of increased restriction digest efficiency. Having said that, even if weaker crosslinking may be beneficial for the restriction digest, is this proposed advantage (if indeed it really is an advantage) perhaps offset by a suboptimal performance of other parts of the protocol? For example, I would worry about artefacts due to inefficient crosslinking. How does the % of trans interactions in HR 3C compare to traditional 3C? A very high percentage of trans interactions is usually a sign of insufficient crosslinking which increases the chances of artefact ligation products.

Response: We thank the reviewer for the constructive comments, which will help us optimize related experiments and results. Previous study¹⁵ have reported in detail that changes to the fixation time and formaldehyde concentration could change the production of noncut fragment, which may affect the resolution (**Fig. 1, borrowed from the original**

paper). Consistent with above findings, our previous results found that moderate reduction of the fixation time and the formaldehyde concentration could increase the efficiency of enzyme digestion: actually, we made a series of explorations before the experiment began, we found that the cross-linked SC cells was difficult to enzyme (continuous enzyme digestion for four times) according to the published 4C protocol, so we reduced the fixation time and formaldehyde concentration to find the optimal enzyme digestion conditions, and chosen the optimum condition (1% formaldehyde + 8 min) (**Fig. 2**). In our study, lots of bioinformatics evidence supported that rs4728142 is a causal variant which regulates *IRF5* expression, so we aimed to detect the chromosome interaction around *IRF5* in a targeted way. According to our 4C results, we mainly focus on the *cis*-interactions around *IRF5*, not the *trans*-interactions, although we tried to reveal middle-range interactions with in 2Mb (**Reviewer 1, Question 4**). We conducted three repeated experiments and set up control samples, which showed that rs4728142 modulates the chromosomal loop by regulating ZBTB3 binding.

To eliminate the influence of noise signal, in this revision, we also used 293T cells (*IRF5* hardly expresses) as a control, and performed the 3C assays on 293T and SC cells. As the result, compared with 293T cells, rs4728142-containing region in SC cells has a stronger interaction with R5 (between *IRF5*-long and *IRF5*-short promoters, Fig. 3b in the revised manuscript) ($P < 0.0001$) (**Fig. 3**), and the rs4728142-associated enhancer in SC cells has a stronger interaction with R2 (*IRF5*-long promoter core region, Fig. 4b in the revised manuscript) ($P < 0.001$) and R5 (*IRF5*-short promoter core region, Fig. 4b in the revised manuscript) ($P < 0.0001$) (**Fig. 4**). In addition, we also performed 3C assays on other cell lines (sKO, ZBTB3-KO, and ZBTB3-OE SC cells) and made the SC cells as a control, and obtained the reliable results consistent with our mechanism model (as mentioned below, **Question 6: Fig. 3–8**).

As the reviewer worried, without major changes, we cannot say our 3C assay has advantages than the previous version. In this study, we mainly focused on the mechanism exploration and further validated our 4C results, not the technology itself, so in order to maintain scientific rigor, we changed “HR-3C” into “3C” in the revised manuscript. Thanks for the reminding.

Fig. 1 | The effect of different crosslinking conditions on 4C results¹⁸. Box plots showing the distribution of self-ligated and noncut reads (as their percentage of total number of mapped reads), as measured across experiments applying 2% formaldehyde to fix chromatin and using *HindIII* (n = 187), *BglII* (n = 7), *DpnII* (n = 196), or *NlaIII* (n = 200) as the primary restriction enzyme. Boxes indicate median and 25–75 percentile range, plus outliers. Same, but for experiments fixing chromatin with 0.5%, 1%, or 2% formaldehyde (and digesting with either *DpnII* or *NlaIII* as RE1; n = 7). **These results were derived from the original paper.**

1, 6: DL15000 marker; 2, 7: undigest, 3: 1.5% formaldehyde + 8 min; 4,9: 1% formaldehyde + 8 min; 5: 1% formaldehyde + 5 min; 8: 2% formaldehyde + 10 min; 10: 1.5% formaldehyde + 8 min

Fig. 2 | Exploration of enzyme digestion conditions of 3C and 4C assays on SC cells. As shown in the figure, different cross-linking conditions (different combinations of formaldehyde concentration and times) were used to explore the best enzyme digestion conditions.

Fig. 3 | The 3C results between rs4728142-containing region and the *IRF5* downstream alternative promoter in SC and 293T cells. Enrichment quantification of 3C assay at the rs4728142-containing region normalized to control in SC and 293T cells ($n = 3$). **a**, Electrophoresis results of 3C assay for the rs4728142-containing region in SC and 293T cells. **b**, Relative interaction frequencies between the rs4728142-containing region and representative *DpnII* cutting fragments indicated in Fig. 3b in the revised manuscript. Data are represented as the means \pm standard deviations (SD), and unpaired two-tailed Student's *t*-test is used to calculate *P*-values in **b**: *, $P < 0.05$; **, $P < 0.01$; ***, $P < 0.001$; ****, $P < 0.0001$.

Fig. 4 | The 3C results between rs4728142-associated enhancer and the *IRF5* promoter in SC and 293T cells. Enrichment quantification of 3C assay at the rs4728142-associated enhancer normalized to control in SC and 293T cells (n = 3). **a**, Electrophoresis results of 3C assay for the rs4728142-containing region in SC and 293T cells. **b**, Relative interaction frequencies between the rs4728142-associated enhancer and representative *DpnII* cutting fragments indicated in Fig. 4b in the revised manuscript. Data are represented as the means \pm standard deviations (SD), and unpaired two-tailed Student's *t*-test is used to calculate *P*-values in **b**: *, *P* < 0.05; **, *P* < 0.01; ***, *P* < 0.001; ****, *P* < 0.0001.

3.3 Are the authors using in solution ligation or in situ ligation (also referred to in nucleus ligation) in their 3C and 4C assays? This is not specified in the methods section but the high volume the ligation reaction is carried out in makes me suspect the authors have used in solution ligation, which has been shown to be inferior to in situ/in nucleus ligation (for example see Nagano et al Genome Biology 2015).

Response: Thanks for the reviewer's concerns. Yes, as the reviewer believes, we previously used in solution ligation (a high ligation volume) in our 3C and 4C assays, we have made specified modifications in the methods section of the revised manuscript. According to reviewer's suggestion, we used *in situ*/in nucleus ligation¹⁹ to perform *in situ* 3C assay for rs4728142-containing region 3C and rs4728142-associated enhancer 3C, and compared the results with in solution 3C results. As the results, rs4728142-containing region has a stronger interaction with R5 (between *IRF5*-long and *IRF5*-short promoters) (**Fig. 1**), and the rs4728142-associated enhancer has a strong interaction with R2 (*IRF5*-long promoter core region) and R5 (*IRF5*-short promoter core region) (**Fig. 2**). These results from *in situ* 3C (**Fig. 1** and **2**) suggest that both in solution ligation and *in situ* ligation can support our conclusions. In addition, we also performed 3C assays on other cell lines (sKO, ZBTB3-KO, and ZBTB3-OE SC cells) and made the SC cells as a control, we obtained the reliable results consistent with our mechanism model (as mentioned below, **Question 6: Fig. 3–8**). Besides, as the reviewer mentioned, it can be seen from the results that *in situ* 3C is really better than in solution 3C, indicating *in situ* 3C is a better choice in future studies.

Fig. 1 | The *in situ* 3C results compared with in solution 3C result between rs4728142-containing region and the *IRF5* downstream alternative promoter in SC cells. Enrichment quantification of in solution 3C and *in situ* 3C assays at the rs4728142-containing region normalized to control in SC cells (n = 3). **a**, Electrophoresis results of in solution 3C and *in situ* 3C assays for the rs4728142-containing region in SC cells. **b**, Relative interaction frequencies between the rs4728142-containing region and representative *DpnII* cutting fragments indicated in Fig. 3b in the revised manuscript. Data are represented as the means \pm standard deviations (SD), and unpaired two-tailed Student's *t*-test is used to calculate *P*-values in **b**: *, *P* < 0.05; **, *P* < 0.01; ***, *P* < 0.001; ****, *P* < 0.0001.

Fig. 2 | The *in situ* 3C results compared with *in solution* 3C results between rs4728142-associated enhancer and the *IRF5* promoter in SC cells. Enrichment quantification of *in solution* 3C and *in situ* 3C assays at the rs4728142-associated enhancer normalized to control in SC cells (n = 3). **a**, Electrophoresis results of *in solution* 3C and *in situ* 3C assays for the rs4728142-containing region in SC cells. **b**, Relative interaction frequencies between the rs4728142-associated enhancer and representative *DpnII* cutting fragments indicated in Fig. 4b in the revised manuscript. Data are represented as the means \pm standard deviations (SD), and unpaired two-tailed Student's *t*-test is used to calculate *P*-values in **b**: *, *P* < 0.05; **, *P* < 0.01; ***, *P* < 0.001; ****, *P* < 0.0001.

4. Figure 5f: I don't find the enrichment for ZBTB3/CTCF/RAD21 terribly convincing, especially given the y axis scale and the plateaus in the ZBTB3 and CTCF tracks over the *IRF5* gene. How many ZBTB3 peaks are there genome-wide? Is the ZBTB3 enrichment at rs4728142 determined by visual inspection? Would a peak-calling algorithm confirm this?

Response: Thanks for the rigorous comments. Yes, our previous ChIP-seq of ZBTB3 and RAD21 show marginal enrichment, instead of CTCF ChIP-seq. According to the reviewer's suggestions, in this revision, we re-performed the ChIP-seq and ChIP-qPCR for ZBTB3/RAD21 on the rs4728142-containing region, we did three replicates for each experiment. As the results, three ChIP-seq replicates were merged for peak calling, yielding 1647 peaks (MACS *P*-value < 0.05) for ZBTB3 and 15,344 peaks (MACS *P*-value < 0.05) for RAD21. The rs4728142-containing region receives a gained signal and obvious peak for both ZBTB3 (chr7: 128,573,630–128,574,431, *P*-value = 0.036) and RAD21 (chr7: 128,573,221–128,574,431, *P*-value = 0.037) (**Fig. 1**). The new ChIP-seq data has been

deposited into NCBI GEO (GSE168045). We have updated our new result into Fig. 5f in the revised manuscript.

Fig. 1 | ZBTB3 RAD21 and CTCF ChIP-seq signals at the *IRF5* nearby region in SC cells. Red dotted line denotes the rs4728142 location. Highlighted boxes denote the rs4728142-containing 3C/4C viewpoint (left red box) and its highest interaction frequency region in the *IRF5* downstream alternative promoter (right red box).

5. Is the CRISPR knockout of ZBTB3 a true ko (Figure 5j)? There is still quite a bit of protein visible in the western blot.

Response: Thanks for this reminding. As the reviewer worried, ZBTB3 does remain quite a bit. To obtain a complete ZBTB3-knockout (KO) cell line, we subcloned the ZBTB3-KO cell line we acquired, and then performed western blotting for ZBTB3 expression and RT-qPCR for the *IRF5* gene, *IRF5*-short transcripts, and *IRF5*-long transcripts. Western blotting results showed that ZBTB3 was almost completely knocked out (**Fig. 1, top**). RT-qPCR results showed that the expression level of both *IR5* gene and *IRF5*-short transcripts were significantly down-regulated ($P < 0.0001$), while that of *IRF5*-long transcripts were significantly up-regulated ($P < 0.01$) after ZBTB3 knockout ($P < 0.01$) (**Fig. 1, bottom**), further confirming our previous findings. We have updated our new result into Fig. 5i in the revised manuscript.

Fig. 1 | ZBTB3 mediates expression switching of the *IRF5*-short and *IRF5*-long transcripts in SC cells. Western blotting for validating the ZBTB3 knockout (KO) (top), and RT-qPCR for detecting the *IRF5* transcript expression in ZBTB3-KO SC cells compared with the unedited SC cells (control) (n = 3) (bottom). Data are represented as the means \pm standard deviations (SD), and unpaired two-tailed Student's *t*-test is used to calculate *P*-values: *, *P* < 0.05; **, *P* < 0.01; ***, *P* < 0.001; ****, *P* < 0.0001.

6. I am not convinced the model proposed in Figure 7 is sufficiently supported by the data. First, I could not find evidence in the manuscript to support that the cohesin complex binds preferentially the A allele of rs4728142? Second, as the authors outline in the discussion, the reporter/enhancer gene results don't support this model. Third (see also above), the ZBTB3 enrichment at rs4728142 is modest at best. Fourth, I don't see direct evidence that ZBTB3 is involved in enhancer-promoter looping. This is testable – for example, is the E-P loop present in dKO or sKO SC cells? Do enhancer and promoter in the rs4728142 region interact in the absence of ZBTB3 (CRISPR ko of ZBTB3 gene)?

Response: Thanks for the raised concerns. According to the reviewer's suggestions, we first performed RAD21 (plus CTCF and ZBTB3) ChIP-qPCR on SC cells. As the result, RAD21, ZBTB3 and CTCF were enriched at rs4728142 locus (all, *P* < 0.001) (**Fig. 1a**), and sequencing of the ChIP-qPCR products amplified at the rs4728142 locus demonstrated increased allele-specific binding for rs4728142-A when immunoprecipitated by anti-RAD21 antibody (**Fig. 1b**). **We have updated Fig. 1a into Extended Data Fig. 9c and the figure legend (L1821), and updated Fig. 1b into Fig. 5g and the figure legend (L1628) in the revised manuscript.**

Next, to confirm whether the rs4728142 site matching ZBTB3 motif is essential for its chromatin association, we used ChIP-qPCR to detect the binding of ZBTB3, RAD21, and CTCF at the rs4728142 locus in SC-sKO cells. As expected, when destroying rs4728142, the bindings of ZBTB3, RAD21, and CTCF are nearly lost (**Fig. 1c**), indicating the DNA sequences at rs4728142 site is essential for chromatin association of the investigated factors.

Yes, the luciferase experiment is only used to help analyze the properties of rs4728142 binding transcription factors *in vitro*, which is conducive for our next experimental design, and cannot reflect the specific functional mechanism under physiological state. Thus, we re-performed the ZBTB3 and RAD21 ChIP-seq on SC cells, and found that rs4728142-containing region overlay significant ZBTB3 and RAD21 binding peaks around the rs4728142-containing region (**Question 4**).

To detect whether ZBTB3 is involved in enhancer-promoter looping, we performed 3C assays on the rs4728142-sKO SC cells, ZBTB3-KO SC cells, and ZBTB3-overexpression (ZBTB3-OE) SC cells. As the result, rs4728142-sKO significantly weakened the interaction between rs4728142-containing region and R5 (between *IRF5*-long and *IRF5*-short promoters, see Fig. 3b in the revised manuscript) (*P* < 0.0001) (**Fig. 2**), and switched the rs4728142-associated enhancer from R6 (*IRF5*-short promoter core region, see Fig. 4b in

the revised manuscript) ($P < 0.001$) to R2 (*IRF5*-long promoter core region, see Fig. 4b in the revised manuscript) ($P < 0.001$) (**Fig. 3**). In consistent, ZBTB3-KO significantly weakened the interaction between rs4728142-containing region and R5 (between *IRF5*-long and *IRF5*-short promoters, see Fig. 3b in the revised manuscript) ($P < 0.0001$) (**Fig. 4**), and switched the rs4728142-associated enhancer from R6 (*IRF5*-short promoter core region, see Fig. 4b in the revised manuscript) ($P < 0.01$) to R2 (*IRF5*-long promoter core region, see Fig. 4b in the revised manuscript) ($P < 0.01$) (**Fig. 5**). On the contrary, ZBTB3-OE significantly strengthened the interaction between rs4728142-containing region and R5 (between *IRF5*-long and *IRF5*-short promoters, see Fig. 3b in the revised manuscript) ($P < 0.01$) (**Fig. 6**), and switched the rs4728142-associated enhancer from R2 (*IRF5*-long promoter core region, see Fig. 4b in the revised manuscript) ($P < 0.05$) to R6 (*IRF5*-short promoter core region, see Fig. 4b in the revised manuscript) ($P < 0.01$) (**Fig. 7**). The above data fully support our proposed mechanism model, and thanks again for the reviewer's valuable suggestions. **We have added these results into Extended Data Fig. 10, and the related description (L443-464) in the revised manuscript.**

Fig. 1 | CTCF, ZBTB3, and RAD21 bind at rs4728142 locus in SC cells. a, ChIP enrichments of CTCF, ZBTB3, and RAD21 at the rs4728142-containing region in SC cells as determined by ChIP-qPCR ($n = 3$). **b**, Sanger sequencing chromatograms of RAD21 ChIP-qPCR at the rs4728142 locus. **c**, ChIP enrichments of CTCF, ZBTB3, and RAD21 at the rs4728142-containing region in SC-sKO cells as determined by ChIP-qPCR ($n = 3$). Data are represented as the means \pm standard deviations (SD), and unpaired two-tailed Student's *t*-test is used to calculate *P*-values in **a** and **c**: *, $P < 0.05$; **, $P < 0.01$; ***, $P < 0.001$; ****, $P < 0.0001$.

Fig. 2 | The 3C results between rs4728142-containing region and the *IRF5* downstream alternative promoter in SC and SC-sKO cells. Enrichment quantification of 3C assay at the rs4728142-containing region normalized to control in SC and SC-sKO cells ($n = 3$). **a**, Electrophoresis results of 3C assay for the rs4728142-containing region in SC and SC-sKO cells. **b**, Relative interaction frequencies between the rs4728142-containing region and representative *DpnII* cutting fragments indicated in Fig. 3b in the revised manuscript. Data are represented as the means \pm standard deviations (SD), and unpaired two-tailed Student's *t*-test is used to calculate *P*-values in **b**: *, $P < 0.05$; **, $P < 0.01$; ***, $P < 0.001$; ****, $P < 0.0001$.

Fig. 3 | The 3C results between rs4728142-associated enhancer and the *IRF5* promoter in SC and SC-sKO cells. Enrichment quantification of 3C assay at the rs4728142-associated enhancer normalized to control in SC and sKO SC cells (n = 3). **a**, Electrophoresis results of 3C assay for the rs4728142-containing region in SC and sKO SC cells. **b**, Relative interaction frequencies between the rs4728142-associated enhancer and representative *DpnII* cutting fragments indicated in Fig. 4b in the revised manuscript. Data are represented as the means \pm standard deviations (SD), and unpaired two-tailed Student's *t*-test is used to calculate *P*-values in **b**: *, *P* < 0.05; **, *P* < 0.01; ***, *P* < 0.001; ****, *P* < 0.0001.

Fig. 4 | The 3C results between rs4728142-containing region and the *IRF5* downstream alternative promoter in SC and ZBTB3-KO SC cells. Enrichment quantification of 3C assay at the rs4728142-containing region normalized to control in SC and ZBTB3-KO SC cells (n = 3). **a**, Electrophoresis results of 3C assay for the rs4728142-containing region in SC and ZBTB3-KO SC cells. **b**, Relative interaction frequencies between the rs4728142-containing region and representative *DpnII* cutting fragments indicated in Fig. 3b in the revised manuscript. Data are represented as the means \pm standard deviations (SD), and unpaired two-tailed Student's *t*-test is used to calculate *P*-values in **b**: *, *P* < 0.05; **, *P* < 0.01; ***, *P* < 0.001; ****, *P* < 0.0001.

Fig. 5 | The 3C results between rs4728142-associated enhancer and the *IRF5* promoter in SC and ZBTB3-KO SC cells. Enrichment quantification of 3C assay at the rs4728142-associated enhancer normalized to control in SC and ZBTB3-KO SC cells (n = 3). **a**, Electrophoresis results of 3C assay for the rs4728142-associated enhancer in SC and ZBTB3-KO SC cells. **b**, Relative interaction frequencies between the rs4728142-associated enhancer and representative *DpnII* cutting fragments indicated in Fig. 4b in the revised manuscript. Data are represented as the means \pm standard deviations (SD), and unpaired two-tailed Student's *t*-test is used to calculate *P*-values in **b**: *, *P* < 0.05; **, *P* < 0.01; ***, *P* < 0.001; ****, *P* < 0.0001.

Fig. 6 | The 3C results between rs4728142-containing region and the *IRF5* downstream alternative promoter in SC and ZBTB3-OE SC cells. Enrichment quantification of 3C assay at the rs4728142-containing region normalized to control in SC and ZBTB3-OE SC cells (n = 3). **a**, Electrophoresis results of 3C assay for the rs4728142-containing region in SC and ZBTB3-OE SC cells. **b**, Relative interaction frequencies between the rs4728142-containing region and representative *DpnII* cutting fragments indicated in Fig. 3b in the revised manuscript. Data are represented as the means \pm standard deviations (SD), and unpaired two-tailed Student's *t*-test is used to calculate *P*-values in **b**: *, *P* < 0.05; **, *P* < 0.01; ***, *P* < 0.001; ****, *P* < 0.0001.

Fig. 7 | The 3C results between rs4728142-associated enhancer and the *IRF5* promoter in SC and ZBTB3-OE SC cells. Enrichment quantification of 3C assay at the rs4728142-associated enhancer normalized to control in SC and ZBTB3-OE SC cells (n = 3). **a**, Electrophoresis results of 3C assay for the rs4728142-associated enhancer in SC and ZBTB3-OE SC cells. **b**, Relative interaction frequencies between the rs4728142-associated enhancer and representative *DpnII* cutting fragments indicated in Fig. 4b in the revised manuscript. Data are represented as the means \pm standard deviations (SD), and unpaired two-tailed Student's *t*-test is used to calculate *P*-values in **b**: *, *P* < 0.05; **, *P* < 0.01; ***, *P* < 0.001; ****, *P* < 0.0001.

Minor points:

Line 134: should be 'interacting' instead of 'interacted' genes

Response: Thanks for the reminding. According to the reviewer's suggestion, we have corrected 'interacting' instead of 'interacted' genes it in the revised manuscript.

Typo in Figure 2a: should be regulatory instead of reglatory

Response: Thanks for the reminding. According to the reviewer's suggestion, we have corrected 'regulatory' instead of 'regulatory' in Fig. 2a of the revised manuscript.

I think Extended Data Fig. 2b shows IRF5 expression data and Extended Data Fig. 2c shows chromatin accessibility (ATAC-seq).

Response: Thanks for the suggestion. According to the reviewer's suggestion, we have corrected IRF5 expression data in the Extended Data Fig. 2b and Chromatin accessibility (ATAC-seq) in the Extended Data Fig. 2c in the revised manuscript, which is more conducive to showing and understanding our results for readers.

Lines 195-200: "Moreover, the genomic region surrounding the pleiotropic variant rs4728142 had relatively higher chromatin accessibility in monocytes compared with other hematopoietic cells (Extended Data Fig. 2b), and the potential target gene IRF5 also showed distinct expression in monocytes according to the DICE dataset⁴¹ (Extended Data Fig. 2c), which highlights the lineage-specific role of rs4728142 in the development of autoimmune comorbidities."

It may be true that rs4728142 has relatively higher chromatin accessibility in monocytes, but in my view it would be fairer to say chromatin accessibility is low in monocytes and even lower in other immune cells.

Response: Thanks for the suggestion! As the reviewer mentioned, our statement is biased. To ensure the preciseness and correctness of the description, we have corrected it to "Moreover, the genomic region surrounding the pleiotropic variant rs4728142 obtains low chromatin accessibility in monocytes and has absent open chromatin in other hematopoietic cells (Extended Data Fig. 2b)." in the revised manuscript.

Extended Data Fig. 2d: how relevant is the GM12878 data here, given that the rest of the manuscripts describes the role of rs4728142 in monocytes or SC cells (monocyte-derived cell line)?

Response: Thanks for the suggestion! Actually, our initial idea was to show the consistence of epigenetic status in GM12878 cells. As the reviewer mentioned, to maintain the logic of the preceding and following, we have deleted it and its relevant description in the revised manuscript.

Lines 215-217: "Consistent with the public data, the rs4728142 surrounding region in the SC cells exhibited both promoter and enhancer activities (all, $P < 0.01$) (Fig. 2a, b)."

Again – not an accurate description of what the data is showing. I think "...exhibited chromatin marks associated with both promoter and enhancer activities" is better.

Response: Thanks for the good suggestion. Indeed, as the reviewer's think, our statement is not rigorous enough. As the reviewer mentioned, we have corrected "Consistent with the public data, the rs4728142 surrounding region in the SC cells exhibited chromatin marks associated with both promoter and enhancer activities (all, $P < 0.01$) (Fig. 2a, b)." instead of "Consistent with the public data, the rs4728142 surrounding region in the SC cells exhibited both promoter and enhancer activities (all, $P < 0.01$) (Fig. 2a, b)."

Figure 2c-f: this strong dampening effect of rs4728142 on the enhancer/promoter activity of its flanking region is reminiscent of a transcriptional silencer – have the authors considered this possibility? It may be interesting especially in the context of ZBTB3 as a proposed transcriptional repressor. Related: the enrichment of active histone mods (H3K27ac, H3K4me1 and H3K4me3) is modest at best for the rs4728142 region itself – is it conceivable that this region rather limits the spreading of these active chromatin marks (see Figure 2a)? The authors refer to this region as located at the boundary of an enhancer several times in the manuscript, but I couldn't figure out how this observation is reflected in the proposed mechanism for rs4728142.

Response: Thanks for the reviewer's constructive comments. Given the chromatin status pattern surrounding rs4728142 and overlapped MNase-seq peak in GM12878, we originally wanted to emphasize that the rs4728142-containing region could be one associated with nucleosome-bound protein. Luckily, we revealed that ZBTB3 could function as a structural protein that could mediate the loop formation and affect *IRF5* promoter usage at rs4728142 locus. Instead of the previous finding regarding the repressive functional of the BTB/POZ domain, our bioinformatics analysis and experiments in this study could provide primary evidence, at least in this locus, supporting the premise that ZBTB3 may function as a structural factor.

In general, can the authors explain why they use both 3C and 4C? They are technically similar so 4C should not be used to validate results obtained with 3C, or vice versa.

Response: Thanks for the good question. As the reviewer mentioned, 3C and 4C technology are technically similar. 4C technology is a "one vs all" three-dimensional (3D) genomics technology, while 3C technology is a "one vs one" 3D genomic technology^{8,20}. 4C technology helps us to locate the general location of chromosome interactions and convenient for us to better design the target area of 3C. On the other hand, to ensure the correctness and preciseness of our initial results, we used these two related technologies to determine the phenotype. As the reviewer mentioned, one of the technologies can explain the chromosome interaction, so in the revised manuscript, we use the 3C assay which is more time-saving and cost-effective.

Lines 269-270: "Compared with previous 3C versions^{47,48}, HR-3C utilizes a 4-bp cutting enzyme instead of the traditional 6-bp cutting enzyme,..."

There are a lot (a lot!) of 3C/4C/Hi-C studies out there using 4 bp cutting enzymes such as Mbol or DpnII. This is not new!

Response: Thanks for the reminding. As the reviewer mentioned, we just explored some cross-linking and enzyme digestion conditions, and had not improved the 3C technology, so we have corrected "3C" instead of "HR-3C" and corrected the corresponding descriptions in the revised manuscript.

Figure 3c: does this result suggest no interactions with any other fragment than R5? This would be quite unlikely given the nature of 3C and 4C type assays. Why is this result so different from Figure 4c (I know the bait regions differ slightly, but can this be the only

explanation?)

Response: Thanks for the concern. After careful proofreading, we found that the 3C result of rs4728142-containing region was indeed abnormal, this result may be caused by improper technical operation and lacks of control. To eliminate the influence of noise signal, we used 293T cells (*IRF5* hardly expresses) as a control, and performed the 3C assays on 293T and SC cells. As the results, compared with 293T cells, rs4728142-containing region in SC cells has a stronger interaction with R5 (between *IRF5*-long and *IRF5*-short promoters, see Fig. 3b in the revised manuscript) ($P < 0.0001$) (Fig. 1). **We have updated our data into Fig. 3c and Extended Data Fig. 5c in the revised manuscript.**

Fig. 1 | The 3C results between rs4728142-containing region and the *IRF5* downstream alternative promoter in SC and 293T cells. Enrichment quantification of 3C assay at the rs4728142-containing region normalized to control in SC and 293T cells ($n = 3$). **a**, Electrophoresis results of 3C assay for the rs4728142-containing region in SC and 293T cells. **b**, Relative interaction frequencies between the rs4728142-containing region and representative *DpnII* cutting fragments indicated in Fig. 3b in the revised manuscript. Data are represented as the means \pm standard deviations (SD), and unpaired two-tailed Student's *t*-test is used to calculate *P*-values: *, $P < 0.05$; **, $P < 0.01$; ***, $P < 0.001$; ****, $P < 0.0001$.

Figure 5 i and j minor point: for clarity please add ZBTB3 overexpression and ZBTB3 knockdown/knockout, respectively, to the legends next to the bar graphs (orange rectangle)

Response: Thanks for the good suggestions. we have added ZBTB3 overexpression (ZBTB3-OE) and ZBTB3 knockout (ZBTB3-KO) to the legends next to the bar graphs of Fig. 5i and j, and corrected the corresponding description in the revised manuscript.

Model Figure 7 minor point: given that the *IRF5* short transcripts are light blue, for clarity I would colour the *IRF5* short promoter light blue too. To me, the *IRF5* long promoter and

IRF5 short promoter look the same colour, whereas the respective transcripts are coloured differently.

Response: Thanks for the good suggestions. To distinguish the promoter regions corresponding to different transcripts, we have corrected the *IRF5*-long promoter with dark blue instead of light blue, and *IRF5*-short promoter with light blue in the same color number in Fig. 7 of the revised manuscript.

As a general note: I have the impression that the second half of the paper, including the discussion, is better stylistically and linguistically than the first.

Response: Thanks for your comments. We have further optimized the first half in the revised manuscript according to your and other reviewers' suggestions.

Reviewer #3 (Remarks to the Author):

In this manuscript, the authors tried to delineate the causal pathways of how genetic non-coding variants leading to autoimmune diseases by studying the shared genetic events on the autoimmune comorbidity. A group of causal pleiotropic variants and their target genes were generated through systematic analysis. Within which, rs4728142 and its target IRF5 were top ranked. Further investigation on the underlying molecular mechanism showed that the risk allele of rs4728142-A tends to switch IRF5 alternative promoter usage mediated by ZBTB3 which is a chromatin looping structure regulator. The aberrant epigenetic and transcriptional regulation of IRF5 transcripts ultimately results in macrophage M1 polarization leading to autoimmune diseases.

The manuscript was well-written with sufficient background introduction and adequate discussion. This work provided a valuable resource for studying the function of noncoding pleiotropic variants causally associated with autoimmune diseases. Furthermore, the mechanism study on the structure of the enhancer and alternative promoter interaction determining the outcome of differential expressed transcript isoforms of essential gene is very thorough with nice supporting data. Although ZBTB3 has already been shown to interact with rs4728142 risk allele in the prior published study, this work presented a relatively intact model and detailed mechanism explanation on the epigenetic and transcriptional regulation of IRF5 gene. Several weaknesses are identified that need to be addressed to improve the quality of the current manuscript.

Response: Thanks for the reviewer's excellent summary and commendation.

1. Focusing on pleiotropic SNPs in autoimmune diseases could emphasize the biological significance and efficiently narrow down the candidate lists. However, in this study the characterized candidate and target gene regulatory mechanism were not clearly related to autoimmune comorbidity. Any supportive evidences indicating either the expression level or different isoforms of IRF5 contributes to comorbidity will improve the quality of this work.

Response: Thanks for the constructive suggestion. First, we leveraged blood-derived (peripheral blood mononuclear cells) RNA-seq cohorts of paired RA/SLE patients and healthy individuals from two publications (Shchetynsky K et al. *Arthritis Res Ther.* 2017¹ and Mistry P et al. *Proc Natl Acad Sci U S A.* 2019²). Based on these independent datasets, we noticed that both *IRF5* gene (RA, $P = 0.0014$; SLA, $P = 0.0057$) and *IRF5*-short transcripts (RA, $P = 0.0044$) show higher expression in RA or SLE patients than healthy individuals, while *IRF5*-long transcripts (RA, $P = 0.082$; SLA, $P = 0.057$) only exhibit weak differences (**Fig. 1**). **We have added this result into Extended Data Fig. 9h, and the related description (L480-483) in the revised manuscript.** The evidence partially supports our major findings and indicates a likely pathogenic role of *IRF5* gene and *IRF5*-short transcripts in RA/SLE development. Second, we also inspected disease-specific *IRF5* expression patterns in Classical Monocytes from the ImmuNexUT project (Ota M et al. *Cell.* 2021¹⁰), which includes 28 distinct immune cell subsets from 337 patients diagnosed with 10 categories of immune-mediated diseases and 79 healthy volunteers. We found that *IRF5* gene expression is elevated in many autoimmune diseases, such as SLE and IBD (**Fig. 2**).

Fig. 1 | Investigation of *IRF5* gene, *IRF5*-short, and *IRF5*-long transcripts expressions in autoimmune disease patients. a, Comparison of *IRF5* gene expression in blood RNA-seq samples between healthy individuals and RA/SLE patients (n = 23 healthy, n = 12 RA, n = 11 SLE, Mann Whitney U Test). b, Comparison of *IRF5*-short transcripts expression in blood RNA-seq samples between healthy individuals and RA/SLE patients (n = 23 healthy, n = 12 RA, n = 11 SLE, Mann Whitney U Test). c, Comparison of *IRF5*-long transcripts expression in blood RNA-seq samples between healthy individuals and RA/SLE patients (n = 23 healthy, n = 12 RA, n = 11 SLE, Mann Whitney U Test).

Fig. 2 | Investigation of *IRF5* expression in Classical Monocytes among different autoimmune diseases.

2. As to the valuable data resources of pleiotropic SNPs in autoimmune diseases, I only found 3 candidates with similar score as rs4728142, according to the scoring system. Therefore, I'm curious about how many pleiotropic variant candidates have high confidence that they have causal regulatory roles on their targets? Which analysis have

more weight contribute to the prediction? And whether this analysis could be expanded to non-pleiotropic SNP scoring?

Response: Thanks for the good questions! According to our current logic, we evaluated the potential target genes of these pleiotropic SNPs (Table S9) and the causality of them (Table S10) independently. For the issue of ascertaining causal regulatory roles of pleiotropic variant candidates on their gene targets, we combined both causality evidence and target gene assignment to identify high confidence hits. As the result, by requesting the mandatory condition for “Is Likely Regulatory Variant (functional prediction score satisfies regBase PHRED score ≥ 15 and cepip combined score ≥ 0.6 and RegulomeDB score $\geq 2c$)”, we in total uncovered 22 causal pleiotropic variants with strong regulatory potentials, in which each variant could link to 1-7 target genes. However, someone may focus far more on whether the causal variant is colocalized with eQTLs, thus, more weight need be counted to satisfy such condition. Both causal variant prioritization and target gene identification strategies can be expanded to other GWAS results when all necessary data are available, and some of these functions had been integrated to our previously developed tool VarNote-REG (Huang D et al. *Genome Res.* 2020¹⁰). **We have briefly discussed the prioritization strategy and its generalization usage in the discussion part (L555-558) of our revised manuscript.**

3. What is ZBTB3 expression level and chromatin binding sites across immune cell types? ATAC-seq showed different open chromatin structure among monocyte, B cell and T cell. IRF5 showed high expression and specificity in monocytes and macrophage but not other lymphocytes might be owing to the pre-existing epigenetic traits and the present of regulatory molecular such as ZBTB3. Furthermore, did ZBTB3 knock-down in Fig. 5j disrupt chromatin loop formation between the rs4728142 enhancer region and the short IRF5 promoter?

Response: Thanks for these constructive comments. By querying the expression level of *ZBTB3* across different immune cells from the DICE project (Schmiedel BJ et al. *Cell.* 2018⁹), we noticed that *ZBTB3* shows relatively higher expression level in a type of Monocytes (**Fig. 1**). After querying on QTLbase¹¹, we also found that rs4728142 obtains the largest effect size being eQTLs in Monocytes than other tissue/cell types (**Fig. 2**), implying a more active chromatin states at rs4728142-containing region in Monocytes. **We have added Fig. 1b and Fig. 2 into Extended Data Fig. 2d and e, and related description (L200-203) in the revised manuscript.**

To obtain a complete ZBTB3 knockout (KO) cell line, we subcloned the ZBTB3-KO SC cell line we previous acquired, and then carried out western blotting for ZBTB3 expression and RT-qPCR for the *IRF5* gene, *IRF5*-short transcripts, and *IRF5*-long transcripts. Western blotting results showed that ZBTB3 was almost completely knocked out (**Fig. 3, top**). RT-qPCR results showed that both *IRF5* gene and *IRF5*-short transcripts were significantly down-regulated ($P < 0.0001$), while *IRF5*-long transcripts were significantly up-regulated ($P < 0.01$) (**Fig 3, bottom**). **We have updated the result into Fig. 5i in the revised manuscript.** Next, we performed 3C assays on the rs4728142-sKO SC cells, ZBTB3-KO SC cells, and ZBTB3-overexpression (OE) SC cells. As the result, ZBTB3-KO significantly

weakened the interaction between rs4728142 and R5 (between *IRF5*-long and *IRF5*-short promoters, Fig. 3b in the revised manuscript) ($P < 0.0001$) (**Fig. 4**), and switched the rs4728142-associated enhancer from R6 (*IRF5*-short promoter core region, Fig. 4b in the revised manuscript) ($P < 0.01$) to R2 (*IRF5*-long promoter core region, Fig. 4b in the revised manuscript) ($P < 0.01$) (**Fig. 5**). In contrast, rs4728142-sKO significantly weakened the interaction between rs4728142-containing region and R5 (between *IRF5*-long and *IRF5*-short promoters, Fig. 3b in the revised manuscript) ($P < 0.0001$) (**Fig. 6**), and switched the rs4728142-associated enhancer from R6 (*IRF5*-short promoter core region, Fig. 4b in the revised manuscript) ($P < 0.001$) to R2 (*IRF5*-long promoter core region, Fig. 4b in the revised manuscript) ($P < 0.001$) (**Fig. 7**). On the contrary, ZBTB3-OE significantly strengthened the interaction between rs4728142 and R5 (between *IRF5*-long and *IRF5*-short *IRF5* promoters, Fig. 3b in the revised manuscript) ($P < 0.0001$) (**Fig. 8**), and switched the rs4728142-associated enhancer from R2 (*IRF5*-long promoter core region, Fig. 4b in the revised manuscript) ($P < 0.01$) to R6 (*IRF5*-short promoter core region, Fig. 4b in the revised manuscript) ($P < 0.01$) (**Fig. 9**). We added these results into Extended Data Fig. 10, and the related description (L443–464) in the revised manuscript.

Fig. 1 | Comparison of *ZBTB3* gene expression levels among different immune cell types in DICE cohort. a, *ZBTB3* expression in different immune cell types. b, Pairwise comparison of *ZBTB3* gene expression across different immune cell types.

All eQTL data (+/- 10M region)

Total: 256 entries, showing 1 to 8

Type keywords to filter Show All Data (256)

Trait#	Tissue#	Effective Allele	P-Value#	Effect Size#	SE#	FDR#	Dataset#
IRF5	Blood-Monocyte	-	4.760e-31	-12.9058	-	4.910e-28	24604202_EUR_2014
IRF5	Blood-Monocyte	-	3.400e-28	-11.9807	-	2.850e-25	24604202_EUR_2014
IRF5	Blood-Monocyte	-	8.260e-24	-11.0534	-	7.470e-21	24604202_EUR_2014
IRF5	Blood-Monocyte	-	2.900e-21	-10.5366	-	3.830e-18	24604202_EUR_2014
IRF5	Blood-T cell CD4+	A	2.910e-9	-5.9365	0.9692	-	28248954_EUR_2017
IRF5	Blood-T cell CD8+	A	4.190e-5	-4.0968	0.9843	0.3220	28248954_EUR_2017
IRF5	Blood	-	1.060e-21	-0.7002	0.0732	-	25951796_EUR_2015
IRF5	Blood	A	1.700e-133	-0.6845	0.0278	-	28065468_MIX_2017
IRF5	Skin	-	1.810e-8	-0.6639	0.1179	-	25951796_EUR_2015
IRF5	Blood	-	6.950e-16	-0.5912	0.0682	1.860e-12	26917434_EUR_2016
IRF5	Brain-Cerebellum	A	5.210e-13	-0.5448	0.0699	1.050e-8	22685416_EUR_2012
IRF5	Lymphocyte	G	2.590e-27	-0.5287	0.0488	-	25951796_EUR_2015

Fig. 2 | Effect size comparison of eQTL (rs4728142-*IRF5*) among different tissue/cell types in QTLbase.

Fig. 3 | ZBTB3 mediates expression switching of *IRF5*-short and *IRF5*-long transcripts in SC cells. Western blotting for validating the ZBTB3 knockout (KO) (top), and RT-qPCR for detecting the *IRF5* transcript expression in ZBTB3-KO SC cells compared with the unedited SC cells (control) (n = 3) (bottom). Data are represented as the means \pm standard deviations (SD), and unpaired two-tailed Student's *t*-test is used to calculate *P*-values: *, *P* < 0.05; **, *P* < 0.01; ***, *P* < 0.001; ****, *P* < 0.0001.

Fig. 4 | The 3C results between rs4728142-containing region and the *IRF5* downstream alternative promoter in SC and ZBTB3-KO SC cells. Enrichment quantification of 3C assay at the rs4728142-containing region normalized to control in SC and ZBTB3-KO cells (n = 3). **a**, Electrophoresis results of 3C assay for the rs4728142-containing region in SC and ZBTB3-KO SC cells. **b**, Relative interaction frequencies between the rs4728142-containing region and representative *DpnII* cutting fragments indicated in Fig. 3b in the revised manuscript. Data are represented as the means \pm standard deviations (SD), and unpaired two-tailed Student's *t*-test is used to calculate *P*-values in **b**: *, *P* < 0.05; **, *P* < 0.01; ***, *P* < 0.001; ****, *P* < 0.0001.

Fig. 5 | The 3C results between rs4728142-associated enhancer and the *IRF5* promoter in SC and ZBTB3-KO SC cells. Enrichment quantification of 3C assay at the rs4728142-associated enhancer normalized to control in SC and ZBTB3-KO SC cells ($n = 3$). **a**, Electrophoresis results of 3C assay for the rs4728142-associated enhancer in SC and ZBTB3-KO SC cells. **b**, Relative interaction frequencies between the rs4728142-associated enhancer and representative *DpnII* cutting fragments indicated in Fig. 4b in the revised manuscript. Data are represented as the means \pm standard deviations (SD), and unpaired two-tailed Student's *t*-test is used to calculate *P*-values in **b**: *, $P < 0.05$; **, $P < 0.01$; ***, $P < 0.001$; ****, $P < 0.0001$.

Fig. 6 | The 3C results between rs4728142-containing region and the *IRF5* downstream alternative promoter in SC and SC-sKO cells. Enrichment quantification of 3C assay at the rs4728142-containing region normalized to control in SC and SC-sKO cells (n = 3). **a**, Electrophoresis results of 3C assay for the rs4728142-containing region in SC and SC-sKO cells. **b**, Relative interaction frequencies between the rs4728142-containing region and representative *DpnII* cutting fragments indicated in Fig. 3b in the revised manuscript. Data are represented as the means \pm standard deviations (SD), and unpaired two-tailed Student's *t*-test is used to calculate *P*-values in **b**: *, *P* < 0.05; **, *P* < 0.01; ***, *P* < 0.001; ****, *P* < 0.0001.

Fig. 7 | The 3C results between rs4728142-associated enhancer and the *IRF5* promoter in SC and sKO SC cells. Enrichment quantification of 3C assay at the rs4728142-associated enhancer normalized to control in SC and SC-sKO cells (n = 3). **a**, Electrophoresis results of 3C assay for the rs4728142-containing region in SC and SC-sKO cells. **b**, Relative interaction frequencies between the rs4728142-associated enhancer and representative *DpnII* cutting fragments indicated in Fig. 4b in the revised manuscript. Data are represented as the means \pm standard deviations (SD), and unpaired two-tailed Student's *t*-test is used to calculate *P*-values in **b**: *, *P* < 0.05; **, *P* < 0.01; ***, *P* < 0.001; ****, *P* < 0.0001.

Fig. 8 | The 3C results between rs4728142-containing region and the *IRF5* downstream alternative promoter in SC and ZBTB3-OE SC cells. Enrichment quantification of 3C assay at the rs4728142-containing region normalized to control in SC and ZBTB3-OE SC cells (n = 3). **a**, Electrophoresis results of 3C assay for the rs4728142-containing region in SC and ZBTB3-OE SC cells. **b**, Relative interaction frequencies between the rs4728142-containing region and representative *DpnII* cutting fragments indicated in Fig. 3b in the revised manuscript. Data are represented as the means \pm standard deviations (SD), and unpaired two-tailed Student's *t*-test is used to calculate *P*-values in **b**: *, *P* < 0.05; **, *P* < 0.01; ***, *P* < 0.001; ****, *P* < 0.0001.

Fig. 9 | The 3C results between rs4728142-associated enhancer and the *IRF5* promoter in SC and ZBTB3-OE SC cells. Enrichment quantification of 3C assay at the rs4728142-associated enhancer normalized to control in SC and ZBTB3-OE SC cells (n = 3). **a**, Electrophoresis results of 3C assay for the rs4728142-associated enhancer in SC and ZBTB3-OE SC cells. **b**, Relative interaction frequencies between the rs4728142-associated enhancer and representative *DpnII* cutting fragments indicated in Fig. 4b in the revised manuscript. Data are represented as the means \pm standard deviations (SD), and unpaired two-tailed Student's *t*-test is used to calculate *P*-values in **b**: *, *P* < 0.05; **, *P* < 0.01; ***, *P* < 0.001; ****, *P* < 0.0001.

4. The authors compared gene expression profile of the control SC cells (rs4728142 A/G allele) to sKO or dKO cells, and concluded that the rs4728142 (A) allele's main effect is promoting M1 macrophage differentiation which drives autoimmune diseases. The evidence provided was not sufficient. To have a fair comparison, the authors need to generate SC cells with A/A and G/G alleles to compare to A/G control cells. If low efficiency of HDR is the problem, any other cell line available should be tested. The authors also need to test if the long *IRF5* and the short *IRF5* can influence M1 macrophage polarization in opposite ways. Will ZBTB3 over-expression and knock-down (shown in Fig. 5i and 5j) also affect SC cell's M1 polarization?

Response: Thanks for the constructive comments. According to the reviewer's suggestion, we acquired the rs4728142 with homozygous GG (rs4728142-GG) and homozygous AA (rs4728142-AA) SC cells by Cas9-initiated homology-directed repair (HDR) with extensive efforts (**Fig. 1a**). We used RT-qPCR to detect the expression level of *IRF5* gene and two major forms of *IRF5* transcripts. Expectedly, compared with the rs4728142-GG SC cells, the rs4728142-AA SC cells increased the expression of *IRF5* gene (*P* < 0.001) and *IRF5*-short transcripts (*P* < 0.05), and decreased the expression of *IRF5*-long transcripts (*P* < 0.05), but had no effect on the nearest gene, *TNPO3* (**Fig. 1b**). Next, we induced the rs4728142-GG and rs4728142-AA SC cells to M1 macrophages, and detected the macrophage M1 polarization markers (*ATF3*, *COX2*, *INDO*, *SLC7A5*, and *CCR7*) by RT-qPCR. Compared with the rs4728142-GG induced-SC cells, all M1 polarization markers in the rs4728142-AA induced-SC cells were significantly increased, indicating that rs4728142 could modulate the macrophage M1 polarization (**Fig. 1c**). In addition, we detected the expression of key M1 proinflammatory cytokines^{6,8}. As the result, compared with the rs4728142-GG induced-SC cells, the expression of *IL-1 β* , *IL-6*, *IL-8*, and *TNF α* in the rs4728142-AA induced-SC cells were significantly increased (all, *P* < 0.05) (**Fig. 1d**). Together, the above data suggest that the allele A of rs4728142 could promote proinflammatory cytokine expression through up-regulating the expression of *IRF5*-short transcripts and *IRF5* gene, and down-regulating the expression of the *IRF5*-long transcripts, and then contributes to the M1 macrophage polarization. We have added these results into Fig. 7a–d, the related description (L503–521), and the attachment information (Supplementary Table 14) in the revised manuscript.

To investigate the functional difference between the *IRF5*-short and *IRF5*-long transcripts, we designed the sgRNAs targeting *IRF5*-short promoter or *IRF5*-long promoter, and used

CRISPRa technology to specifically activate the *IRF5*-short or *IRF5*-long transcripts. As the result, sgRNA targeting *IRF5*-short promoter significantly promotes the expression of *IRF5* gene ($P < 0.0001$) and *IRF5*-short transcripts ($P < 0.0001$) without affecting the expression of *IRF5*-long transcripts, while sgRNA targeting *IRF5*-long promoter significantly promote the expression of *IRF5* gene ($P < 0.001$) and *IRF5*-long transcripts ($P < 0.0001$) without altering the expression of *IRF5*-short transcripts (**Fig. 2a**). Next, we induced the *IRF5*-short promoter-activated or *IRF5*-long promoter-activated SC cells to M1 macrophages, and detected the macrophage M1 polarization markers (*ATF3*, *COX2*, *INDO*, *SLC7A5*, and *CCR7*) by RT-qPCR. Expectedly, all M1 polarization markers in both the induced-SC cells were significantly increased (all, $P < 0.01$) (**Fig. 2b**), inferring that both *IRF5*-short and *IRF5*-long transcripts could influence M1 macrophage polarization in the same way.

As the reviewer mentioned, we also detected if ZBTB3 OE and ZBTB3 KO affect SC cell's M1 polarization. Consequently, compared with SC cells, all M1 polarization markers in the ZBTB3-OE SC cells were significantly increased (all, $P < 0.05$), and all M1 polarization markers in the ZBTB3-KO SC cells were significantly decreased (all, $P < 0.01$) (**Fig. 3**), indicating that ZBTB3 could affect SC cell's M1 polarization.

Fig. 1 | Base editing at rs4728142 in SC cells and the related phenotype detection. a, the strategy of sing-base editing by CRISPR knock-in technology. The sKO sgRNA and single-stranded donor oligonucleotides (ssODN) with major allele-G or minor allele-A were co-transfected into the single cell-derived SC-Cas9 cells, and single cells were sorted for identification by Sanger sequencing. The green multilateral shape denotes Cas9 protein, and the red triangle denotes the cutting site. **b**, RT-qPCR for *IRF5* transcripts in rs4728142-AA SC cells compared with rs4728142-GG SC cells (n = 3). **c**, RT-qPCR for M1 polarization markers (*ATF3*, *COX2*, *INDO*, *SLC7A5*, and *CCR7*) in rs4728142-AA induced-SC cells compared with rs4728142-GG induced-SC cells (n = 3). **d**, RT-qPCR for key M1

proinflammatory cytokines (*IL-1 β* , *IL-6*, *IL-8*, and *TNF α*) in rs4728142-AA induced-SC cells compared with rs4728142-GG induced-SC cells (n = 3). Data are represented as the means \pm standard deviations (SD), and unpaired two-tailed Student's *t*-test is used to calculate *P*-values in **b–d**: *, *P* < 0.05; **, *P* < 0.01; ***, *P* < 0.001; ****, *P* < 0.0001.

Fig. 2 | *IRF5*-short and *IRF5*-long transcripts promote M1 macrophage polarization. **a**, RT-qPCR for *IRF5* gene, *IRF5*-short, and *IRF5*-long transcripts in *IRF5*-short promoter-activated or *IRF5*-long promoter-activated SC cells. **b**, RT-qPCR for M1 polarization markers (*ATF3*, *COX2*, *INDO*, *SLC7A5*, and *CCR7*) in *IRF5*-short transcripts-activated or *IRF5*-long transcripts-activated induced-SC cells (n = 3). Data are represented as the means \pm standard deviations (SD), and unpaired two-tailed Student's *t*-test is used to calculate *P*-values in **a** and **b**: *, *P* < 0.05; **, *P* < 0.01; ***, *P* < 0.001; ****, *P* < 0.0001.

Fig. 3 | The function of ZBTB3 in M1 macrophage polarization. RT-qPCR for M1 polarization markers (*ATF3*, *COX2*, *INDO*, *SLC7A5*, and *CCR7*) in ZBTB3 overexpression (ZBTB3-OE) SC cells (**a**) or ZBTB3 knockout (ZBTB3-KO) SC-Cas9 cells (**b**) compared with SC cells or the unedited SC-Cas9 cells (n = 3). Data are represented as the means \pm standard deviations (SD), and unpaired two-tailed Student's *t*-test is used to calculate *P*-values in **a** and **b**: *, *P* < 0.05; **, *P* < 0.01; ***, *P* < 0.001; ****, *P* < 0.0001.

Reviewer #4 (Remarks to the Author):

The manuscript by Li and colleagues conducted systems survey on the human autoimmune disease pleiotropic loci, aiming to prioritize causal pleiotropic variants and genes. The study produced a rich set of functional annotation data and certainly inform in depth functional follow-up studies in the field. The authors then focused the top-ranked pleiotropic variant rs4728142 and proved its causal roles in regulating IRF5 isoform expression through a promoter usage mechanism. Thus, they identified previously unappreciated mechanisms within this well-known autoimmune disease-associated locus and may explain in a new angle how the risk haplotype define precise expression of IRF5 isoforms. Overall, the work is well designed, the topic is important, and the manuscript is well written with a clear scientific logic behind it. I suggest the following points for a major revision to improve the work.

Response: Thanks for the reviewer's highlight and positive response.

1. There should be in the figure 1 an overall workflow for systems functional annotation of autoimmune disease-associated pleiotropic loci, thus making the computational and statistical analysis pipeline sufficiently clear for the readers to understand.

Response: Thanks for the good suggestion! To make the computational and statistical analysis pipeline sufficiently clear for the readers to understand, we made an overall workflow in the revised manuscript. Our workflow for systems functional annotation of autoimmune disease-associated pleiotropic loci includes four parts: first, 440 sentinel pleiotropic variants from 24 autoimmune diseases were collected and curated; second, 54 full GWAS summary statistics of autoimmune diseases were evaluated by Variant-level pleiotropy estimation (gwas-pw), GWAS fine-mapping analysis (FINEMAP), and Variant annotation (VEP, RegulomeDB, regBase, and GWAS4D); third, cis-eQTL colocalization analysis, variant-target gene analysis, variant-gene bipartite network, and cis/trans-gene-regulatory network were used to target gene identification and network analysis; last, Evidence-based scoring schemes, including Pleiotropy evidence, Causality evidence, Colocalization evidence, and Regulatory potential evidence, were used to prioritize the variants (Fig. 1). We have added this overall workflow into Fig. 1a, the related description, and figure legend (L1508–1509) in the revised manuscript.

Fig. 1 | Computational workflow for systematically functional annotation and prioritization of autoimmune disease-associated pleiotropic variants.

2. Figure 2c-f and Extended Data Fig. 3e/f: How many times were each reporter assay

repeated? There are huge difference in regulatory activity between adjacent upstream sequence of rs4728142 and the sequence plus rs4728142-containing region measured by promoter assays in 293T but not in SC cells as shown in the panels d and f. Does this mean rs4728142-containing sequence dependent, experimental reproducibility issue and/or cell-type-specific manner? Please clarify the discrepancies and perform additional experiments to support the finding?

Response: Thanks for the reminding. After careful proofreading, we were sorry to find that we mistakenly used a duplicate result in Fig. 2e (293T enhancer assay) as Fig. 2f (293T promoter assay) when we edited the Figures. For Fig. 2d (293T promoter assay), When preparing the draft, we only focused on the differences. We speculated that the abnormal results might be caused by the use of different batches of plasmids. We are sorry to make these mistakes. After precise determination of plasmid concentration, we re-performed all of the luciferase assays on 293T cells, and each assay did three replicates. Expectedly, compared with SC cells, we acquired the similar phenotypes of promoter and enhancer assays in 293T cells (**Fig. 1 and 2**). **We have updated the corresponding figures in the revised manuscript.** Thanks again for the reviewer's preciseness.

Fig. 1 | The luciferase assays on 293T cells. Luciferase reporter assays utilizing vectors harboring the rs4728142-containing sequences (GRCh37/hg19 chr7: 128,573,817–128,574,098) with different alleles (G or A) or the upstream regulatory element (GRCh37/hg19 chr7: 128,573,081–128,573,816, termed Flank) for enhancer (**a**, corresponding to Fig. 2c in the revised manuscript) or promoter assay (**b**, corresponding to Fig. 2d in the revised manuscript) in 293T (n = 3). pGL3-Basic vector without a promoter was used for the promoter assay, and pGL3-Promoter vector with SV40 promoter was used for the enhancer assay. Luciferase signals were normalized to renilla signals. Luciferase reporter assays utilizing vectors containing the integrated sequences (GRCh37/hg19 chr7: 128,573,081–128,574,098) harboring the upstream regulatory element and rs4728142-containing sequence with different alleles (G or A) for enhancer (**c**, corresponding to Fig. 2e in the revised manuscript) or promoter assay (**d**, corresponding to Fig. 2f in the revised manuscript) in 293T (n = 3). Luciferase reporter assays utilizing vectors harboring the rs4728142-containing sequences (GRCh37/hg19 chr7: 128,573,817–128,574,098) with different alleles (the non-risk allele G or the risk allele A) or the upstream regulatory element (GRCh37/hg19 chr7: 128,573,081–128,573,816, termed Flank) (**e**, corresponding to Extended Data Fig. 4e in the revised manuscript), or the integrated sequences (GRCh37/hg19 chr7: 128,573,081–128,574,098) harboring the upstream enhancer and rs4728142-containing region with different alleles (G or A) (**f**, corresponding to Extended Data Fig. 4f in the revised manuscript) for enhancer assay in 293T (n = 3). Promoter luciferase assays for the putative *IRF5*-short promoter in 293T cells (**g**, corresponding to Extended Data Fig. 6d in the revised manuscript) (n = 3). Data are represented as the means ± standard deviations (SD), and unpaired two-tailed Student's *t*-test is used to calculate *P*-values in **a–g**: *, *P* < 0.05; **, *P* < 0.01; ***, *P* < 0.001; ****, *P* < 0.0001.

3. In addition to the above in vitro luciferase reporter assays, can the authors count sequencing reads over rs4728142 in the ChIP-seq assays (Figure 2a)? Given that SC cells are heterozygous for rs4728142, we could expect to see allelic imbalance if the rs4728142-containing region shows a truly allele-specific enhancer activity. In parallel, allele-specific ChIP-qPCR and ChIP-Sanger sequencing could be further applied to corroborate in vitro reporter assays. In a similar vein, the analysis can be applied to the chromatin data of Extended Data Fig. 2c/d and 3a if the given cell lines are heterozygous for rs4728142. These analyses may inform us if this allele-specific enhancer activity is prevalent in vivo, and further corroborate its allele-specific interaction with the *IRF5* downstream alternative promoter in the following session.

Response: Thanks for the great suggestions. To investigate whether the rs4728142-containing region shows a truly allele-specific enhancer activity, we first applied allelic imbalance analysis using ChIP-seq of H3K4me1, H3K27ac, H3K4me3, and ATAC-seq, based on our SC cells with heterozygous genotype at rs4728142. Expectedly, we noticed that rs4728142 shows distinct allelic imbalance for both H3K4me1 and H3K27ac marks, in which risk allele A exhibits more read coverage than allele G (**Fig. 1**). In addition, by querying our QTLbase database¹¹, we found that rs4728142 has been identified as histone

modification QTLs (e.g., H3K27ac QTLs) in Monocytes²¹ (**Fig. 2**), further suggesting a prevalent allele-specific enhancer activity at rs4728142.

According to the reviewer's suggestions, we also performed the allele-specific ChIP-qPCR and ChIP-Sanger sequencing for H3K4me1 and H3K27ac on the rs4728142-containing region in SC cells. As the result, allele A at the rs4728142-containing region has more H3K4me1 and H3K27ac enrichment than allele G (all, $P < 0.001$) (**Fig. 3a**). Sequencing of the ChIP-PCR products amplified at the rs4728142 locus detected increased allele-specific binding for rs4728142-A when immunoprecipitated by anti-H3K4me1 and anti-H3K27ac antibody (**Fig. 3b**), indicating allele A of rs4728142 may has more enhancer activity than allele G in SC cells.

We have added the above results into Extended Data Fig. 3, the related description (L225–240), and Figure legend (L1721–1733) in the revised manuscript.

Fig. 1 | Allelic imbalance analysis of rs4728142 based on H3K4me1 and H3K27ac ChIP-seq data of SC cells with heterozygous genotype. For H3K4me1, rs4728142 allele A receives 90% read coverage than 10% in allele G. For H3K27ac, rs4728142 allele A receives 70% read coverage than 30% in allele G.

Fig. 2 | Histone modification QTLs evidence of rs4728142 among different blood cell types in QTLbase. rs4728142 is a H3K27ac QTL in Monocytes, a H3K4me1 QTL in Neutrophils, and a H3K4me3 QTL in Lymphocytes.

Fig. 3 | H3K4me1 and H3K27ac enriched at the rs4728142 locus in an allele-specific manner. **a**, ChIP enrichments of H3K4me1 and H3K27ac at the rs4728142-containing region in SC cells as determined by allele-specific ChIP-qPCR ($n = 3$). **b**, Sanger sequencing chromatograms of H3K4me1 and H3K27ac ChIP-qPCR at the rs4728142 locus. Data are represented as the means \pm standard deviations (SD), and unpaired two-tailed Student's t -test is used to calculate P -values in **a**: *, $P < 0.05$; **, $P < 0.01$; ***, $P < 0.001$; ****, $P < 0.0001$.

4. In Figure 3 and the relevant supporting data, the authors proved an allele-specific interaction between the rs4728142-containing region and the IRF5 downstream alternative promoter. The referee would suggest using RACE experiments to clone the full length of the short and long IRF5 transcripts despite that the authors discussed this shortcoming in the discussion. This is useful to specify the consequence of IRF5 promoter usage via the SNP region contributing to the expression of specific IRF5 isoforms. Moreover, the authors may perform similar enhancer reporter assays with the rs4728142 region targeting IRF5 alternative promoters instead of SV40 in the pGL3 construct.

Response: Thanks for the constructive suggestions. As the reviewer mentioned, we have tried to clone the full length of the short and long *IRF5* transcripts using the RACE experiments (Invitrogen, 5' RACE System for Rapid Amplification of cDNA Ends, Version 2.0, Cat. 18374-058). Unfortunately, after many attempts (replacing amplification primers, exploring annealing template, and so on), we did not obtain specific full-length bands. Determining the full-length of the short and long *IRF5* transcripts will help reveal the specific functions of different *IRF5* transcripts, which is of great significance for revealing the mechanism of *IRF5*-related autoimmune diseases. In future study, as the reviewer's suggestion, we will continue to try to clone the full length of the short- and long-*IRF5* transcripts, and further explore their functions. Thanks again for the reviewer's valuable suggestions.

As the reviewer mentioned, to further validate our findings, we inserted the corresponding sequences into the pGL3-*IRF5*-short promoter vector (**Fig. 2a** and **Fig. 3a**). As the result, we acquired the similar results with the SV40 promoter, and the phenotypes were lost given rs4728142 allele deletion (KO) (**Fig. 2b–e** and **Fig. 3b–e**), further indicating the regulatory potential of rs4728142-containing sequence.

We have added these results into Extended Data Fig. 7 and 8, the related description (L325–327), and figure legend (L1787–1815) in the revised manuscript.

Fig. 1 | rs4728142 affects the activity of a promoter-like enhancer adjacent to *IRF5* promoter. Strategies of luciferase reporter assays. All types of sequences were cloned into upstream of *IRF5*-short promoter in the pGL3 vector for enhancer activity detection (a). Luciferase reporter assays utilizing vectors harboring the rs4728142-containing sequences (GRCh37/hg19 chr7: 128,573,817–128,574,098) with different alleles (G or A), rs4728142 allele knockout (KO), or the upstream regulatory element (GRCh37/hg19 chr7: 128,573,081–128,573,816, termed Flank) for enhancer assay in 293T (n = 3) (b) and SC cells (n = 3) (c). Luciferase reporter assays utilizing vectors harboring the integrated sequences (GRCh37/hg19 chr7: 128,573,081–128,574,098) harboring the upstream regulatory element and rs4728142-containing sequence with different alleles (G or A) or rs4728142-KO for enhancer assay in 293T (n = 3) (d) and SC cells (n = 3) (e). Data are represented as the means \pm standard deviations (SD), and unpaired two-tailed Student's *t*-test is used to calculate *P*-values in b–e: *, *P* < 0.05; **, *P* < 0.01; ***, *P* < 0.001; ****, *P* < 0.0001.

Fig. 2 | rs4728142 affects the activity of a promoter-like enhancer adjacent to *IRF5* promoter. **a**, Strategies of luciferase reporter assays. All types of sequences were cloned into downstream of *IRF5*-short promoter in the pGL3 vector for enhancer activity detection. **b** and **c**, Luciferase reporter assays utilizing vectors harboring the rs4728142-containing sequences (GRCh37/hg19 chr7: 128,573,817–128,574,098) with different alleles (G or A), rs4728142-KO, or the upstream regulatory element (GRCh37/hg19 chr7: 128,573,081–128,573,816, termed Flank) for enhancer assay in 293T (n = 3) (**b**) and SC cells (n = 3) (**c**). Luciferase reporter assays utilizing vectors harboring the integrated sequences (GRCh37/hg19 chr7: 128,573,081–128,574,098) harboring the upstream regulatory element and rs4728142-containing sequence with different alleles (G or A) or rs4728142-KO for enhancer assay in 293T (n = 3) (**d**) and SC cells (n = 3) (**e**). Data are represented as the means \pm standard deviations (SD), and unpaired two-tailed Student's *t*-test is used to calculate *P*-values in **b–e**: *, *P* < 0.05; **, *P* < 0.01; ***, *P* < 0.001; ****, *P* < 0.0001.

5. In Figure 4d/f, why are *IRF5*-short or -long but not *IRF5* expression so sensitive to CRISPRa, whereas all of *IRF5* transcript expression are sensitive to CRISPRi for

downregulation? Did the authors independently replicate the experiments? It appears that both IRF5 and its short transcript (Figure 4f/g) were downregulated upon targeted deletion of the SNP site (sKO), suggesting the site is likely to be important for the chromatin interaction. Similar enhancer reporter assays as suggested above may be applied with the SNP site deleted genomic sequences. Moreover, the authors could utilize the sKO cells (Figure 4f/g) to pinpoint whether the rs4728142 and surrounding nucleotides are essential for chromatin looping via similar 3C experiments as described above. Without defining the full length of the IRF5-short or -long, how did the authors design the transcript-specific qPCR primers? In Figure 4h, can the authors specify the IDs of IRF5-short and -long transcripts?

Response: We thank the reviewer for the constructive comments, which will help us optimize the related experiments and results. As the reviewer mentioned, *IRF5* gene expression was not so sensitive to CRISPRa instead of the expression of *IRF5*-short or -long transcripts. To validate the phenomenon, we re-performed the CRISPRa and CRISPRi assays. Similar to the previous results, when the enhancer was activated by CRISPRa, both the *IRF5*-short and *IRF5*-long transcripts were markedly upregulated (all, $P < 0.001$) (**Fig. 1a**), as for *IRF5* gene, the expression level is also gained. On the contrary, the transcript expression levels were significantly downregulated when the enhancer was inhibited by CRISPRi (all, $P < 0.001$) (**Fig. 1b**). We have updated this result in the revised manuscript. Meanwhile, we had applied with the SNP site deleted genomic sequences in the luciferase assays (as mentioned above, **Question4: Fig. 2 and 3**). **We have updated Fig. 1a and b into Fig. 4d and e in the revised manuscript.**

According to the reviewer's suggestion, we performed the 3C assays on SC-sKO cells. Consequently, compared with WT SC cells, the rs4728142-containing region in SC-sKO cells has a weaker interaction with R5 (between *IRF5*-long and *IRF5*-short promoters, Fig. 3b in the revised manuscript) ($P < 0.0001$) (**Fig. 2**), suggesting deletion of rs4728142 can disrupt the loop formation between *IRF5*-long promoter and *IRF5*-short promoter; and the rs4728142-associated enhancer has a stronger interaction with R2 (*IRF5*-long promoter core region, Fig. 4b in the revised manuscript) ($P < 0.001$) and a weaker interaction with R5 (*IRF5*-short promoter core region, Fig. 4b in the revised manuscript) ($P < 0.001$) (**Fig. 3**), indicating deletion of rs4728142 can enhance the loop switching from the promoter of *IRF5*-short isoforms to the promoter of *IRF5*-long isoforms. **We have added the results into Extended Data Fig. 10a and b, and related description (L443–464) in the revised manuscript.**

We designed the transcript-specific qPCR primers accord to the previous studies^{22,23}. Graham et al. designed the specific primers for *IRF5* exon 1A, exon 1B, and exon 1C, and found that the rs2004640 T allele is associated with expression of variant exon 1B *IRF5* mRNA transcripts²³ (**Fig. 4, borrowed from the original paper**). Nordang et al. found that the IFN pathway gene, *IRF5*, is a common susceptibility factor for several rheumatic and autoimmune diseases, and risk variants are correlated with expression of alternative *IRF5* transcripts in thymus implying a regulatory role²² (**Fig. 5, borrowed from the original paper**). In these studies, the authors designed the transcript-specific RT-qPCR primers to

detect the different *IRF5* transcripts (Total *IRF5*, Exon 1A, Exon 1B, and Exon 1C) expression, and not accurately define the different *IRF5* transcripts. In our study, to better explain how rs4728142 different alleles regulate the expression of *IRF5*, we used their verified primers for different *IRF5* transcripts, and defined the long-*IRF5* (detected by the Exon 1A specific primers) and short-*IRF5* (detected by the Exon 1C specific primers) transcripts according to the interaction region found by our 3C/4C experiments, which offers us a good idea to clarify the rs4728142 regulatory mechanism in our study. **We have added the related instructions and references for the used primers in the revised manuscript (L331).**

The transcripts of *IRF5* are heterogeneously expressed in different immune cells. Among over 15 known *IRF5* transcripts, many can translate the same IRF5 full-length protein, but some transcripts may translate different forms of truncated proteins, even some cannot be translated and function as non-coding RNAs. These different forms of IRF5 protein could play different biological functions, and the immune effects on the disease process may be a result of one transcript or the interaction of multiple transcripts. Thus, the putative mechanism underlying the disease-causal effect of different *IRF5* isoforms on different autoimmune diseases is a complex and biologically significant issue which requires long-term exploration in the future.

Fig. 1 | *IRF5* transcript expression in the rs4728142-associated enhancer activation or inhibition SC cells as determined by RT-qPCR. sgRNAs targeting the rs4728142-associated enhancer were used in CRISPR activation (CRISPRa) (a) (n = 3) and CRISPR interference (CRISPRi) assays (b) (n = 3). Data are represented as the means \pm standard deviations (SD), and unpaired two-tailed Student's *t*-test is used to calculate *P*-values in a and b: *, *P* < 0.05; **, *P* < 0.01; ***, *P* < 0.001; ****, *P* < 0.0001.

Fig. 2 | The 3C results between rs4728142-containing region and the *IRF5* downstream alternative promoter in SC and SC-sKO cells. Enrichment quantification of 3C assay at the rs4728142-containing region normalized to control in SC and SC-sKO cells ($n = 3$). **a**, Electrophoresis results of 3C assay for the rs4728142-containing region in SC and SC-sKO cells. **b**, Relative interaction frequencies between the rs4728142-containing region and representative *DpnII* cutting fragments indicated in Fig. 3b in the revised manuscript. Data are represented as the means \pm standard deviations (SD), and unpaired two-tailed Student's *t*-test is used to calculate *P*-values in **b**: *, $P < 0.05$; **, $P < 0.01$; ***, $P < 0.001$; ****, $P < 0.0001$.

Fig. 3 | The 3C results between rs4728142-associated enhancer and the *IRF5* promoter in SC and SC-sKO cells. Enrichment quantification of 3C assay at the rs4728142-associated enhancer normalized to control in SC and SC-sKO cells (n = 3). **a**, Electrophoresis results of 3C assay for the rs4728142-containing region in SC and SC-sKO cells. **b**, Relative interaction frequencies between the rs4728142-associated enhancer and representative *DpnII* cutting fragments indicated in Fig. 4b in the revised manuscript. Data are represented as the means \pm standard deviations (SD), and unpaired two-tailed Student's *t*-test is used to calculate *P*-values in **b**: *, *P* < 0.05; **, *P* < 0.01; ***, *P* < 0.001; ****, *P* < 0.0001.

Fig. 4 | The rs2004640 T allele is associated with expression of variant exon 1B *IRF5* mRNA transcripts²³. **a**, mRNA isoforms of *IRF5*. Three sets of isoforms derive from three alternative promoters in the *IRF5* 5' region. The location of the exons encoding the DNA-binding, PEST and protein interaction domains, as well as the 3' untranslated region, are annotated. Protein translation begins 10 bp from the 5' end of exon 2 at a consensus ATG. Shown in the box is the location of the rs2004640 SNP, 2 bp downstream of exon 1B. Two polyadenylation sites are present in the *IRF5* 3' UTR, and the lengths of the 3' UTRs for V5, V6, V7 and V8 are unknown. Exon/intron structure is not shown to scale. **b**, RT-PCR analysis of total RNA purified from human PBMCs. Representative results for individuals with defined rs2004640 genotypes are shown. Exon 1B transcripts are found only in individuals carrying the rs2004640 T allele. The identities of most of the bands observed in these reactions correlate with known isoforms of the gene, confirmed by sequencing. **c**, Summary of TaqMan real-time quantitative RT-PCR analysis for transcripts associated with exons 1A, 1B and 1C. Each bar represents mean \pm s.e.m. expression level. n = 8 SLE cases for each genotype. Similar data were obtained for normal controls (data not shown).

ΔC_t values were calculated from duplicate samples normalized to human $\beta 2$ -microglobulin and were converted to linear scale. **These results were derived from the original paper.**

Fig. 5 | Relative expression of *IRF5* transcripts in thymic tissue according to genotype distribution²². **a**, Correlation of rs2004640 with total *IRF5*. **b**, Correlation of rs2004640 with exon 1A transcripts. **c**, Correlation of rs2004640 with exon1B transcripts. **d**, Correlation of CGGGG-indel with exon 1A transcripts. Samples were run in triplicate and relative mRNA expression was calculated using the ΔC_t method. Total *IRF5* were normalized to *RNaseP* and exon 1A and 1B transcripts were normalized to $\beta 2M$. Significant effects were investigated by unpaired two-tailed *t*-tests. Error bars represent mean (SEM). The genotype distribution of thymic samples is shown in parentheses.

6. It appears that ZBTB3 is a transcriptional regulator to transform the causal effect of the rs4728142 site. The sKO cells with breakdown of ZBTB3 binding site at rs4728142 for 3C analysis may prove if ZBTB3 directly involve in chromatin interaction. Moreover, the authors could design CHIP-3C assay to further prove if ZBTB3 is directing the chromatin looping. For ZBTB3 CHIP results, Figure S6c - the authors need to explain clearly how the CHIP-qPCR results were normalized as the enrichment of CTCF and ZBTB3 is extremely high, whereas CHIP-seq signals showing in Figure 5d appears not very strong, in particular at the rs4728142 region. Additional independent replication of the assays is encouraging. Similar CHIP-qPCR assays for CTCF and ZBTB3 (Figure S6c) could be applied to the sKO cells, confirming whether the rs4728142 site matching ZBTB3 motif is essential for its chromatin association. Figure 5j: it appears that the KO is not very efficient. The authors could perform shRNA and/or siRNA against ZBTB3 knockdown assays to observe if the results are consistent with the KO?

Response: Thanks for these constructive comments. According to the reviewer's suggestion, we first performed the 3C assays on SC-sKO cells. As the result, compared with SC cells, the rs4728142-containing region in SC-sKO cells has a weaker interaction with R5 (between *IRF5*-long promoter and *IRF5*-short promoter, Fig. 3b in the revised manuscript) ($P < 0.0001$) (**Fig. 1**), indicating deletion of rs4728142-containing region can disrupt the loop formation between rs4728142-containing region and the *IRF5* downstream alternative promoter; and the rs4728142-associated enhancer has a stronger interaction with R2 (*IRF5*-long promoter core region, Fig. 4b in the revised manuscript) ($P < 0.001$) and a weaker interaction with R5 (*IRF5*-short promoter core region, Fig. 4b in the revised manuscript) ($P < 0.001$) (**Fig. 2**), indicating deletion of rs4728142-containing region can promote the loop switching from the promoter of *IRF5*-short isoforms to the promoter of *IRF5*-long isoforms.

Next, to further test whether ZBTB3 is directing the chromatin looping, we performed 3C assays on the ZBTB3-knockout (KO) SC cells and ZBTB3-overexpression (OE) SC cells instead of ChIP-3C assay. As expected, ZBTB3-KO significantly weakened the interaction between rs4728142 and R5 (between *IRF5*-long and *IRF5*-short promoters, Fig. 3b in the revised manuscript) ($P < 0.0001$) (**Fig. 3**), and switched the rs4728142-associated enhancer from R6 (*IRF5*-short promoter core region, Fig. 4b in the revised manuscript) ($P < 0.01$) to R2 (*IRF5*-long promoter core region, Fig. 4b in the revised manuscript) ($P < 0.01$) (**Fig. 4**). On the contrary, ZBTB3-OE significantly strengthened the interaction between rs4728142 and R5 (between *IRF5*-long and *IRF5*-short *IRF5* promoters, Fig. 3b in the revised manuscript) ($P < 0.01$) (**Fig. 5**), and switched the rs4728142-associated enhancer from R2 (*IRF5*-long promoter core region, Fig. 4b in the revised manuscript) ($P < 0.05$) to R6 (*IRF5*-short promoter core region, Fig. 4b in the revised manuscript) ($P < 0.01$) (**Fig. 6**). **We have added these results into Extended Data Fig. 10, and the related description (L443–464) in the revised manuscript.**

Further, we re-performed the ChIP-seq and ChIP-qPCR for ZBTB3/RAD21 on the rs4728142-containing region, we did three replicates for each experiment. As the result, three ChIP-seq replicates were merged for peak calling, yielding 1647 peaks (MACS P -value < 0.05) for ZBTB3 and 15,344 peaks (MACS P -value < 0.05) for RAD21. The rs4728142-containing region receives a gained signal and obvious peak for both ZBTB3 (chr7:128,573,630–128,574,431, P -value = 0.036) and RAD21 (chr7:128,573,221–128,574,431, P -value = 0.037) (**Fig. 7**). **We have updated our new result into Fig. 5f in the revised manuscript.** Sequencing of the ChIP-qPCR products amplified at the rs4728142 locus demonstrated increased allele-specific binding for rs4728142-A when immunoprecipitated by anti-RAD21 antibody (**Fig. 8a and b**). **We have updated Fig. 8a and b into Fig. 5g and Extended Data Fig. 9c in the revised manuscript.** Next, to confirm whether the rs4728142 site matching ZBTB3 motif is essential for its chromatin association, we used ChIP-qPCR to detect the binding of ZBTB3, RAD21, and CTCF at the rs4728142 locus in SC-sKO cells. Expectedly, when destroying rs4728142, the bindings of ZBTB3, RAD21, and CTCF are nearly lost (**Fig. 8c**), indicating the DNA sequences at rs4728142 site is essential for chromatin association of the investigated factors.

To obtain a complete ZBTB3 knockout (KO) cell line, we subcloned the ZBTB3-KO cell line we acquired, and then carried out western blotting for ZBTB3 expression and RT-qPCR for the *IRF5* gene, *IRF5*-short, and *IRF5*-long transcripts. Western blotting results showed that ZBTB3 was almost completely knocked out (**Fig. 9a**). RT-qPCR results showed that the expression of both *IRF5* gene and *IRF5*-short transcripts were significantly down-regulated ($P < 0.01$), while that of *IRF5*-long transcripts were significantly up-regulated ($P < 0.01$) (**Fig. 9b**), which is consistent with our previous result in the manuscript. We have updated our new results in the revised manuscript. Besides, we used shRNA technology to silence ZBTB3 mRNA and observed similar results with the ZBTB3-KO results (**Fig. 9a and c**). We have updated the ZBTB3-KO results in Fig. 5i of the revised manuscript. **We have updated the ZBTB3-KO related data into Fig. 5i in the revised manuscript.**

Fig. 1 | The 3C results between rs4728142-containing region and the *IRF5* downstream alternative promoter in SC and SC-sKO cells. Enrichment quantification of 3C assay at the rs4728142-containing region normalized to control in SC and SC-sKO cells ($n = 3$). **a**, Electrophoresis results of 3C assay for the rs4728142-containing region in SC and SC-sKO cells. **b**, Relative interaction frequencies between the rs4728142-containing region and representative *DpnII* cutting fragments indicated in Fig. 3b in the revised manuscript. Data are represented as the means \pm standard deviations (SD), and unpaired two-tailed Student's *t*-test is used to calculate *P*-values in **b**: *, $P < 0.05$; **, $P < 0.01$; ***, $P < 0.001$; ****, $P < 0.0001$.

Fig. 2 | The 3C results between rs4728142-associated enhancer and the *IRF5* promoter in SC and SC-sKO cells. Enrichment quantification of 3C assay at the rs4728142-associated enhancer normalized to control in SC and SC-sKO cells (n = 3). **a**, Electrophoresis results of 3C assay for the rs4728142-containing region in SC and SC-sKO cells. **b**, Relative interaction frequencies between the rs4728142-associated enhancer and representative *DpnII* cutting fragments indicated in Fig. 4b in the revised manuscript. Data are represented as the means \pm standard deviations (SD), and unpaired two-tailed Student's *t*-test is used to calculate *P*-values in **b**: *, *P* < 0.05; **, *P* < 0.01; ***, *P* < 0.001; ****, *P* < 0.0001.

Fig. 3 | The 3C results between rs4728142-containing region and the *IRF5* downstream alternative promoter in SC and ZBTB3-KO SC cells. Enrichment quantification of 3C assay at the rs4728142-containing region normalized to control in SC and ZBTB3-KO SC cells (n = 3). **a**, Electrophoresis results of 3C assay for the rs4728142-containing region in SC and ZBTB3-KO SC cells. **b**, Relative interaction frequencies between the rs4728142-containing region and representative *DpnII* cutting fragments indicated in Fig. 3b in the revised manuscript. Data are represented as the means \pm standard deviations (SD), and unpaired two-tailed Student's *t*-test is used to calculate *P*-values in **b**: *, *P* < 0.05; **, *P* < 0.01; ***, *P* < 0.001; ****, *P* < 0.0001.

Fig. 4 | The 3C results between rs4728142-associated enhancer and the *IRF5* promoter in SC and ZBTB3-KO SC cells. Enrichment quantification of 3C assay at the rs4728142-associated enhancer normalized to control in SC and ZBTB3-KO SC cells (n = 3). **a**, Electrophoresis results of 3C assay for the rs4728142-associated enhancer in SC and ZBTB3-KO SC cells. **b**, Relative interaction frequencies between the rs4728142-associated enhancer and representative *DpnII* cutting fragments indicated in Fig. 4b in the revised manuscript. Data are represented as the means \pm standard deviations (SD), and unpaired two-tailed Student's *t*-test is used to calculate *P*-values in **b**: *, *P* < 0.05; **, *P* < 0.01; ***, *P* < 0.001; ****, *P* < 0.0001.

Fig. 5 | The 3C results between rs4728142-containing region and the *IRF5* downstream alternative promoter in SC and ZBTB3-OE SC cells. Enrichment quantification of 3C assay at the rs4728142-containing region normalized to control in SC and ZBTB3-OE SC cells (n = 3). **a**, Electrophoresis results of 3C assay for the rs4728142-containing region in SC and ZBTB3-OE SC cells. **b**, Relative interaction frequencies between the rs4728142-containing region and representative *DpnII* cutting fragments indicated in Fig. 3b in the revised manuscript. Data are represented as the means \pm standard deviations (SD), and unpaired two-tailed Student's *t*-test is used to calculate *P*-values in **b**: *, *P* < 0.05; **, *P* < 0.01; ***, *P* < 0.001; ****, *P* < 0.0001.

Fig. 6 | The 3C results between rs4728142-associated enhancer and the *IRF5* promoter in SC and ZBTB3-OE SC cells. Enrichment quantification of 3C assay at the rs4728142-associated enhancer normalized to control in SC and ZBTB3-OE SC cells (n = 3). **a**, Electrophoresis results of 3C assay for the rs4728142-associated enhancer in SC and ZBTB3-OE SC cells. **a**, Relative interaction frequencies between the rs4728142-associated enhancer and representative *DpnII* cutting fragments indicated in Fig. 4b in the revised manuscript. Data are represented as the means \pm standard deviations (SD), and unpaired two-tailed Student's *t*-test is used to calculate *P*-values in **b**: *, *P* < 0.05; **, *P* < 0.01; ***, *P* < 0.001; ****, *P* < 0.0001.

Fig. 7 | ZBTB3, RAD21 and CTCF ChIP-seq signals at the *IRF5* nearby region in SC cells. Red dotted line denotes the rs4728142 location. Highlighted boxes denote the rs4728142-containing 3C/4C viewpoint (left red box) and its highest interaction frequency region in the *IRF5* downstream alternative promoter (right red box).

Fig. 8 | CTCF, ZBTB3, and RAD21 enriched at the rs4728142 locus in SC cells. **a**, ChIP enrichments of CTCF, ZBTB3, and RAD21 at the rs4728142-containing region in SC cells as determined by ChIP-qPCR ($n = 3$). **b**, Sanger sequencing chromatograms of RAD21 ChIP-qPCR at the rs4728142 locus. **c**, ChIP enrichments of CTCF, ZBTB3, and RAD21 at the rs4728142-containing region in sKO SC cells as determined by ChIP-qPCR ($n = 3$). Data are represented as the means \pm standard deviations (SD), and unpaired two-tailed Student's *t*-test is used to calculate *P*-values in **a** and **c**: *, $P < 0.05$; **, $P < 0.01$; ***, $P < 0.001$; ****, $P < 0.0001$.

Fig. 9 | ZBTB3 mediates expression switching of *IRF5-short* and *IRF5-long* transcripts in SC cells. **a**, Western blotting for detecting the ZBTB3 in ZBTB3-KO and ZBTB3-silenced SC cells. **b** and **c**, RT-qPCR for detecting the *IRF5* transcript expression in ZBTB3-KO SC cells compared with the unedited SC cells (Control) ($n = 3$) (**b**) and in ZBTB3-silenced SC cells compared with the control SC cells ($n = 3$) (**c**). Data are represented as the means \pm standard deviations (SD), and unpaired two-tailed Student's *t*-test is used to calculate *P*-values in **b** and **c**: *, $P < 0.05$; **, $P < 0.01$; ***, $P < 0.001$; ****, $P < 0.0001$.

7. An important finding of the work is to define a novel mechanism underlying the rs4728142-risk allele A containing enhancer in upregulating *IRF5-short* while inhibiting *IRF5-long* transcript. Thus, the authors could compare their functional difference, e.g. RNA-seq plus CHIP-seq of *IRF5* isoforms. Current results of M1 polarization observed in rs4728142 sKO cells are virtually reflecting known biological roles of *IRF5*. The authors could back to check whether the relevant previous studies were indeed for the *IRF5-short* transcript as defined in this work.

Response: Thanks for the constructive suggestions. At first, we tried to perform *IRF5* CHIP-seq experiment on SC and sKO SC cells, but it was not successful. Then, we have purchased several *IRF5* antibodies (CST, Abcam, etc.) which spent a long time due to the impact of the epidemic. Unfortunately, after many attempts, no positive results (low enrichment and low FRiP score) have been obtained.

Alternatively, to investigate the functional difference between the *IRF5-short* and *IRF5-long* transcripts, we designed the sgRNAs targeting *IRF5-short* or *IRF5-long* promoter, and

used CRISPRa technology to specifically activate the *IRF5*-short or *IRF5*-long transcripts. As the result, sgRNA targeting *IRF5*-short promoter significantly promotes the expression of *IRF5* gene ($P < 0.0001$) and *IRF5*-short transcripts ($P < 0.0001$) without affecting the expression of *IRF5*-long transcripts, while sgRNA targeting *IRF5*-long promoter significantly promotes the expression of *IRF5* gene ($P < 0.001$) and *IRF5*-long transcripts ($P < 0.0001$) without altering the expression of *IRF5*-short transcripts (**Fig. 1a**). Next, we induced the *IRF5*-short promoter-activated or *IRF5*-long promoter-activated SC cells to M1 macrophages, and detected the macrophage M1 polarization markers (*ATF3*, *COX2*, *INDO*, *SLC7A5*, and *CCR7*) by RT-qPCR. As the result, all M1 polarization markers in both the induced-SC cells were significantly increased (all, $P < 0.001$) (**Fig. 1b**), inferring that both *IRF5*-short and *IRF5*-long transcripts could influence M1 macrophage polarization in the same way.

The *IRF5* transcripts are heterogeneously expressed in different immune cells. Among over 15 known *IRF5* transcripts, many can translate the same IRF5 full-length protein, but some transcripts may translate different forms of truncated proteins, even some cannot be translated and function as non-coding RNAs. These different forms of IRF5 protein could play different biological functions, but it is hard to precisely define the full lengthen of *IRF5*-short and *IRF5*-long transcripts. We used the transcript-specific RT-qPCR primers to detect the *IRF5*-short and *IRF5*-long transcripts based on the previous studies^{22,23} (**Question 5**). These studies only focused on the relationship between SNP (rs2004640) and the differential expression of different *IRF5* transcripts in autoimmune diseases, and the role of different *IRF5* transcripts in M1 macrophage polarization were not mentioned. Matijia Hedl and Clara Abraham et al. found that rs2004640 *IRF5* risk-associated allele carriers secrete increased cytokines upon individual or synergistic PRR stimulation in a gene dose- and ligand dose-dependent manner in both monocyte-derived dendritic cells and monocyte-derived macrophages⁷. Previous studies²⁴⁻²⁶ found that IRF5 regulates proinflammatory M1 macrophage polarization, which contributes to the elevation of the inflammatory and immune response. However, these studies focused on the function of IRF5 protein and its related SNP in M1 macrophage polarization, not involved in the specific *IRF5* transcripts. Up to now, we have not found related reports on the relationship between different *IRF5* transcripts and M1 macrophage polarization.

Fig. 1 | Both *IRF5*-short and *IRF5*-long transcripts promote M1 macrophage polarization. a, RT-qPCR for *IRF5* gene, *IRF5*-short, and *IRF5*-long transcripts in *IRF5*-short promoter-activated or *IRF5*-long promoter-activated SC cells. **b**, RT-qPCR for M1 polarization markers (*ATF3*, *COX2*, *INDO*, *SLC7A5*, and *CCR7*) in *IRF5*-short transcripts-

activated or *IRF5*-long transcripts-activated induced-SC cells (n = 3). Data are represented as the means \pm standard deviations (SD), and unpaired two-tailed Student's *t*-test is used to calculate *P*-values in **a** and **b**: *, *P* < 0.05; **, *P* < 0.01; ***, *P* < 0.001; ****, *P* < 0.0001.

8. Minor comments: Line 114: altering SNVs in known immune-related disease genes, such as *IFIH1* – encourage to provide proper references; Lines 196-199: Extended Data Fig. 2b/c need to be changed in order.

Response: Thanks for the reviewer's reminding! We are sorry we did not quote the appropriate references and used wrong order graphs in our first manuscript, and we have provided the proper reference²⁷ (PMID: 28974542, this paper published by Anthony J Sadler introduces in detail the role of MDA5 (encoded by *IFIH1*) in the development of autoimmune diseases) and changed Extended data. Fig. 2b/c in order in the Extended Data Fig. 2 of the revised manuscript.

Reference:

- 1 Shchetynsky, K. *et al.* Discovery of new candidate genes for rheumatoid arthritis through integration of genetic association data with expression pathway analysis. *Arthritis Res Ther* **19**, 19, doi:10.1186/s13075-017-1220-5 (2017).
- 2 Mistry, P. *et al.* Transcriptomic, epigenetic, and functional analyses implicate neutrophil diversity in the pathogenesis of systemic lupus erythematosus. *Proc Natl Acad Sci U S A* **116**, 25222-25228, doi:10.1073/pnas.1908576116 (2019).
- 3 Wang, C. *et al.* Genome-wide profiling of target genes for the systemic lupus erythematosus-associated transcription factors IRF5 and STAT4. *Ann Rheum Dis* **72**, 96-103, doi:10.1136/annrheumdis-2012-201364 (2013).
- 4 Geeven, G., Teunissen, H., de Laat, W. & de Wit, E. peakC: a flexible, non-parametric peak calling package for 4C and Capture-C data. *Nucleic Acids Res* **46**, e91, doi:10.1093/nar/gky443 (2018).
- 5 Perez, M. *et al.* Genetic Predisposition for Immune System, Hormone, and Metabolic Dysfunction in Myalgic Encephalomyelitis/Chronic Fatigue Syndrome: A Pilot Study. *Front Pediatr* **7**, 206, doi:10.3389/fped.2019.00206 (2019).
- 6 Atri, C., Guerfali, F. Z. & Laouini, D. Role of Human Macrophage Polarization in Inflammation during Infectious Diseases. *Int J Mol Sci* **19**, doi:10.3390/ijms19061801 (2018).
- 7 Hedl, M. & Abraham, C. IRF5 risk polymorphisms contribute to interindividual variance in pattern recognition receptor-mediated cytokine secretion in human monocyte-derived cells. *J Immunol* **188**, 5348-5356, doi:10.4049/jimmunol.1103319 (2012).
- 8 de Laat, W. & Dekker, J. 3C-based technologies to study the shape of the genome. *Methods* **58**, 189-191, doi:10.1016/j.ymeth.2012.11.005 (2012).
- 9 Schmiedel, B. J. *et al.* Impact of Genetic Polymorphisms on Human Immune Cell Gene Expression. *Cell* **175**, 1701-1715 e1716, doi:10.1016/j.cell.2018.10.022 (2018).
- 10 Ota, M. *et al.* Dynamic landscape of immune cell-specific gene regulation in immune-mediated diseases. *Cell* **184**, 3006-3021 e3017, doi:10.1016/j.cell.2021.03.056 (2021).
- 11 Zheng, Z. *et al.* QTLbase: an integrative resource for quantitative trait loci across multiple human molecular phenotypes. *Nucleic Acids Res* **48**, D983-D991, doi:10.1093/nar/gkz888

- (2020).
- 12 Tolhuis, B., Palstra, R. J., Splinter, E., Grosveld, F. & de Laat, W. Looping and interaction between hypersensitive sites in the active beta-globin locus. *Mol Cell* **10**, 1453-1465, doi:10.1016/S1097-2765(02)00781-5 (2002).
- 13 Rao, S. S. *et al.* A 3D map of the human genome at kilobase resolution reveals principles of chromatin looping. *Cell* **159**, 1665-1680, doi:10.1016/j.cell.2014.11.021 (2014).
- 14 van de Werken, H. J. *et al.* 4C technology: protocols and data analysis. *Methods Enzymol* **513**, 89-112, doi:10.1016/B978-0-12-391938-0.00004-5 (2012).
- 15 Splinter, E., de Wit, E., van de Werken, H. J., Klous, P. & de Laat, W. Determining long-range chromatin interactions for selected genomic sites using 4C-seq technology: from fixation to computation. *Methods* **58**, 221-230, doi:10.1016/j.ymeth.2012.04.009 (2012).
- 16 van de Werken, H. J. *et al.* Robust 4C-seq data analysis to screen for regulatory DNA interactions. *Nature methods* **9**, 969-972, doi:10.1038/nmeth.2173 (2012).
- 17 Vos, E. S. M. *et al.* Interplay between CTCF boundaries and a super enhancer controls cohesin extrusion trajectories and gene expression. *Mol Cell* **81**, 3082-3095 e3086, doi:10.1016/j.molcel.2021.06.008 (2021).
- 18 Allahyar, A. *et al.* Robust detection of translocations in lymphoma FFPE samples using targeted locus capture-based sequencing. *Nat Commun* **12**, 3361, doi:10.1038/s41467-021-23695-8 (2021).
- 19 Nagano, T. *et al.* Comparison of Hi-C results using in-solution versus in-nucleus ligation. *Genome Biol* **16**, doi:ARTN 175 10.1186/s13059-015-0753-7 (2015).
- 20 Denker, A. & de Laat, W. The second decade of 3C technologies: detailed insights into nuclear organization. *Genes Dev* **30**, 1357-1382, doi:10.1101/gad.281964.116 (2016).
- 21 Chen, L. *et al.* Genetic Drivers of Epigenetic and Transcriptional Variation in Human Immune Cells. *Cell* **167**, 1398-1414 e1324, doi:10.1016/j.cell.2016.10.026 (2016).
- 22 Nordang, G. B. *et al.* Interferon regulatory factor 5 gene polymorphism confers risk to several rheumatic diseases and correlates with expression of alternative thymic transcripts. *Rheumatology (Oxford)* **51**, 619-626, doi:10.1093/rheumatology/ker364 (2012).
- 23 Graham, R. R. *et al.* A common haplotype of interferon regulatory factor 5 (IRF5) regulates splicing and expression and is associated with increased risk of systemic lupus erythematosus. *Nat Genet* **38**, 550-555, doi:10.1038/ng1782 (2006).
- 24 Hedl, M., Yan, J. & Abraham, C. IRF5 and IRF5 Disease-Risk Variants Increase Glycolysis and Human M1 Macrophage Polarization by Regulating Proximal Signaling and Akt2 Activation. *Cell Rep* **16**, 2442-2455, doi:10.1016/j.celrep.2016.07.060 (2016).
- 25 Corbin, A. L. *et al.* IRF5 guides monocytes toward an inflammatory CD11c(+) macrophage phenotype and promotes intestinal inflammation. *Sci Immunol* **5**, doi:10.1126/sciimmunol.aax6085 (2020).
- 26 Dalmas, E. *et al.* Irf5 deficiency in macrophages promotes beneficial adipose tissue expansion and insulin sensitivity during obesity. *Nat Med* **21**, 610-618, doi:10.1038/nm.3829 (2015).
- 27 Sadler, A. J. The role of MDA5 in the development of autoimmune disease. *J Leukoc Biol* **103**, 185-192, doi:10.1189/jlb.4MR0617-223R (2018).

REVIEWERS' COMMENTS

Reviewer #1 (Remarks to the Author):

The authors have provided additional evidence to support their assumptions and have made significant improvements in this revision. The authors have adequately addressed all of the criticisms raised in the previous review.

Reviewer #2 (Remarks to the Author):

Since the initial submission, the authors have included a large volume of new data supporting their claims, and addressed my concerns, which I believe has substantially improved the manuscript. I feel comfortable to recommend publication at this stage, but would like the authors to address the following - mostly minor - points:

Lines 193-194: "We found that autoimmune diseases (e.g., UC and SLE) caused by rs4728142 were enriched in granulocyte-macrophage progenitors,..."

I would caution against stating that these autoimmune diseases are caused by rs4728142...are the authors confident of implying causality?

Lines 196 - 198: "Moreover, the genomic region surrounding the pleiotropic variant rs4728142 obtains low chromatin accessibility in monocytes and has absent open chromatin in other hematopoietic cells."

Consider changing to "Moreover, the genomic region surrounding the pleiotropic variant rs4728142 shows low chromatin accessibility in monocytes, whereas the region's chromatin is not accessible in other hematopoietic cells."

Lines 239 and 240: "...indicating allele A of rs4728142 may has more enhancer activity than allele G in SC cells."

'...may have...' or simply '...has...'

Lines 263-264: "We further cloned the regulatory sequences into downstream of the luciferase reporter gene (Extended Data Fig. 4d)."

Better without 'into': "We further cloned the regulatory sequences downstream of the luciferase reporter gene (Extended Data Fig. 4d)."

Figure 4i: can the authors include a western blot with ZBTB3 antibody in the ZBTB3-OE line and control, so that readers get an impression of the ZBTB3 levels in the over-expressing line relative to normal?

Lines 514-515: "In addition, we detected the expression of key M1 proinflammatory cytokine."

Perhaps better: "In addition, we analysed the expression of key M1 proinflammatory cytokines."

Lines 520-521: "...transcripts and IRF5 gene, and down-regulating the expression of the IRF5-long transcripts,

and then contributes to the M1 macrophage polarization."

Perhaps better: "...transcripts and IRF5 gene, and down-regulating the expression of the IRF5-long transcripts,

which then contributes to the M1 macrophage polarization."

Lines 538-539: "...particularly for the variants in the non-coding genomic region."

Perhaps better: "...particularly for the variants in the non-coding genome."

Lines 581-582: "...the disease-causal variant rs4728142 hijacks enhancers to interact with different alternative promoters of the IRF5 gene."

Do the authors mean 'enhancers' or 'enhancer' here?

Lines 581-590: "Public ATAC-seq, DNase-seq, and histone modification ChIP-seq and ours of the relevant cell types show that rs4728142 is located on the boundary of an open chromatin and could be associated with nucleosome bound proteins."

Not clear to me – I think this sentence needs rewording/restructuring.

Lines 592-593: "To decipher such superficial unreasonable phenomena,..."

Again, not clear to me. Do the authors mean to say something along the lines of "To reconcile these at first glance paradoxical observations,..."

Lines 618-622: "Besides, this ZBTB3-mediated binding and loop formation is accompanied by weak CTCF and RAD21 occupancy. Similarly, a recent paper has revealed that, at sites where CTCF binding corresponds to gene activation, there is significant enrichment for ZBTB3, and ZBTB3 might be a CTCF cofactor that mediates gene activation by loop formation in regenerating liver (74)."

It is not clear to me how this can be seen as 'similarly'? The first sentence states that at the IRF5 locus, ZBTB3 binding is accompanied by weak CTCF and cohesin occupancy. In the following sentence, the authors cite a reference (74) which describes co-occurrence of strong binding of ZBTB3 and CTCF. Does this not call for a 'In contrast,...' rather than a 'Similarly,...'?

Line 626: "...CTCF-dependent structural regulator in regulating enhancer-promoter interaction."

Perhaps better to avoid regulator in regulating by simplifying to "...CTCF-dependent structural regulator of enhancer-promoter interactions."?

Lines 630-630: "...there are more than four IRF5 alternative promoters."

Please specify how many alternative IRF5 promoters current knowledge has identified. Or say 'at least four...'

Reviewer #3 (Remarks to the Author):

In this revised version, the authors have improved the significance of studying pleiotropic SNP on the enhancer/insulator of IRF5 and showed a close link between alternative transcription start sites of IRF5 and multiple autoimmune diseases. Although the analysis pipeline haven't identify many pleiotropic SNPs candidates, the whole picture of the story is intact and good enough for publish. It is well appreciated that, for Q3 and Q4, authors did all the necessary experiments to improve the quality of the manuscript. Well done! I don't have further suggestions on this version and suggest to accept this version.

Reviewer #4 (Remarks to the Author):

The authors have done extensive amount of work for the revision thus I spent more time to look into the revised manuscript and delayed.

We thank you and reviewers for evaluating our manuscript again and giving us valuable comments. We have carefully addressed all the reviewers' comments below and in the manuscript accordingly. Accompanying this letter, please find the final revised version of our manuscript.

Reviewer #1 (Remarks to the Author):

The authors have provided additional evidence to support their assumptions and have made significant improvements in this revision. The authors have adequately addressed all of the criticisms raised in the previous review.

Response: Thanks again for the reviewer's excellent suggestions for our manuscript.

Reviewer #2 (Remarks to the Author):

Since the initial submission, the authors have included a large volume of new data supporting their claims, and addressed my concerns, which I believe has substantially improved the manuscript. I feel comfortable to recommend publication at this stage, but would like the authors to address the following - mostly minor - points:

Response: Thanks for the reviewer's rigorous suggestions which further improving the quality of our manuscript.

Lines 193-194: "We found that autoimmune diseases (e.g., UC and SLE) caused by rs4728142 were enriched in granulocyte-macrophage progenitors,..."

I would caution against stating that these autoimmune diseases are caused by rs4728142...are the authors confident of implying causality?

Response: Thanks for the suggestion. According to the reviewer's suggestion, we have changed "We found that autoimmune diseases (e.g., UC and SLE) caused by rs4728142 were enriched in granulocyte-macrophage progenitors,..." into "We found that autoimmune diseases (e.g., UC and SLE) associated with rs4728142 were enriched in granulocyte-macrophage progenitors,..." in the revised manuscript.

Lines 196 – 198: "Moreover, the genomic region surrounding the pleiotropic variant rs4728142 obtains low chromatin accessibility in monocytes and has absent open chromatin in other hematopoietic cells."

Consider changing to "Moreover, the genomic region surrounding the pleiotropic variant rs4728142 shows low chromatin accessibility in monocytes, whereas the region's chromatin is not accessible in other hematopoietic cells."

Response: Thanks for the professional suggestion. I'm sorry we didn't describe it clearly. According to the reviewer's suggestion, we have changed "Moreover, the genomic region surrounding the pleiotropic variant rs4728142 obtains low chromatin accessibility in monocytes and has absent open chromatin in other hematopoietic cells." into "Moreover, the genomic region surrounding the pleiotropic variant rs4728142 shows low chromatin

accessibility in monocytes, whereas the region's chromatin is not accessible in other hematopoietic cells." in the revised manuscript.

Lines 239 and 240: "...indicating allele A of rs4728142 may has more enhancer activity than allele G in SC cells."

'...may have...' or simply '...has...'

Response: Thanks for the suggestion. I'm sorry for our negligence! We have changed "may has" into "may have" in the revised manuscript.

Lines 263-264: "We further cloned the regulatory sequences into downstream of the luciferase reporter gene (Extended Data Fig. 4d)."

Better without 'into': "We further cloned the regulatory sequences downstream of the luciferase reporter gene (Extended Data Fig. 4d)."

Response: Thanks for the suggestion. According to the reviewer's suggestion, we have changed "We further cloned the regulatory sequences into downstream of the luciferase reporter gene (Extended Data Fig. 4d)." into "We further cloned the regulatory sequences downstream of the luciferase reporter gene (Extended Data Fig. 4d)." in the revised manuscript.

Figure 4i: can the authors include a western blot with ZBTB3 antibody in the ZBTB3-OE line and control, so that readers get an impression of the ZBTB3 levels in the over-expressing line relative to normal?

Response: Thanks for the good suggestion. According to the reviewer's suggestion, we have performed western blot with ZBTB3 antibody in the ZBTB3-OE line and control.

Lines 514-515: "In addition, we detected the expression of key M1 proinflammatory cytokine."

Perhaps better: "In addition, we analysed the expression of key M1 proinflammatory cytokines."

Response: Thanks for the suggestion. According to the reviewer's suggestion, we have changed "In addition, we detected the expression of key M1 proinflammatory cytokine." into "In addition, we analysed the expression of key M1 proinflammatory cytokines." in the revised manuscript.

Lines 520-521: "...transcripts and IRF5 gene, and down-regulating the expression of the IRF5-long transcripts,
and then contributes to the M1 macrophage polarization."

Perhaps better: "...transcripts and IRF5 gene, and down-regulating the expression of the IRF5-long transcripts,
which then contributes to the M1 macrophage polarization."

Response: Thanks for the good suggestion. According to the reviewer's suggestion, we have changed "...transcripts and IRF5 gene, and down-regulating the expression of the IRF5-long transcripts, and then contributes to the M1 macrophage polarization." into "...transcripts and IRF5 gene, and down-regulating the expression of the IRF5-long

transcripts, which then contributes to the M1 macrophage polarization.” in the revised manuscript.

Lines 538-539: “...particularly for the variants in the non-coding genomic region.”

Perhaps better: “...particularly for the variants in the non-coding genome.”

Response: Thanks for the suggestion. According to the reviewer’s suggestion, we have changed “...particularly for the variants in the non-coding genomic region.” into “...particularly for the variants in the non-coding genome.” in the revised manuscript.

Lines 581-582: “...the disease-causal variant rs4728142 hijacks enhancers to interact with different alternative promoters of the IRF5 gene.”

Do the authors mean ‘enhancers’ or ‘enhancer’ here?

Response: Thanks for the reminding. According to the reviewer’s suggestion, we have changed “enhancers” into “enhancer” in the revised manuscript.

Lines 581-590: “Public ATAC-seq, DNase-seq, and histone modification ChIP-seq and ours of the relevant cell types show that rs4728142 is located on the boundary of an open chromatin and could be associated with nucleosome bound proteins.”

Not clear to me – I think this sentence needs rewording/restructuring.

Response: Thanks for the good suggestion. We have changed “Public ATAC-seq, DNase-seq, and histone modification ChIP-seq and ours of the relevant cell types show that rs4728142 is located on the boundary of an open chromatin and could be associated with nucleosome bound proteins.” into “Public and our ATAC-seq, DNase-seq, and histone modification ChIP-seq of the relevant cell types show that rs4728142 is located on the boundary of an open chromatin and could be associated with the nucleosome-bound proteins” in the revised manuscript.

Lines 592-593: “To decipher such superficial unreasonable phenomena,...”

Again, not clear to me. Do the authors mean to say something along the lines of “To reconcile these at first glance paradoxical observations,...”

Response: Thanks for the good suggestion. According to the reviewer’s suggestion, we have changed “To decipher such superficial unreasonable phenomena,...” into “To reconcile these at first glance paradoxical observations,...” in the revised manuscript.

Lines 618-622: “Besides, this ZBTB3-mediated binding and loop formation is accompanied by weak CTCF and RAD21 occupancy. Similarly, a recent paper has revealed that, at sites where CTCF binding corresponds to gene activation, there is significant enrichment for ZBTB3, and ZBTB3 might be a CTCF cofactor that mediates gene activation by loop formation in regenerating liver (74).”

It is not clear to me how this can be seen as ‘similarly’? The first sentence states that at the IRF5 locus, ZBTB3 binding is accompanied by weak CTCF and cohesin occupancy. In the following sentence, the authors cite a reference (74) which describes co-occurrence of strong binding of ZBTB3 and CTCF. Does this not call for a ‘In contrast,...’ rather than a

'Similarly,...'?

Response: Thanks for the reminding. I'm sorry we didn't describe it clearly. We have changed "Similarly" into "In contrast" in the revised manuscript.

Line 626: "...CTCF-dependent structural regulator in regulating enhancer–promoter interaction."

Perhaps better to avoid regulator in regulating by simplifying to "...CTCF-dependent structural regulator of enhancer–promoter interactions."?

Response: Thanks for the good suggestion. According to the reviewer's suggestion, we have changed "...CTCF-dependent structural regulator in regulating enhancer–promoter interaction." into "...CTCF-dependent structural regulator of enhancer–promoter interactions." in the revised manuscript.

Lines 630-630: "...there are more than four IRF5 alternative promoters."

Please specify how many alternative IRF5 promoters current knowledge has identified. Or say 'at least four...'

Response: Thanks for the rigorous suggestion. I'm sorry we didn't describe it clearly. According to the reviewer's suggestion, we have changed "...there are more than four IRF5 alternative promoters." into "...there are at least four IRF5 alternative promoters." in the revised manuscript.

Reviewer #3 (Remarks to the Author):

In this revised version, the authors have improved the significance of studying pleiotropic SNP on the enhancer/insulator of IRF5 and showed a close link between alternative transcription start sites of IRF5 and multiple autoimmune diseases. Although the analysis pipeline haven't identify many pleiotropic SNPs candidates, the whole picture of the story is intact and good enough for publish. It is well appreciated that, for Q3 and Q4, authors did all the necessary experiments to improve the quality of the manuscript. Well done! I don't have further suggestions on this version and suggest to accept this version.

Response: Thanks again for the reviewer's highlight response and excellent suggestions for improving our manuscript.

Reviewer #4 (Remarks to the Author):

The authors have done extensive amount of work for the revision thus I spent more time to look into the revised manuscript and delayed.

Response: Thanks again for the reviewer's excellent suggestions for improving our manuscript.